# Partially Observable RL with B-Stability: Unified Structural Condition and Sharp Sample-Efficient Algorithms

**Fan Chen**
Peking University
chern@pku.edu.cn

**Yu Bai**[*]
Salesforce Research
yu.bai@salesforce.com

**Song Mei**[*]
UC Berkeley
songmei@berkeley.edu

## Abstract

Partial Observability—where agents can only observe partial information about the true underlying state of the system—is ubiquitous in real-world applications of Reinforcement Learning (RL). Theoretically, learning a near-optimal policy under partial observability is known to be hard in the worst case due to an exponential sample complexity lower bound. Recent work has identified several tractable subclasses that are learnable with polynomial samples, such as Partially Observable Markov Decision Processes (POMDPs) with certain revealing or decodability conditions. However, this line of research is still in its infancy, where (1) unified structural conditions enabling sample-efficient learning are lacking; (2) existing sample complexities for known tractable subclasses are far from sharp; and (3) fewer sample-efficient algorithms are available than in fully observable RL.

This paper advances all three aspects above for Partially Observable RL in the general setting of Predictive State Representations (PSRs). First, we propose a natural and unified structural condition for PSRs called *B-stability*. B-stable PSRs encompasses the vast majority of known tractable subclasses such as weakly revealing POMDPs, low-rank future-sufficient POMDPs, decodable POMDPs, and regular PSRs. Next, we show that any B-stable PSR can be learned with polynomial samples in relevant problem parameters. When instantiated in the aforementioned subclasses, our sample complexities improve substantially over the current best ones. Finally, our results are achieved by three algorithms simultaneously: Optimistic Maximum Likelihood Estimation, Estimation-to-Decisions, and Model-Based Optimistic Posterior Sampling. The latter two algorithms are new for sample-efficient learning of POMDPs/PSRs. We additionally design a variant of the Estimation-to-Decisions algorithm to perform sample-efficient *all-policy model estimation* for B-stable PSRs, which also yields guarantees for reward-free learning as an implication.

## 1 Introduction

Partially Observable Reinforcement Learning (RL)—where agents can only observe partial information about the true underlying state of the system—is ubiquitous in real-world applications of RL such as robotics (Akkaya et al., 2019), strategic games (Brown & Sandholm, 2018; Vinyals et al., 2019; Berner et al., 2019), economic simulation (Zheng et al., 2020), and so on. Partially observable RL defies standard efficient approaches for learning and planning in the fully observable case (e.g. those based on dynamical programming) due to the non-Markovian nature of the observations (Jaakkola et al., 1994), and has been a hard challenge for RL research.

Theoretically, it is well-established that learning in partial observable RL is statistically hard in the worst case—In the standard setting of Partially Observable Markov Decision Processes (POMDPs), learning a near-optimal policy has an exponential sample complexity lower bound in the horizon length (Mossel & Roch, 2005; Krishnamurthy et al., 2016), which in stark contrast to fully observable MDPs where polynomial sample complexity is possible (Kearns & Singh, 2002; Jaksch et al.,

---

[*]Equal contribution.

Table 1: **Comparisons of sample complexities** for learning an $\varepsilon$ near-optimal policy in POMDPs and PSRs. Definitions of the problem parameters can be found in Section 3.2. The last three rows refer to the $m$-step versions of the problem classes (e.g. the third row considers $m$-step $\alpha_{\mathrm{rev}}$-revealing POMDPs). The current best results within the last four rows are due to Zhan et al. (2022); Liu et al. (2022a); Wang et al. (2022); Efroni et al. (2022) respectively[1]. All results are scaled to the setting with total reward in $[0, 1]$.

| Problem Class | Current Best | Ours |
|---|---|---|
| $\Lambda_{\mathsf{B}}$-stable PSR | - | $\widetilde{\mathcal{O}}\left(d_{\mathsf{PSR}} A U_A H^2 \log \mathcal{N}_\Theta \cdot \Lambda_{\mathsf{B}}^2/\varepsilon^2\right)$ |
| $\alpha_{\mathsf{psr}}$-regular PSR | $\widetilde{\mathcal{O}}\left(d_{\mathsf{PSR}}^4 A^4 U_A^9 H^6 \log(\mathcal{N}_\Theta O)/(\alpha_{\mathsf{psr}}^6 \varepsilon^2)\right)$ | $\widetilde{\mathcal{O}}\left(d_{\mathsf{PSR}} A U_A^2 H^2 \log \mathcal{N}_\Theta/(\alpha_{\mathsf{psr}}^2 \varepsilon^2)\right)$ |
| $\alpha_{\mathsf{rev}}$-revealing tabular POMDP | $\widetilde{\mathcal{O}}\left(S^4 A^{6m-4} H^6 \log \mathcal{N}_\Theta/(\alpha_{\mathsf{rev}}^4 \varepsilon^2)\right)$ | $\widetilde{\mathcal{O}}\left(S^2 A^m H^2 \log \mathcal{N}_\Theta/(\alpha_{\mathsf{rev}}^2 \varepsilon^2)\right)$ |
| $\nu$-future-suff. rank-$d_{\mathsf{trans}}$ POMDP | $\widetilde{\mathcal{O}}\left(d_{\mathsf{trans}}^4 A^{5m+3l+1} H^2 (\log \mathcal{N}_\Theta)^2 \cdot \nu^4 \gamma^2/\varepsilon^2\right)$ | $\widetilde{\mathcal{O}}\left(d_{\mathsf{trans}} A^{2m-1} H^2 \log \mathcal{N}_\Theta \cdot \nu^2/\varepsilon^2\right)$ |
| decodable rank-$d_{\mathsf{trans}}$ POMDP | $\widetilde{\mathcal{O}}\left(d_{\mathsf{trans}} A^m H^2 \log \mathcal{N}_\mathcal{G}/\varepsilon^2\right)$ | $\widetilde{\mathcal{O}}\left(d_{\mathsf{trans}} A^m H^2 \log \mathcal{N}_\Theta/\varepsilon^2\right)$ |

2010; Azar et al., 2017). A later line of work identifies various additional structural conditions or alternative learning goals that enable sample-efficient learning, such as reactiveness (Jiang et al., 2017), revealing conditions (Jin et al., 2020a; Liu et al., 2022c; Cai et al., 2022; Wang et al., 2022), decodability (Du et al., 2019; Efroni et al., 2022), and learning memoryless or short-memory policies (Azizzadenesheli et al., 2018; Uehara et al., 2022b).

Despite these progresses, research on sample-efficient partially observable RL is still at an early stage, with several important questions remaining open. First, to a large extent, existing tractable structural conditions are mostly identified and analyzed in a case-by-case manner and lack a more unified understanding. This question has just started to be tackled in the very recent work of Zhan et al. (2022), who show that sample-efficient learning is possible in the more general setting of Predictive State Representations (PSRs) (Littman & Sutton, 2001)—which include POMDPs as a special case—with a certain *regularity* condition. However, their regularity condition is defined in terms of additional quantities (such as "core matrices") not directly encoded in the definition of PSRs, which makes it unnatural in many known examples and unable to subsume important tractable problems such as decodable POMDPs.

Second, even in known sample-efficient problems such as revealing POMDPs (Jin et al., 2020c; Liu et al., 2022a), existing sample complexities involve large polynomial factors of relevant problem parameters that are likely far from sharp. Third, relatively few principles are known for designing sample-efficient algorithms in POMDPs/PSRs, such as spectral or tensor-based approaches (Hsu et al., 2012; Azizzadenesheli et al., 2016; Jin et al., 2020c), maximum likelihood or density estimation (Liu et al., 2022a; Wang et al., 2022; Zhan et al., 2022), or learning short-memory policies (Efroni et al., 2022; Uehara et al., 2022b). This contrasts with fully observable RL where the space of sample-efficient algorithms is much more diverse (Agarwal et al., 2019). It is an important question whether we can expand the space of algorithms for partially observable RL.

This paper advances all three aspects above for partially observable RL. We define *B-stablility*, a natural and general structural condition for PSRs, and design sharp algorithms for learning any B-stable PSR sample-efficiently. Our contributions can be summarized as follows.

- We identify a new structural condition for PSRs termed *B-stability*, which simply requires its *B-representation* (or observable operators) to be bounded in a suitable operator norm (Section 3.1). B-stable PSRs subsume most known tractable subclasses such as revealing POMDPs, decodable POMDPs, low-rank future-sufficient POMDPs, and regular PSRs (Section 3.2).

- We show that B-stable PSRs can be learned sample-efficiently by three algorithms simultaneously with sharp sample complexities (Section 4): Optimistic Maximum Likelihood Estimation (OMLE), Explorative Estimation-to-Decisions (EXPLORATIVE E2D), and Model-based Optimistic Posterior Sampling (MOPS). To our best knowledge, the latter two algorithms are first shown to be sample-efficient in partially observable RL.

- Our sample complexities improve substantially over the current best when instantiated in both regular PSRs (Section 4.1) and known tractable subclasses of POMDPs (Section 5). For example, for $m$-step $\alpha_{\mathsf{rev}}$-revealing POMDPs with $S$ latent states, our algorithms find an $\varepsilon$ near-optimal policy within $\widetilde{\mathcal{O}}\left(S^2 A^m \log \mathcal{N}/(\alpha_{\mathsf{rev}}^2 \varepsilon^2)\right)$ episodes of play (with $S^2/\alpha_{\mathsf{rev}}^2$ replaced by

---

[1]For $\nu$-future-sufficient POMDPs, Wang et al. (2022)'s sample complexity depends on $\gamma$, which is an additional $l$-*step past-sufficiency* parameter that they require.

$S\Lambda_\mathsf{B}^2$ if measured in B-stability), which improves significantly over the current best result of $\widetilde{\mathcal{O}}\left(S^4 A^{6m-4} \log \mathcal{N}/(\alpha_{\mathsf{rev}}^4 \varepsilon^2)\right)$. A summary of such comparisons is presented in Table 1.

- As a variant of the E2D algorithm, we design the ALL-POLICY MODEL-ESTIMATION E2D algorithm that achieves sample-efficient *all-policy model estimation*—and as an application, reward-free learning—for B-stable PSRs (Section 4.2 & Appendix H.2).

- Technically, our three algorithms rely on a unified sharp analysis of B-stable PSRs that involves a careful error decomposition in terms of its B-representation, along with a new generalized $\ell_2$-type Eluder argument, which may be of future interest (Appendix B).

**Related work** Our work is closely related to the long lines of work on sample-efficient learning of fully/partially observable RL (with/without function approximation), especially the lines of work on POMDPs and PSRs. We review these related works in Appendix A due to the space limit.

## 2 PRELIMINARIES

**Sequential decision processes with observations** An episodic sequential decision process is specified by a tuple $\left\{H, \mathcal{O}, \mathcal{A}, \mathbb{P}, \{r_h\}_{h=1}^H\right\}$, where $H \in \mathbb{Z}_{\geqslant 1}$ is the horizon length; $\mathcal{O}$ is the observation space with $|\mathcal{O}| = O$; $\mathcal{A}$ is the action space with $|\mathcal{A}| = A$; $\mathbb{P}$ specifies the transition dynamics, such that the initial observation follows $o_1 \sim \mathbb{P}_0(\cdot) \in \Delta(\mathcal{O})$, and given the *history* $\tau_h := (o_1, a_1, \cdots, o_h, a_h)$ up to step $h$, the observation follows $o_{h+1} \sim \mathbb{P}(\cdot|\tau_h)$; $r_h : \mathcal{O} \times \mathcal{A} \to [0,1]$ is the reward function at $h$-th step, which we assume is a known deterministic function of $(o_h, a_h)$.

A policy $\pi = \{\pi_h : (\mathcal{O} \times \mathcal{A})^{h-1} \times \mathcal{O} \to \Delta(\mathcal{A})\}_{h=1}^H$ is a collection of $H$ functions. At step $h \in [H]$, an agent running policy $\pi$ observes the observation $o_h$ and takes action $a_h \sim \pi_h(\cdot|\tau_{h-1}, o_h) \in \Delta(\mathcal{A})$ based on the history $(\tau_{h-1}, o_h) = (o_1, a_1, \ldots, o_{h-1}, a_{h-1}, o_h)$. The agent then receives their reward $r_h(o_h, a_h)$, and the environment generates the next observation $o_{h+1} \sim \mathbb{P}(\cdot|\tau_h)$ based on $\tau_h = (o_1, a_1, \cdots, o_h, a_h)$. The episode terminates immediately after the dummy observation $o_{H+1} = o_{\mathrm{dum}}$ is generated. We use $\Pi$ to denote the set of all deterministic policies, and identify $\Delta(\Pi)$ as both the set of all policies and all distributions over deterministic policies interchangeably. For any $(h, \tau_h)$, let $\mathbb{P}(\tau_h) := \prod_{h' \leqslant h} \mathbb{P}(o_{h'}|\tau_{h'-1})$, $\pi(\tau_h) := \prod_{h' \leqslant h} \pi_{h'}(a_{h'}|\tau_{h'-1}, o_{h'})$, and let $\mathbb{P}^\pi(\tau_h) := \mathbb{P}(\tau_h) \times \pi(\tau_h)$ denote the probability of observing $\tau_h$ (for the first $h$ steps) when executing $\pi$. The *value* of a policy $\pi$ is defined as the expected cumulative reward $V(\pi) := \mathbb{E}^\pi\left[\sum_{h=1}^H r_h(o_h, a_h)\right]$. We assume that $\sum_{h=1}^H r_h(o_h, a_h) \leqslant 1$ almost surely for any policy $\pi$.

**POMDPs** A Partially Observable Markov Decision Process (POMDP) is a special sequential decision process whose transition dynamics are governed by *latent states*. An episodic POMDP is specified by a tuple $\{H, \mathcal{S}, \mathcal{O}, \mathcal{A}, \{\mathbb{T}_h\}_{h=1}^H, \{\mathbb{O}_h\}_{h=1}^H, \{r_h\}_{h=1}^H, \mu_1\}$, where $\mathcal{S}$ is the latent state space with $|\mathcal{S}| = S$, $\mathbb{O}_h(\cdot|\cdot) : \mathcal{S} \to \Delta(\mathcal{O})$ is the emission dynamics at step $h$ (which we identify as an emission matrix $\mathbb{O}_h \in \mathbb{R}^{\mathcal{O} \times \mathcal{S}}$), $\mathbb{T}_h(\cdot|\cdot, \cdot) : \mathcal{S} \times \mathcal{A} \to \Delta(\mathcal{S})$ is the transition dynamics over the latent states (which we identify as transition matrices $\mathbb{T}_h(\cdot|\cdot, a) \in \mathbb{R}^{\mathcal{S} \times \mathcal{S}}$ for each $a \in \mathcal{A}$), and $\mu_1 \in \Delta(\mathcal{S})$ specifies the distribution of initial state. At each step $h$, given latent state $s_h$ (which the agent cannot observe), the system emits observation $o_h \sim \mathbb{O}_h(\cdot|s_h)$, receives action $a_h \in \mathcal{A}$ from the agent, emits the reward $r_h(o_h, a_h)$, and then transits to the next latent state $s_{h+1} \sim \mathbb{T}_h(\cdot|s_h, a_h)$ in a Markov fashion. Note that a POMDP can be fully described by the parameter $\theta := (\mathbb{T}, \mathbb{O}, \mu_1)$.

### 2.1 PREDICTIVE STATE REPRESENTATIONS

We consider Predictive State Representations (PSRs) (Littman & Sutton, 2001), a broader class of sequential decision processes that generalize POMDPs by removing the explicit assumption of latent states, but still requiring the system dynamics to be described succinctly by a core test set.

**PSR, core test sets, and predictive states** A *test* $t$ is a sequence of future observations and actions (i.e. $t \in \mathfrak{T} := \bigcup_{W \in \mathbb{Z}_{\geqslant 1}} \mathcal{O}^W \times \mathcal{A}^{W-1}$). For some test $t_h = (o_{h:h+W-1}, a_{h:h+W-2})$ with length $W \geqslant 1$, we define the probability of test $t_h$ being successful conditioned on (reachable) history $\tau_{h-1}$ as $\mathbb{P}(t_h|\tau_{h-1}) := \mathbb{P}(o_{h:h+W-1}|\tau_{h-1}; \mathrm{do}(a_{h:h+W-2}))$, i.e., the probability of observing $o_{h:h+W-1}$ if the agent deterministically executes actions $a_{h:h+W-2}$, conditioned on history $\tau_{h-1}$. We follow the convention that, if $\mathbb{P}^\pi(\tau_{h-1}) = 0$ for any $\pi$, then $\mathbb{P}(t|\tau_{h-1}) = 0$.

**Definition 1** (PSR, core test sets, and predictive states). *For any $h \in [H]$, we say a set $\mathcal{U}_h \subset \mathfrak{T}$ is a* core test set *at step $h$ if the following holds: For any $W \in \mathbb{Z}_{\geqslant 1}$, any possible future (i.e., test) $t_h = (o_{h:h+W-1}, a_{h:h+W-2}) \in \mathcal{O}^W \times \mathcal{A}^{W-1}$, there exists a vector $b_{t_h,h} \in \mathbb{R}^{\mathcal{U}_h}$ such that*

$$\mathbb{P}(t_h | \tau_{h-1}) = \langle b_{t_h,h}, [\mathbb{P}(t|\tau_{h-1})]_{t \in \mathcal{U}_h} \rangle, \qquad \forall \tau_{h-1} \in \mathcal{T}^{h-1} := (\mathcal{O} \times \mathcal{A})^{h-1}. \tag{1}$$

*We refer to the vector $\mathbf{q}(\tau_{h-1}) := [\mathbb{P}(t|\tau_{h-1})]_{t \in \mathcal{U}_h}$ as the* predictive state *at step $h$ (with convention $\mathbf{q}(\tau_{h-1}) = 0$ if $\tau_{h-1}$ is not reachable), and $\mathbf{q}_0 := [\mathbb{P}(t)]_{t \in \mathcal{U}_1}$ as the initial predictive state. A (linear) PSR is a sequential decision process equipped with a core test set $\{\mathcal{U}_h\}_{h \in [H]}$.*

The predictive state $\mathbf{q}(\tau_{h-1}) \in \mathbb{R}^{\mathcal{U}_h}$ in a PSR acts like a "latent state" that governs the transition $\mathbb{P}(\cdot | \tau_{h-1})$ through the linear structure (1). We define $\mathcal{U}_{A,h} := \{\mathbf{a} : (\mathbf{o}, \mathbf{a}) \in \mathcal{U}_h$ for some $\mathbf{o} \in \bigcup_{W \in \mathbb{N}^+} \mathcal{O}^W\}$ as the set of action sequences (possibly including an empty sequence) in $\mathcal{U}_h$, with $U_A := \max_{h \in [H]} |\mathcal{U}_{A,h}|$. Further define $\mathcal{U}_{H+1} := \{o_{\text{dum}}\}$ for notational simplicity. Throughout the paper, we assume the core test sets $(\mathcal{U}_h)_{h \in [H]}$ are known and the same within the PSR model class.

**B-representation** We define the *B-representation* of a PSR, a standard notion for PSRs (also known as the observable operators (Jaeger, 2000)).

**Definition 2** (B-representation). *A B-representation of a PSR with core test set $(\mathcal{U}_h)_{h \in [H]}$ is a set of matrices[2] $\{(\mathbf{B}_h(o_h, a_h) \in \mathbb{R}^{\mathcal{U}_{h+1} \times \mathcal{U}_h})_{h,o_h,a_h}, \mathbf{q}_0 \in \mathbb{R}^{\mathcal{U}_1}\}$ such that for any $0 \leqslant h \leqslant H$, policy $\pi$, history $\tau_h = (o_{1:h}, a_{1:h}) \in \mathcal{T}^h$, and core test $t_{h+1} = (o_{h+1:h+W}, a_{h+1:h+W-1}) \in \mathcal{U}_{h+1}$, the quantity $\mathbb{P}(\tau_h, t_{h+1})$, i.e. the probability of observing $o_{1:h+W}$ upon taking actions $a_{1:h+W-1}$, admits the decomposition*

$$\mathbb{P}(\tau_h, t_{h+1}) = \mathbb{P}(o_{1:h+W} | \mathrm{do}(a_{1:h+W-1})) = \mathbf{e}_{t_{h+1}}^\top \cdot \mathbf{B}_{h:1}(\tau_h) \cdot \mathbf{q}_0, \tag{2}$$

*where $\mathbf{e}_{t_{h+1}} \in \mathbb{R}^{\mathcal{U}_{h+1}}$ is the indicator vector of $t_{h+1} \in \mathcal{U}_{h+1}$, and*

$$\mathbf{B}_{h:1}(\tau_h) := \mathbf{B}_h(o_h, a_h) \mathbf{B}_{h-1}(o_{h-1}, a_{h-1}) \cdots \mathbf{B}_1(o_1, a_1).$$

It is a standard result (see e.g. Thon & Jaeger (2015)) that any PSR admits a B-representation, and the converse also holds—any sequential decision process admitting a B-representation on test sets $(\mathcal{U}_h)_{h \in [H]}$ is a PSR with core test set $(\mathcal{U}_h)_{h \in [H]}$ (Proposition D.1). However, the B-representation of a given PSR may not be unique. We also remark that the B-representation is used in the structural conditions and theoretical analyses only, and will not be explicitly used in our algorithms.

**Rank** An important complexity measure of a PSR is its *PSR rank* (henceforth also "rank").

**Definition 3** (PSR rank). *Given a PSR, its* PSR rank *is defined as $d_{\mathsf{PSR}} := \max_{h \in [H]} \mathrm{rank}(D_h)$, where $D_h := [\mathbf{q}(\tau_h)]_{\tau_h \in \mathcal{T}^h} \in \mathbb{R}^{\mathcal{U}_{h+1} \times \mathcal{T}^h}$ is the matrix formed by predictive states at step $h \in [H]$.*

The PSR rank measures the inherent dimension[3] of the space of predictive state vectors, which always admits the upper bound $d_{\mathsf{PSR}} \leqslant \max_{h \in [H]} |\mathcal{U}_h|$, but may in addition be much smaller.

**POMDPs as low-rank PSRs** As a primary example, all POMDPs are PSRs with rank at most $S$ (Zhan et al., 2022, Lemma 2). First, Definition 1 can be satisfied trivially by choosing $\mathcal{U}_h = \bigcup_{1 \leqslant W \leqslant H-h+1} \{(o_h, a_h, \ldots, o_{h+W-1})\}$ as the set of all possible tests, and $b_{t_h,h} = \mathbf{e}_{t_h} \in \mathbb{R}^{\mathcal{U}_h}$ as indicator vectors. For concrete subclasses of POMDPs, we will consider alternative choices of $(\mathcal{U}_h)_{h \in [H]}$ with much smaller cardinalities than this default choice. Second, to compute the rank (Definition 3), note that by the latent state structure of POMDPs, we have $\mathbb{P}(t_{h+1} | \tau_h) = \sum_{s_{h+1}} \mathbb{P}(t_{h+1} | s_{h+1}) \mathbb{P}(s_{h+1} | \tau_h)$ for any $(h, \tau_h, t_{h+1})$. Therefore, the associated matrix $D_h = [\mathbb{P}(t_{h+1} | \tau_h)]_{(t_{h+1}, \tau_h) \in \mathcal{U}_{h+1} \times \mathcal{T}^h}$ always has the following decomposition:

$$D_h = [\mathbb{P}(t_{h+1} | s_{h+1})]_{(t_{h+1}, s_{h+1}) \in \mathcal{U}_{h+1} \times \mathcal{S}} \times [\mathbb{P}(s_{h+1} | \tau_h)]_{(s_{h+1}, \tau_h) \in \mathcal{S} \times \mathcal{T}^h},$$

which implies that $d_{\mathsf{PSR}} = \max_{h \in [H]} \mathrm{rank}(D_h) \leqslant S$.

**Learning goal** We consider the standard PAC learning setting, where we are given a model class of PSRs $\Theta$ and interact with a ground truth model $\theta^\star \in \Theta$. Note that, as we do not put further restrictions on the parametrization, this setting allows any general function approximation for

---

[2]This definition can be generalized to continuous $\mathcal{U}_h$, where $\mathbf{B}_h(o_h, a_h) \in \mathcal{L}(L^1(\mathcal{U}_h), L^1(\mathcal{U}_{h+1}))$ are linear operators instead of (finite-dimensional) matrices.

[3]This definition using matrix ranks may be further relaxed, e.g. by considering the effective dimension.

the model class. For any model class $\Theta$, we define its (optimistic) covering number $\mathcal{N}_\Theta(\rho)$ for $\rho > 0$ in Definition C.4. Let $V_\theta(\pi)$ denote the value function of policy $\pi$ under model $\theta$, and $\pi_\theta := \arg\max_{\pi \in \Pi} V_\theta(\pi)$ denote the optimal policy of model $\theta$. The goal is to learn a policy $\widehat{\pi}$ that achieves small suboptimality $V_\star - V_{\theta^\star}(\widehat{\pi})$ within as few episodes of play as possible, where $V_\star := V_{\theta^\star}(\pi_{\theta^\star})$. We refer to an algorithm as sample-efficient if it finds an $\varepsilon$-near optimal policy within $\mathrm{poly}$(relevant problem parameters, $1/\varepsilon)^4$ episodes of play.

## 3 PSRs with B-stability

We begin by proposing a natural and general structural condition for PSR called *B-stability* (or also *stability*). We show that B-stable PSRs encompass and generalize a variety of existing tractable POMDPs and PSRs, and can be learned sample-efficiently as we show in the sequel.

### 3.1 The B-stability condition

For any PSR with an associated B-representation, we define its $\mathcal{B}$-operators $\{\mathcal{B}_{H:h}\}_{h \in [H]}$ as

$$\mathcal{B}_{H:h} : \mathbb{R}^{\mathcal{U}_h} \to \mathbb{R}^{(\mathcal{O} \times \mathcal{A})^{H-h+1}}, \qquad \mathbf{q} \mapsto \left[ \mathbf{B}_{H:h}(\tau_{h:H}) \cdot \mathbf{q} \right]_{\tau_{h:H} \in (\mathcal{O} \times \mathcal{A})^{H-h+1}}.$$

Operator $\mathcal{B}_{H:h}$ maps any predictive state $\mathbf{q} = \mathbf{q}(\tau_{h-1})$ at step $h$ to the vector $\mathcal{B}_{H:h}\mathbf{q} = (\mathbb{P}(\tau_{h:H}|\tau_{h-1}))_{\tau_{h:H}}$ which governs the probability of transitioning to all possible futures, by properties of the B-representation (cf. (17) & Corollary D.2). For each $h \in [H]$, we equip the image space of $\mathcal{B}_{H:h}$ with the $\Pi$-*norm*: For a vector $\mathbf{b}$ indexed by $\tau_{h:H} \in (\mathcal{O} \times \mathcal{A})^{H-h+1}$, we define

$$\|\mathbf{b}\|_\Pi := \max_{\bar{\pi}} \sum_{\tau_{h:H} \in (\mathcal{O} \times \mathcal{A})^{H-h+1}} \bar{\pi}(\tau_{h:H}) |\mathbf{b}(\tau_{h:H})|, \tag{3}$$

where the maximization is over all policies $\bar{\pi}$ starting from step $h$ (ignoring the history $\tau_{h-1}$) and $\bar{\pi}(\tau_{h:H}) = \prod_{h \le h' \le H} \bar{\pi}_{h'}(a_{h'}|o_{h'}, \tau_{h:h'-1})$. We further equip the domain $\mathbb{R}^{\mathcal{U}_h}$ with a *fused-norm* $\|\cdot\|_*$, which is defined as the maximum of $(1,2)$-norm and $\Pi'$-norm[5]:

$$\|\mathbf{q}\|_* := \max\{\|\mathbf{q}\|_{1,2}, \|\mathbf{q}\|_{\Pi'}\}, \tag{4}$$

$$\|\mathbf{q}\|_{1,2} := \left( \sum_{\mathbf{a} \in \mathcal{U}_{A,h}} \left( \sum_{\mathbf{o}:(\mathbf{o},\mathbf{a}) \in \mathcal{U}_h} |\mathbf{q}(\mathbf{o},\mathbf{a})| \right)^2 \right)^{1/2}, \qquad \|\mathbf{q}\|_{\Pi'} := \max_{\bar{\pi}} \sum_{t \in \overline{\mathcal{U}}_h} \bar{\pi}(t) |\mathbf{q}(t)|, \tag{5}$$

where $\overline{\mathcal{U}}_h := \{t \in \mathcal{U}_h : \nexists t' \in \mathcal{U}_h \text{ such that } t \text{ is a prefix of } t'\}$.

We now define the B-stability condition, which simply requires the $\mathcal{B}$-operators $\{\mathcal{B}_{H:h}\}_{h \in [H]}$ to have bounded operator norms from the fused-norm to the $\Pi$-norm.

**Definition 4** (B-stability). *A PSR is* B-stable with parameter $\Lambda_\mathrm{B} \ge 1$ *(henceforth also $\Lambda_\mathrm{B}$-stable) if it admits a B-representation with associated $\mathcal{B}$-operators $\{\mathcal{B}_{H:h}\}_{h \in [H]}$ such that*

$$\sup_{h \in [H]} \max_{\|\mathbf{q}\|_* = 1} \|\mathcal{B}_{H:h}\mathbf{q}\|_\Pi \le \Lambda_\mathrm{B}. \tag{6}$$

When using the B-stability condition, we will often take $\mathbf{q} = \mathbf{q}_1(\tau_{h-1}) - \mathbf{q}_2(\tau_{h-1})$ to be the difference between two predictive states at step $h$. Intuitively, Definition 4 requires that the propagated $\Pi$-norm error $\|\mathcal{B}_{H:h}(\mathbf{q}_1 - \mathbf{q}_2)\|_\Pi$ to be controlled by the original fused-norm error $\|\mathbf{q}_1 - \mathbf{q}_2\|_*$.

The fused-norm $\|\cdot\|_*$ is equivalent to the vector 1-norm up to a $|\mathcal{U}_{A,h}|^{1/2}$-factor (despite its seemingly involved form): We have $\|\mathbf{q}\|_* \le \|\mathbf{q}\|_1 \le |\mathcal{U}_{A,h}|^{1/2} \|\mathbf{q}\|_*$ (Lemma D.6), and thus assuming a relaxed condition $\max_{\|\mathbf{q}\|_1 = 1} \|\mathcal{B}_{H:h}\|_\Pi \le \Lambda$ will also enable sample-efficient learning of PSRs. However, we consider the fused-norm in order to obtain the sharpest possible sample complexity guarantees. Finally, all of our theoretical results still hold under a more relaxed (though less intuitive) *weak B-stability condition* (Definition D.4), with the same sample complexity guarantees. (See also the additional discussions in Appendix D.2.)

---

[4]For the $m$-step versions of our structural conditions, we allow an exponential dependence on $m$ but not $H$. Such a dependence is necessary, e.g. in $m$-step decodable POMDPs (Efroni et al., 2022).

[5]The $\Pi'$-norm is in general a semi-norm.

## 3.2 RELATION WITH KNOWN SAMPLE-EFFICIENT SUBCLASSES

We show that the B-stability condition encompasses many known structural conditions of PSRs and POMDPs that enable sample-efficient learning. Throughout, for a matrix $A \in \mathbb{R}^{m \times n}$, we define its operator norm $\|A\|_{p \to q} := \max_{\|x\|_p \leqslant 1} \|Ax\|_q$, and use $\|A\|_p := \|A\|_{p \to p}$ for shorthand.

**Weakly revealing POMDPs** (Jin et al., 2020a; Liu et al., 2022a) is a subclass of POMDPs that assumes the current latent state can be probabilistically inferred from the next $m$ emissions.

**Example 5** (Multi-step weakly revealing POMDPs). A POMDP is called $m$-step $\alpha_{\mathsf{rev}}$-*weakly revealing* (henceforth also "$\alpha_{\mathsf{rev}}$-revealing") with $\alpha_{\mathsf{rev}} \leqslant 1$ if $\max_{h \in [H-m+1]} \|\mathbb{M}_h^\dagger\|_{2 \to 2} \leqslant \alpha_{\mathsf{rev}}^{-1}$, where for $h \in [H-m+1]$, $\mathbb{M}_h \in \mathbb{R}^{\mathcal{O}^m \mathcal{A}^{m-1} \times \mathcal{S}}$ is the $m$-step emission-action matrix at step $h$, defined as

$$[\mathbb{M}_h]_{(\mathbf{o},\mathbf{a}),s} := \mathbb{P}(o_{h:h+m-1} = \mathbf{o}|s_h = s, a_{h:h+m-2} = \mathbf{a}), \forall (\mathbf{o},\mathbf{a}) \in \mathcal{O}^m \times \mathcal{A}^{m-1}, s \in \mathcal{S}. \quad (7)$$

We show that any $m$-step $\alpha_{\mathsf{rev}}$-weakly revealing POMDP is a $\Lambda_{\mathsf{B}}$-stable PSR with core test sets $\mathcal{U}_h = (\mathcal{O} \times \mathcal{A})^{\min\{m-1,H-h\}} \times \mathcal{O}$, and $\Lambda_{\mathsf{B}} \leqslant \sqrt{S}\alpha_{\mathsf{rev}}^{-1}$ (Proposition D.7). A similar result holds for the $\ell_1$ variant of the revealing condition (see Appendix D.3.1). $\Diamond$

When the transition matrix $\mathbb{T}_h$ of the POMDP has a low rank structure, Wang et al. (2022) show that a subspace-aware generalization of the $\ell_1$-revealing condition—the **future-sufficiency condition**—enables sample-efficient learning of POMDPs with possibly enormous state/observation spaces (see also Cai et al. (2022)). We consider the following generalized definition of future-sufficiency.

**Example 6** (Low-rank future-sufficient POMDPs). We say a POMDP has transition rank $d_{\mathsf{trans}}$ if for each $h \in [H-1]$, the transition kernel of the POMDP has rank at most $d_{\mathsf{trans}}$ (i.e. $\max_h \mathrm{rank}(\mathbb{T}_h) \leqslant d_{\mathsf{trans}}$). It is clear that low-rank POMDPs with transition rank $d_{\mathsf{trans}}$ has PSR rank $d_{\mathsf{PSR}} \leqslant d_{\mathsf{trans}}$.

A transition rank-$d_{\mathsf{trans}}$ (henceforth rank-$d_{\mathsf{trans}}$) POMDP is called $m$-*step $\nu$-future-sufficient* with $\nu \geqslant 1$, if for $h \in [H-1]$, there exists $\mathbb{M}_h^\natural \in \mathbb{R}^{\mathcal{S} \times \mathcal{U}_h}$ such that $\mathbb{M}_h^\natural \mathbb{M}_h \mathbb{T}_{h-1} = \mathbb{T}_{h-1}$ and $\|\mathbb{M}_h^\natural\|_{1 \to 1} \leqslant \nu$, where $\mathbb{M}_h$ is the $m$-step emission-action matrix defined in (7). [6]

We show that any $m$-step $\nu$-future sufficient rank-$d_{\mathsf{trans}}$ POMDP is a B-stable PSR with core test sets $\mathcal{U}_h = (\mathcal{O} \times \mathcal{A})^{\min\{m-1,H-h\}} \times \mathcal{O}$, $d_{\mathsf{PSR}} \leqslant d_{\mathsf{trans}}$, and $\Lambda_{\mathsf{B}} \leqslant \sqrt{A^{m-1}}\nu$ (Proposition D.12). $\Diamond$

**Decodable POMDPs** (Efroni et al., 2022), as a multi-step generalization of Block MDPs (Du et al., 2019), assumes the current latent state can be perfectly decoded from the recent $m$ observations.

**Example 7** (Multi-step decodable POMDPs). A POMDP is called $m$-*step decodable* if there exists (unknown) decoders $\phi^\star = \{\phi_h^\star\}_{h \in [H]}$, such that for every reachable trajectory $(s_1, o_1, a_1, \cdots, s_h, o_h)$ we have $s_h = \phi_h^\star(z_h)$, where $z_h = (o_{m(h)}, a_{m(h)}, \cdots, o_h)$ and $m(h) = \max\{h-m+1, 1\}$. We show that any $m$-step decodable POMDP is a B-stable PSR with core test sets $\mathcal{U}_h = (\mathcal{O} \times \mathcal{A})^{\min\{m-1,H-h\}} \times \mathcal{O}$ and $\Lambda_{\mathsf{B}} = 1$ (Proposition D.17). $\Diamond$

Finally, Zhan et al. (2022) define the following **regularity condition for general PSRs**.

**Example 8** (Regular PSRs). A PSR is called $\alpha_{\mathsf{psr}}$-*regular* if for all $h \in [H]$ there exists a *core matrix* $K_h \in \mathbb{R}^{\mathcal{U}_{h+1} \times \mathrm{rank}(D_h)}$, which is a column-wise sub-matrix of $D_h$ such that $\mathrm{rank}(K_h) = \mathrm{rank}(D_h)$ and $\max_{h \in [H]} \|K_h^\dagger\|_{1 \to 1} \leqslant \alpha_{\mathsf{psr}}^{-1}$. We show that any $\alpha_{\mathsf{psr}}$-regular PSR is $\Lambda_{\mathsf{B}}$-stable with $\Lambda_{\mathsf{B}} \leqslant \sqrt{U_A}\alpha_{\mathsf{psr}}^{-1}$ (Proposition D.18). $\Diamond$

We emphasize that B-stability not only encompasses $\alpha_{\mathsf{psr}}$-regularity, but is also strictly more expressive. For example, decodable POMDPs are not $\alpha_{\mathsf{psr}}$-regular unless with additional assumptions on $K_h^\dagger$ (Zhan et al., 2022, Section 6.5), whereas they are B-stable with $\Lambda_{\mathsf{B}} = 1$ (Example 7). Also, any $\alpha_{\mathsf{rev}}$-revealing POMDP is $\alpha_{\mathsf{psr}}$-regular with some $\alpha_{\mathsf{psr}}^{-1} < \infty$, but with $\alpha_{\mathsf{psr}}^{-1}$ potentially not polynomially bounded by $\alpha_{\mathsf{rev}}^{-1}$ (and other problem parameters) due to the restriction of $K_h$ being a column-wise sub-matrix of $D_h$; By contrast it is B-stable with $\Lambda_{\mathsf{B}} \leqslant \sqrt{S}\alpha_{\mathsf{rev}}^{-1}$ (Example 5).

## 4 LEARNING B-STABLE PSRs

In this section, we show that B-stable PSRs can be learned sample-efficiently, achieved by three model-based algorithms simultaneously. We instantiate our results to POMDPs in Section 5.

---

[6]It is straightforward to generalize this example to the case when $\mathcal{S}$ and $\mathcal{O}$ are infinite by replacing vectors with $L_1$ integrable functions, and matrices with linear operators between these spaces.

---

**Algorithm 1** OPTIMISTIC MAXIMUM LIKELIHOOD ESTIMATION (OMLE)

---

1: **Input:** Model class $\Theta$, parameter $\beta > 0$.
2: **Initialize:** $\Theta^1 = \Theta$, $\mathcal{D} = \{\}$.
3: **for** iteration $k = 1, \ldots, K$ **do**
4:     Set $(\theta^k, \pi^k) = \arg\max_{\theta \in \Theta^k, \pi} V_\theta(\pi)$.
5:     **for** $h = 0, \ldots, H-1$ **do**
6:         Set exploration policy $\pi_{h,\exp}^k := \pi^k \circ_h \text{Unif}(\mathcal{A}) \circ_{h+1} \text{Unif}(\mathcal{U}_{A,h+1})$.
7:         Execute $\pi_{h,\exp}^k$ to collect a trajectory $\tau^{k,h}$, and add $(\pi_{h,\exp}^k, \tau^{k,h})$ into $\mathcal{D}$.
8:     Update confidence set

$$\Theta^{k+1} = \left\{ \widehat{\theta} \in \Theta : \sum_{(\pi,\tau) \in \mathcal{D}} \log \mathbb{P}_{\widehat{\theta}}^\pi(\tau) \geqslant \max_{\theta \in \Theta} \sum_{(\pi,\tau) \in \mathcal{D}} \log \mathbb{P}_\theta^\pi(\tau) - \beta \right\}.$$

**Output:** $\widehat{\pi}_{\text{out}} := \text{Unif}(\{\pi^k\}_{k \in [K]})$.

---

### 4.1 OPTIMISTIC MAXIMUM LIKELIHOOD ESTIMATION (OMLE)

The OMLE algorithm is proposed by Liu et al. (2022a) for learning revealing POMDPs and adapted[7] by Zhan et al. (2022) for learning regular PSRs, achieving polynomial sample complexity (in relevant problem parameters) in both cases. We show that OMLE works under the broader condition of B-stability, with significantly improved sample complexities.

**Algorithm and theoretical guarantee** The OMLE algorithm (described in Algorithm 1) takes in a class of PSRs $\Theta$, and performs two main steps in each iteration $k \in [K]$:

1. (Optimism) Construct a confidence set $\Theta^k \subseteq \Theta$, which is a superlevel set of the log-likelihood of all trajectories within dataset $\mathcal{D}$ (Line 8). The policy $\pi^k$ is then chosen as the greedy policy with respect to the most optimistic model within $\Theta^k$ (Line 4).
2. (Data collection) Execute exploration policies $(\pi_{h,\exp}^k)_{0 \leqslant h \leqslant H-1}$, where each $\pi_{h,\exp}^k$ is defined via the $\circ_h$ notation as follows: Follow $\pi^k$ for the first $h-1$ steps, take a uniform action $\text{Unif}(\mathcal{A})$ at step $h$, take an action sequence sampled from $\text{Unif}(\mathcal{U}_{A,h+1})$ at step $h+1$, and behave arbitrarily afterwards (Line 6). All collected trajectories are then added into $\mathcal{D}$ (Line 7).

Intuitively, the concatenation of the current policy $\pi^k$ with $\text{Unif}(\mathcal{A})$ and $\text{Unif}(\mathcal{U}_{A,h+1})$ in Step 2 above is designed according to the structure of PSRs to foster exploration.

**Theorem 9** (Guarantee of OMLE). *Suppose every $\theta \in \Theta$ is $\Lambda_{\mathsf{B}}$-stable (Definition 4) and the true model $\theta^\star \in \Theta$ has rank $d_{\mathsf{PSR}} \leqslant d$. Then, choosing $\beta = C \log(\mathcal{N}_\Theta(1/KH)/\delta)$ for some absolute constant $C > 0$, with probability at least $1 - \delta$, Algorithm 1 outputs a policy $\widehat{\pi}_{\text{out}} \in \Delta(\Pi)$ such that $V_\star - V_{\theta^\star}(\widehat{\pi}_{\text{out}}) \leqslant \varepsilon$, as long as the number of episodes*

$$T = KH \geqslant \mathcal{O}\Big( dAU_A H^2 \log(\mathcal{N}_\Theta(1/T)/\delta)\iota \cdot \Lambda_{\mathsf{B}}^2/\varepsilon^2 \Big), \tag{8}$$

*where $\iota := \log(1 + KdU_A\Lambda_{\mathsf{B}}R_{\mathsf{B}})$, with $R_{\mathsf{B}} := \max_h\{1, \max_{\|v\|_1=1} \sum_{o,a} \|\mathbf{B}_h(o,a)v\|_1\}$.*

Theorem 9 shows that OMLE is sample-efficient for any B-stable PSRs—a broader class than in existing results for the same algorithm (Liu et al., 2022a; Zhan et al., 2022)—with much sharper sample complexities than existing work when instantiated to their settings. Importantly, we achieve the first polynomial sample complexity that scales with $\Lambda_{\mathsf{B}}^2$ dependence B-stability parameter (or regularity parameters alike[8]). Instantiating to $\alpha_{\mathsf{psr}}$-regular PSRs, using $\Lambda_{\mathsf{B}} \leqslant \sqrt{U_A}\alpha_{\mathsf{psr}}^{-1}$ (Example 8), our result implies a $\widetilde{\mathcal{O}}(dAU_A^2 \log \mathcal{N}_\Theta/(\alpha_{\mathsf{psr}}^2\varepsilon^2))$ sample complexity (ignoring $H$ and $\iota$[9]). This improves significantly over the $\widetilde{\mathcal{O}}(d^4A^4U_A^9 \log(\mathcal{N}_\Theta O)/(\alpha_{\mathsf{psr}}^6\varepsilon^2))$ result of Zhan et al. (2022).

---

[7]Named CRANE in (Zhan et al., 2022).

[8]Uehara et al. (2022b) achieves an $A^M\sigma_1^{-2}$ dependence for learning the optimal memory-$M$ policy in (their) $\sigma_1$-revealing POMDPs, which is however easier than learning the globally optimal policy considered here.

[9]The log-factor $\iota$ contains additional parameter $R_{\mathsf{B}}$ that is not always controlled by $\Lambda_{\mathsf{B}}$; this quantity also appears in Zhan et al. (2022); Liu et al. (2022b) but is controlled by their $\alpha_{\mathsf{psr}}^{-1}$ or $\gamma^{-1}$ respectively. Nevertheless, for all of our POMDP instantiations, $R_{\mathsf{B}}$ is polynomially bounded by other problem parameters so that $\iota$ is a mild log-factor. Further, our next algorithm EXPLORATIVE E2D avoids the dependence on $R_{\mathsf{B}}$ (Theorem 10).

**Overview of techniques** The proof of Theorem 9 (deferred to Appendix G) builds upon a sharp analysis for B-stable PSRs: 1) We use a more delicate choice of norm for bounding the errors (in the **B** operators) yielded from performance difference arguments; 2) We develop a generalized $\ell_2$-type Eluder argument that is sharper than the $\ell_1$-Eluder argument of Liu et al. (2022a); Zhan et al. (2022). A more detailed overview of techniques is presented in Appendix B.

## 4.2 EXPLORATIVE ESTIMATION-TO-DECISIONS (EXPLORATIVE E2D)

Estimation-To-Decisions (E2D) is a general model-based algorithm that is sample-efficient for any interactive decision making problem (including MDPs) with a bounded Decision-Estimation Coefficient (DEC), as established in the DEC framework by Foster et al. (2021). However, the E2D algorithm has not been instantiated on POMDPs/PSRs. We show that B-stable PSRs admit a sharp DEC bound, and thus can be learned sample-efficiently by a suitable E2D algorithm.

**EDEC & EXPLORATIVE E2D algorithm** We consider the *Explorative DEC (EDEC)* proposed in the recent work of Chen et al. (2022), which for a PSR class $\Theta$ is defined as

$$\overline{\mathrm{edec}}_\gamma(\Theta) = \sup_{\overline{\mu} \in \Delta(\Theta)} \inf_{\substack{p_{\exp} \in \Delta(\Pi) \\ p_{\mathrm{out}} \in \Delta(\Pi)}} \sup_{\theta \in \Theta} \Big\{ \mathbb{E}_{\pi \sim p_{\mathrm{out}}} \left[ V_\theta(\pi_\theta) - V_\theta(\pi) \right] - \gamma \mathbb{E}_{\pi \sim p_{\exp}} \mathbb{E}_{\overline{\theta} \sim \overline{\mu}} \left[ D_\mathrm{H}^2 \big( \mathbb{P}_\theta^\pi, \mathbb{P}_{\overline{\theta}}^\pi \big) \right] \Big\},$$

where $D_\mathrm{H}^2(\mathbb{P}_\theta^\pi, \mathbb{P}_{\overline{\theta}}^\pi) := \sum_{\tau_H} (\mathbb{P}_\theta^\pi(\tau_H)^{1/2} - \mathbb{P}_{\overline{\theta}}^\pi(\tau_H)^{1/2})^2$ denotes the squared Hellinger distance between $\mathbb{P}_\theta^\pi$ and $\mathbb{P}_{\overline{\theta}}^\pi$. Intuitively, the EDEC measures the optimal trade-off on model class $\Theta$ between gaining information by an "exploration policy" $\pi \sim p_{\exp}$ and achieving near-optimality by an "output policy" $\pi \sim p_{\mathrm{out}}$. Chen et al. (2022) further design the EXPLORATIVE E2D algorithm, a general model-based RL algorithm with sample complexity scaling with the EDEC.

We sketch the EXPLORATIVE E2D algorithm for a PSR class $\Theta$ as follows (full description in Algorithm 2): In each episode $t \in [T]$, we maintain a distribution $\mu^t \in \Delta(\Theta_0)$ over an *optimistic cover* $(\widetilde{\mathbb{P}}, \Theta_0)$ of $\Theta$ with radius $1/T$ (cf. Definition C.4), which we use to compute two policy distributions $(p_{\exp}^t, p_{\mathrm{out}}^t)$ by minimizing the following risk:

$$(p_{\mathrm{out}}^t, p_{\exp}^t) = \underset{(p_{\mathrm{out}}, p_{\exp}) \in \Delta(\Pi)^2}{\arg\min} \sup_{\theta \in \Theta} \mathbb{E}_{\pi \sim p_{\mathrm{out}}} [V_\theta(\pi_\theta) - V_\theta(\pi)] - \gamma \mathbb{E}_{\pi \sim p_{\exp}} \mathbb{E}_{\theta^t \sim \mu^t} \big[ D_\mathrm{H}^2(\mathbb{P}_\theta^\pi, \mathbb{P}_{\theta^t}^\pi) \big].$$

Then, we sample policy $\pi^t \sim p_{\exp}^t$, execute $\pi^t$ and collect trajectory $\tau^t$, and update the model distribution $\mu^t$ using a *Tempered Aggregation* scheme, which performs a Hedge update with initialization $\mu^1 = \mathrm{Unif}(\Theta_0)$, the log-likelihood loss with $\widetilde{\mathbb{P}}_\theta^{\pi^t}(\cdot)$ denoting the optimistic likelihood associated with model $\theta \in \Theta_0$ and policy $\pi^t$ (cf. Definition C.4), and learning rate $\eta \leqslant 1/2$:

$$\mu^{t+1}(\theta) \propto_\theta \mu^t(\theta) \cdot \exp\Big( \eta \log \widetilde{\mathbb{P}}_\theta^{\pi^t}(\tau^t) \Big).$$

After $T$ episodes, we output the average policy $\widehat{\pi}_{\mathrm{out}} := \frac{1}{T} \sum_{t=1}^T p_{\mathrm{out}}^t$.

**Theoretical guarantee** We provide a sharp bound on the EDEC for B-stable PSRs, which implies that EXPLORATIVE E2D can also learn them sample-efficient efficiently.

**Theorem 10** (Bound on EDEC & Guarantee of EXPLORATIVE E2D). *Suppose $\Theta$ is a PSR class with the same core test sets $\{\mathcal{U}_h\}_{h \in [H]}$, and each $\theta \in \Theta$ admits a B-representation that is $\Lambda_\mathrm{B}$-stable and has PSR rank at most $d$. Then we have*

$$\overline{\mathrm{edec}}_\gamma(\Theta) \leqslant \mathcal{O}(dAU_A\Lambda_\mathrm{B}^2 H^2 / \gamma).$$

*As a corollary, with probability at least $1 - \delta$, Algorithm 2 outputs a policy $\widehat{\pi}_{\mathrm{out}} \in \Delta(\Pi)$ such that $V_\star - V_{\theta^\star}(\widehat{\pi}_{\mathrm{out}}) \leqslant \varepsilon$, as long as the number of episodes*

$$T \geqslant \mathcal{O}\big( dAU_A\Lambda_\mathrm{B}^2 H^2 \log(\mathcal{N}_\Theta(1/T)/\delta)/\varepsilon^2 \big). \tag{9}$$

The sample complexity (9) matches OMLE (Theorem 9) and has a slight advantage in avoiding the log factor $\iota$ therein. In return, the $d$ in Theorem 10 needs to upper bound the PSR rank of *all* models in $\Theta$, whereas the $d$ in Theorem 9 only needs to upper bound the rank of the true model $\theta^\star$. We also remark that EXPLORATIVE E2D explicitly requires an optimistic covering of $\Theta$ as an input to the algorithm, which may be another disadvantage compared to OMLE (which uses optimistic covering

implicitly in the analyses only). The proof of Theorem 10 (in Appendix I.2) relies on mostly the same key steps as for analyzing the OMLE algorithm (overview in Appendix B).

**Extension: Reward-free learning & All-policy model estimation** Chen et al. (2022) also design the ALL-POLICY MODEL-ESTIMATION E2D algorithm for reward-free RL (Jin et al., 2020b) and (a harder related task) *all-policy model estimation*, with sample complexity scaling with the *All-policy Model-estimation DEC* (AMDEC) of the model class. We show that for B-stable PSRs, the AMDEC (43) can be upper bounded similar to the EDEC, and thus ALL-POLICY MODEL-ESTIMATION E2D (Algorithm 3) can be used to learn stable PSRs in a reward-free manner (Theorem H.4 & Appendix H.2).

### 4.3 MODEL-BASED OPTIMISTIC POSTERIOR SAMPLING (MOPS)

Finally, we show that MOPS—a general model-based algorithm originally proposed for MDPs by Agarwal & Zhang (2022)—can learn B-stable PSRs with the same sample complexity as OMLE and EXPLORATIVE E2D modulo minor differences (Theorem H.6 & Appendix H.3). The analysis is parallel to that of EXPLORATIVE E2D, building on insights from Chen et al. (2022).

## 5 EXAMPLES: SAMPLE COMPLEXITY OF LEARNING POMDPS

We illustrate the sample complexity of OMLE and EXPLORATIVE E2D given in Theorem 9 & 10 (with MOPS giving similar results) for learning an $\varepsilon$ near-optimal policy in the tractable POMDP subclasses presented in Section 3.2, and compare with existing results. (Obtaining the rates for OMLE also require bounds on the factor $R_B$, which can be found in Appendix D.)

**Weakly revealing tabular POMDPs** $m$-step $\alpha_{\mathsf{rev}}$-weakly revealing tabular POMDPs are B-stable PSRs with $\Lambda_B \leqslant \sqrt{S}\alpha_{\mathsf{rev}}^{-1}$, $d_{\mathsf{PSR}} \leqslant S$, and $U_A = A^{m-1}$ (Example 5). Therefore, both Theorem 9 & 10 achieve sample complexity $\widetilde{\mathcal{O}}\big(S^2 A^m H^2 \log \mathcal{N}_\Theta/(\alpha_{\mathsf{rev}}^2 \varepsilon^2)\big)$. This improves substantially over the current best result $\widetilde{\mathcal{O}}(S^4 A^{6m-4} H^6 \log \mathcal{N}_\Theta/(\alpha_{\mathsf{rev}}^4 \varepsilon^2))$ of Liu et al. (2022a, Theorem 24). For tabular POMDPs, we further have $\log \mathcal{N}_\Theta \leqslant \widetilde{\mathcal{O}}(H(S^2 A + SO))$.

**Low-rank future-sufficient POMDPs** $m$-step $\nu$-future-sufficient rank-$d_{\mathsf{trans}}$ POMDPs are B-stable PSRs with $\Lambda_B \leqslant \sqrt{U_A}\nu$, $d_{\mathsf{PSR}} \leqslant d_{\mathsf{trans}}$, and $U_A = A^{m-1}$ (Example 6). Therefore, Theorem 9 & 10 achieve sample complexity $\widetilde{\mathcal{O}}\big(d_{\mathsf{trans}} A^{2m-1} H^2 \log \mathcal{N}_\Theta \cdot \nu^2/\varepsilon^2\big)$. This improves substantially over the $\widetilde{\mathcal{O}}(d_{\mathsf{trans}}^2 A^{5m+3l+1} H^2 (\log \mathcal{N}_\Theta)^2 \cdot \nu^4 \gamma^2/\varepsilon^2)$ achieved by Wang et al. (2022), which requires an extra $l$-*step* $\gamma$-*past-sufficiency* assumption that we do not require.

**Decodable low-rank POMDPs** $m$-step decodable POMDPs with transition rank $d_{\mathsf{trans}}$ are B-stable PSRs with $\Lambda_B = 1$, $d_{\mathsf{PSR}} \leqslant d_{\mathsf{trans}}$, and $U_A = A^{m-1}$ (Example 7). Therefore, Theorem 9 & 10 achieve sample complexity $\widetilde{\mathcal{O}}\big(d_{\mathsf{trans}} A^m H^2 \log \mathcal{N}_\Theta/\varepsilon^2\big)$. Compared with the sample complexity upper bound $\widetilde{\mathcal{O}}(d_{\mathsf{trans}} A^m H^2 \log \mathcal{N}_{\mathcal{G}}/\varepsilon^2)$ of Efroni et al. (2022), the only difference is that their covering number $\mathcal{N}_{\mathcal{G}}$ is for the value class while $\mathcal{N}_\Theta$ is for the model class. However, this difference is nontrivial if the model class admits a much smaller covering number than the value class required for a concrete problem. For example, for tabular decodable POMDPs, using $d_{\mathsf{trans}} \leqslant S$ and $\log \mathcal{N}_\Theta \leqslant \widetilde{\mathcal{O}}(H(S^2 A + SO))$, we achieve the first $\widetilde{\mathcal{O}}(A^m \mathrm{poly}(H, S, O, A)/\varepsilon^2)$ sample complexity, which resolves the open question of Efroni et al. (2022).

Besides the above, our results can be further instantiated to latent MDPs (Kwon et al. (2021), as a special case of revealing POMDPs) and linear POMDPs (Cai et al., 2022) and improve over existing results, which we present in Appendix D.3.2 & D.3.4.

## 6 CONCLUSION

This paper proposes B-stability—a new structural condition for PSRs that encompasses most of the known tractable partially observable RL problems—and designs algorithms for learning B-stable PSRs with sharp sample complexities. We believe our work opens up many interesting questions, such as the computational efficiency of our algorithms, alternative (e.g. model-free) approaches for learning B-stable PSRs, or extensions to multi-agent settings.

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

## A  RELATED WORK

**Learning POMDPs**  Due to the non-Markovian nature of observations, policies in POMDPs in general depend on the full history of observations, and thus are much harder to learn than in fully observable MDPs. It is well-established that learning a near-optimal policy in POMDPs is indeed statistically hard in the worst-case, due to a sample complexity lower bound that is exponential in the horizon (Mossel & Roch, 2005; Krishnamurthy et al., 2016). Algorithms achieving such upper bounds are developed in (Kearns et al., 1999; Even-Dar et al., 2005). Poupart & Vlassis (2008); Ross

et al. (2007) develop Bayesian methods to learn POMDPs, while Azizzadenesheli et al. (2018) consider learning the optimal memoryless policies with policy gradient methods. Sample-efficient algorithms for learning POMDPs have also been developed in Hsu et al. (2012); Azizzadenesheli et al. (2016); Guo et al. (2016); Xiong et al. (2021); Jahromi et al. (2022); These works assume exploratory data or reachability assumptions, and thus do not address the challenge of exploration.

For learning POMDPs in the online (exploration) setting, sample-efficient algorithms have been proposed under various structural conditions, including reactiveness (Jiang et al., 2017), revealing conditions (Jin et al., 2020a; Liu et al., 2022a;c), revealing (future/past-sufficiency) and low rank (Cai et al., 2022; Wang et al., 2022), decodablity (Efroni et al., 2022), latent MDP (Kwon et al., 2021), learning short-memory policies (Uehara et al., 2022b), and deterministic transitions (Uehara et al., 2022a). Our B-stability condition encompasses most of these structural conditions, through which we provide a unified analysis with significantly sharper sample complexities (cf. Section 3 & 5). We further remark that for tabular revealing POMDPs, our sample complexities are minimax optimal in the accuracy $\varepsilon$ and the revealing constant, and have at most a small polynomial gap in $S, O, A$ factors from the minimax optimal rate, due to the lower bounds established in the work of Chen et al. (2023) (see e.g. their Table 1) after the initial appearance of this work.

For the computational aspect, planning in POMDPs is known to be PSPACE-compete (Papadimitriou & Tsitsiklis, 1987; Littman, 1994; Burago et al., 1996; Lusena et al., 2001). The recent work of Golowich et al. (2022b;a) establishes the belief contraction property in revealing POMDPs, which leads to algorithms with quasi-polynomial statistical and computational efficiency. Uehara et al. (2022a) design computationally efficient algorithms under the deterministic latent transition assumption. We remark that computational efficiency is beyond the scope of this paper, but is an important direction for future work.

Extensive-Form Games with Imperfect Information (EFGs; (Kuhn, 1953)) is an alternative formulation of partial observability in sequential decision-making. EFGs can be formulated as Partially Observable Markov Games (the multi-agent version of POMDPs (Liu et al., 2022c)) with a tree-structure. Learning from bandit feedback in EFGs has been recently studied in Farina et al. (2021); Kozuno et al. (2021); Bai et al. (2022a;b); Song et al. (2022), where the sample complexity scales polynomially in the size of the game tree (typically exponential in the horizon). This line of results is in general incomparable to ours as their tree structure assumption is different from B-stability.

**Learning PSRs** PSRs is proposed in Littman & Sutton (2001); Singh et al. (2012); Rosencrantz et al. (2004); Boots et al. (2013) as a general formulation of partially observable systems, following the idea of Observable Operator Models (Jaeger, 2000). POMDPs can be seen as a special case of PSRs (Littman & Sutton, 2001). Algorithms for learning PSRs have been designed assuming reachability or exploratory data, including spectral algorithms (Boots et al., 2011; Zhang et al., 2021; Jiang et al., 2018), supervised learning (Hefny et al., 2015), and others (Hamilton et al., 2014; Thon & Jaeger, 2015; Grinberg et al., 2018). Closely related to us, the very recent work of Zhan et al. (2022) develops the first sample-efficient algorithm for learning PSRs in the online setting assuming under a regularity condition. Our work provides three algorithms with sharper sample complexities for learning PSRs, under the more general condition of B-stability.

A concurrent work by Liu et al. (2022b) (released on the same day as this work) also identifies a general class of "well-conditioned" PSRs that can be learned sample-efficiently by the OMLE algorithm (Liu et al., 2022a). Our B-stability condition encompasses and is slightly more relaxed than their condition (consisting of two parts), whose part one is similar to the operator norm requirement in B-stability with a different choice of input norm, and which requires an additional second part.

Next, our sample complexity is much tighter than that of Liu et al. (2022b), on both general well-conditioned/B-stable PSRs and the specific examples encompassed (such as revealing POMDPs). For example, for the general class of "$\gamma$ well-conditioned PSRs" considered in their work, our results imply a $\tilde{\mathcal{O}}\left(dAU_A^2 H^2 \log \mathcal{N}_\Theta/(\gamma^2 \varepsilon^2)\right)$ sample complexity, whereas their result scales as $\tilde{\mathcal{O}}\left(d^2 A^5 U_A^3 H^4 \log \mathcal{N}_\Theta/(\gamma^4 \varepsilon^2)\right)$ (extracted from their proofs, cf. Appendix D.4). This originates from several differences between our techniques: First, Liu et al. (2022b)'s analysis of the OMLE algorithm is based on an $\ell_1$-type operator error bound for PSRs, combined with an $\ell_1$-Eluder argument, whereas our analysis is based on a new stronger $\ell_2$-type operator error bound for PSRs (Proposition F.2) combined with a new generalized $\ell_2$-Eluder argument (Proposition E.1), which together results in a sharper rate. Besides, our $\ell_2$-Eluder argument also admits an in-expectation de-

coupling form as a variant (Proposition E.6) that is necessary for bounding the EDEC (and hence the sample complexity of the EXPLORATIVE E2D algorithm) for B-stable PSRs; it is unclear whether their $\ell_1$-Eluder argument can give the same results. Another difference is that our performance decomposition and Eluder argument are done on a slightly difference choice of vectors from Liu et al. (2022b), which is the main reason for our better $1/\gamma$ dependency (or $\Lambda_B$ dependency for B-stable PSRs); See Appendix B for a detailed overview of our technique. Further, in terms of algorithms, Liu et al. (2022b) only study the OMLE algorithm, whereas we study both OMLE and two alternative algorithms Explorative E2D & MOPS in addition, which enjoy similar guarantees (with minor differences) as OMLE. In summary, Liu et al. (2022b) do not overlap with our contributions (2) and (3) highlighted in our abstract.

Finally, complementary to our work, Liu et al. (2022b) identify new concrete problems such as observable POMDPs with continuous observations, and develop new techniques to show that they fall into both of our general PSR frameworks, and thus tractable to sample-efficient learning. In particular, their result implies that this class is contained in (an extension of) the low-rank future-sufficient POMDPs defined in Definition D.11, if we suitably extend the formulation in Definition D.11 to the continuous observation setting by replacing vectors with $L_1$-integrable functions and matrices with linear operators.

**RL with function approximation** (Fully observable) RL with general function approximation has been extensively studied in a recent line of work (Jiang et al., 2017; Sun et al., 2019; Du et al., 2021; Jin et al., 2021; Foster et al., 2021; Agarwal & Zhang, 2022; Chen et al., 2022), where sample-efficient algorithms are constructed for problems admitting bounds in certain general complexity measures. While POMDPs/PSRs can be cast into their settings by treating the history $(\tau_{h-1}, o_h)$ as the state, prior to our work, it was highly unclear whether any sample-efficient learning results can be deduced from their results due to challenges in bounding the complexity measures (Liu et al., 2022a). Our work answers this positively by showing that the Decision-Estimation Coefficient (DEC; Foster et al. (2021)) for B-stable PSRs is bounded, using an explorative variant of the DEC defined by Chen et al. (2022), thereby showing that their EXPLORATIVE E2D algorithm and the closely related MOPS algorithm (Agarwal & Zhang, 2022) are both sample-efficient for B-stable PSRs. Our work further corroborates the connections between E2D, MOPS, and OMLE identified in (Chen et al., 2022) in the setting of partially observable RL.

## B OVERVIEW OF TECHNIQUES

The proof of Theorem 9 consists of three main steps: a careful performance decomposition into certain B-errors, bounding the squared B-errors by squared Hellinger distances, and a generalized $\ell_2$-Eluder argument. The proof of (the EDEC bound in) Theorem 10 follows similar steps except for replacing the final Eluder argument with a decoupling argument (Proposition E.6).

**Step 1: Performance decomposition** By the standard excess risk guarantee for MLE, our choice of $\beta = \mathcal{O}(\log(\mathcal{N}_\Theta(1/T)/\delta))$ guarantees with probability at least $1 - \delta$ that $\theta^\star \in \Theta^k$ for all $k \in [K]$ (Proposition G.2(a)). Thus, the greedy step (Line 4 in Algorithm 1) implies valid optimism: $V_\star \leqslant V_{\theta^k}(\pi^k)$. We then perform an error decomposition (Proposition F.1):

$$V_\star - V_{\theta^\star}(\pi^k) \leqslant V_{\theta^k}(\pi^k) - V_{\theta^\star}(\pi^k) \leqslant D_{\mathrm{TV}}\left(\mathbb{P}_{\theta^k}^{\pi^k}, \mathbb{P}_{\theta^\star}^{\pi^k}\right) \leqslant \sum_{h=0}^{H} \mathbb{E}_{\tau_{h-1} \sim \pi^k}\left[\mathcal{E}_{k,h}^\star(\tau_{h-1})\right], \quad (10)$$

where $\mathcal{E}_{k,0}^\star := \frac{1}{2}\left\|\mathcal{B}_{H:1}^k\left(\mathbf{q}_0^k - \mathbf{q}_0^\star\right)\right\|_\Pi$, and

$$\mathcal{E}_{k,h}^\star(\tau_{h-1}) := \max_\pi \frac{1}{2} \sum_{o_h, a_h} \pi(a_h|o_h)\left\|\mathcal{B}_{H:h+1}^k\left(\mathbf{B}_h^k(o_h, a_h) - \mathbf{B}_h^\star(o_h, a_h)\right)\mathbf{q}^\star(\tau_{h-1})\right\|_\Pi, \quad (11)$$

where for the ground truth PSR $\theta^\star$ and the OMLE estimates $\theta^k$ from Algorithm 1, we have defined respectively $\{\mathbf{B}_h^\star, \mathbf{q}_0^\star\}$ and $\{\mathbf{B}_h^k, \mathbf{q}_0^k\}$ as their B-representations, and $\{\mathcal{B}_{H:h}^\star\}$ and $\{\mathcal{B}_{H:h}^k\}$ as the corresponding $\mathcal{B}$-operators. (10) follows by expanding the $\mathbb{P}_{\theta^k}^{\pi^k}(\tau)$ and $\mathbb{P}_{\theta^\star}^{\pi^k}(\tau)$ (within the TV distance) using the B-representation and telescoping (Proposition F.1). This decomposition is similar as the ones in Liu et al. (2022a); Zhan et al. (2022), and more refined by keeping the $\mathcal{B}_{H:h+1}^k$ term in (11) (instead of bounding it right away), and using the $\Pi$-norm (3) instead of the $\ell_1$-norm as the error metric.

**Step 2: Bounding the squared B-errors** By again the standard fast-rate guarantee of MLE in squared Hellinger distance (Proposition G.2(b)), we have $\sum_{t=1}^{k-1}\sum_{h=0}^{H}D_{\mathrm{H}}^2(\mathbb{P}_{\theta^k}^{\pi_{h,\exp}^t},\mathbb{P}_{\theta^\star}^{\pi_{h,\exp}^t}) \leqslant 2\beta$ for all $k \in [K]$. Next, using the B-stability of the PSR, we have for any $1 \leqslant t < k \leqslant K$ that (Proposition F.2)

$$\sum_{h=0}^{H}\mathbb{E}_{\pi^t}\Big[\mathcal{E}_{k,h}^\star(\tau_{h-1})^2\Big] \leqslant 32\Lambda_{\mathsf{B}}^2 A U_A \sum_{h=0}^{H} D_{\mathrm{H}}^2\left(\mathbb{P}_{\theta^k}^{\pi_{h,\exp}^t},\mathbb{P}_{\theta^\star}^{\pi_{h,\exp}^t}\right). \tag{12}$$

Plugging the MLE guarantee into (12) and summing over $t \in [k-1]$ yields that for all $k \in [K]$,

$$\sum_{t=1}^{k-1}\sum_{h=0}^{H}\mathbb{E}_{\pi^t}\Big[\mathcal{E}_{k,h}^\star(\tau_{h-1})^2\Big] \leqslant \mathcal{O}\left(\Lambda_{\mathsf{B}}^2 A U_A \beta\right). \tag{13}$$

(13) is more refined than e.g. Liu et al. (2022a, Lemma 11), as (13) controls the second moment of $\mathcal{E}_{\theta,h}$, whereas their result only controls the first moment of a similar error.

**Step 3: Generalized $\ell_2$-Eluder argument** We now have (13) as a precondition and bounding (10) as our target. The only remaining difference is that (13) controls the error $\mathcal{E}_k^\star$ with respect to $\{\pi^t\}_{t \leqslant k-1}$, whereas (10) requires controlling the error $\mathcal{E}_k^\star$ with respect to $\pi^k$.

To this end, we perform a generalized $\ell_2$-Eluder dimension argument adapted to the structure of the function $\mathcal{E}_k^\star$'s (Proposition E.1), which implies that when $d_{\mathsf{PSR}} \leqslant d$,

$$\left(\sum_{t=1}^{k}\mathbb{E}_{\pi^t}\big[\mathcal{E}_{t,h}^\star(\tau_{h-1})\big]\right)^2 \lesssim d\iota \cdot \left(k + \sum_{t=1}^{k}\sum_{s=1}^{t-1}\mathbb{E}_{\pi^s}\big[\mathcal{E}_{t,h}^\star(\tau_{h-1})^2\big]\right), \ \forall (k,h) \in [K] \times [H]. \tag{14}$$

Note that such an $\ell_2$-type Eluder argument is allowed precisely as our precondition (13) is in $\ell_2$ whereas our target (10) only requires an $\ell_1$ bound. In comparison, Liu et al. (2022a); Zhan et al. (2022) only obtain a precondition in $\ell_1$, and thus has to perform an $\ell_1$-Eluder argument which results in an additional $d$ factor in the final sample complexity. Combining (10), (13) (summed over $k \in [K]$) and (14) completes the proof of Theorem 9.

## C   TECHNICAL TOOLS

### C.1   TECHNICAL TOOLS

**Lemma C.1** (Hellinger conditioning lemma (Chen et al., 2022, Lemma A.4))**.** *For any pair of random variable $(X, Y)$, it holds that*

$$\mathbb{E}_{X \sim \mathbb{P}_X}\big[D_{\mathrm{H}}^2\left(\mathbb{P}_{Y|X},\mathbb{Q}_{Y|X}\right)\big] \leqslant 2D_{\mathrm{H}}^2\left(\mathbb{P}_{X,Y},\mathbb{Q}_{X,Y}\right).$$

The following strong duality of (generalized) bilinear function is standard, e.g. it follows from the proof of Foster et al. (2021, Proposition 4.2).

**Theorem C.2** (Strong duality)**.** *Suppose that $\mathcal{X}, \mathcal{Y}$ are two topological spaces, such that $\mathcal{X}$ is discrete and $\mathcal{Y}$ is finite (with discrete topology). Then for a function $f : \mathcal{X} \times \mathcal{Y} \to \mathbb{R}$ that is uniformly bounded, it holds that*

$$\sup_{X \in \Delta_0(\mathcal{X})} \inf_{Y \in \Delta(\mathcal{Y})} \mathbb{E}_{x \sim X}\mathbb{E}_{y \sim Y}[f(x,y)] = \inf_{Y \in \Delta(\mathcal{Y})} \sup_{x \in \mathcal{X}} \mathbb{E}_{y \sim Y}[f(x,y)],$$

*where $\Delta_0(\mathcal{X})$ stands for space of the finitely supported distribution on $\mathcal{X}$.*

We will also use the following standard concentration inequality (see e.g. Foster et al. (2021, Lemma A.4)) when analyzing algorithm OMLE.

**Lemma C.3.** *For a sequence of real-valued random variables $(X_t)_{t \leqslant T}$ adapted to a filtration $(\mathcal{F}_t)_{t \leqslant T}$, the following holds with probability at least $1 - \delta$:*

$$\sum_{s=1}^{t} -\log \mathbb{E}\left[\exp(-X_s)|\mathcal{F}_{s-1}\right] \leqslant \sum_{s=1}^{t} X_s + \log\left(1/\delta\right), \qquad \forall t \in [T].$$

## C.2 COVERING NUMBER

In this section, we present the definition of the optimistic covering number $\mathcal{N}_\Theta$. Suppose that we have a model class $\Theta$, such that each $\theta \in \Theta$ parameterizes a sequential decision process. The *$\rho$-optimistic covering number* of $\Theta$ is defined as follows.

**Definition C.4** (Optimistic cover). *Suppose that there is a context space $\mathcal{X}$. An optimistic $\rho$-cover of $\Theta$ is a tuple $(\widetilde{\mathbb{P}}, \Theta_0)$, where $\Theta_0 \subset \Theta$ is a finite set, $\widetilde{\mathbb{P}} = \left\{ \widetilde{\mathbb{P}}_{\theta_0}^\pi(\cdot) \in \mathbb{R}_{\geq 0}^{\mathcal{T}^H} \right\}_{\theta_0 \in \Theta_0, \pi \in \Pi}$ specifies a optimistic likelihood function for each $\theta_0 \in \Theta_0$, such that:*

*(1) For $\theta \in \Theta$, there exists a $\theta_0 \in \Theta_0$ satisfying: for all $\tau \in \mathcal{T}^H$ and $\pi$, it holds that $\widetilde{\mathbb{P}}_{\theta_0}^\pi(\tau) \geq \mathbb{P}_\theta^\pi(\tau)$.*

*(2) For $\theta \in \Theta_0$, $\max_\pi \left\| \mathbb{P}_\theta^\pi(\tau_H = \cdot) - \widetilde{\mathbb{P}}_\theta^\pi(\tau_H = \cdot) \right\|_1 \leq \rho^2$.*

*The optimistic covering number $\mathcal{N}_\Theta(\rho)$ is defined as the minimal cardinality of $\Theta_0$ such that there exists $\widetilde{\mathbb{P}}$ such that $(\widetilde{\mathbb{P}}, \mathcal{M}_0)$ is an optimistic $\rho$-cover of $\Theta$.*

The above definition is taken from Chen et al. (2022); the covering argument in Liu et al. (2022a) essentially uses the above notion of covering number. Besides, the optimistic covering number can be upper bounded by the bracketing number adopted by Zhan et al. (2022).

By an explicit construction, Liu et al. (2022a) show that there is a universal constant $C$ such that for any model class $\Theta$ of tabular POMDPs, it holds that

$$\log \mathcal{N}_\Theta(\rho) \leq CH(S^2 A + SO) \log(CHSOA/\rho).$$

# D PROOFS FOR SECTION 3

## D.1 BASIC PROPERTY OF B-REPRESENTATION

**Proposition D.1** (Equivalence between PSR definition and B-representation). *A sequential decision process is a PSR with core test sets $(\mathcal{U}_h)_{h \in [H]}$ (in the sense of Definition 1) if and only if it admits a B-representation with respect to $(\mathcal{U}_h)_{h \in [H]}$ (in the sense of Definition 2).*

*Proof of Proposition D.1.* We first show that a PSR admits a B-representation. Suppose we have a PSR with core test sets $(\mathcal{U}_h)_{h \in [H]}$ satisfying Definition 1, with associated vectors $\{b_{t_h, h} \in \mathbb{R}^{\mathcal{U}_h}\}_{h \in [H], t_h \in \mathfrak{T}}$ given by (1). Then, define

$$\mathbf{B}_h(o, a) := \begin{bmatrix} | \\ b_{(o,a,t),h}^\top \\ | \end{bmatrix}_{t \in \mathcal{U}_{h+1}} \in \mathbb{R}^{\mathcal{U}_{h+1} \times \mathcal{U}_h}, \quad \mathbf{q}_0 := \begin{bmatrix} | \\ \mathbb{P}(t) \\ | \end{bmatrix}_{t \in \mathcal{U}_1} \in \mathbb{R}^{\mathcal{U}_1}.$$

We show that this gives a B-representation of the PSR. By (1), we have for all $(h, \tau_{h-1}, o, a)$ that

$$\mathbf{B}_h(o, a)\mathbf{q}(\tau_{h-1}) = [\mathbb{P}(o, a, t_{h+1}|\tau_{h-1})]_{t_{h+1} \in \mathcal{U}_{h+1}} = \mathbb{P}(o_h = o|\tau_{h-1}) \times \mathbf{q}(\tau_{h-1}, o, a).$$

Applying this formula recursively, we obtain

$$\mathbf{B}_{h:1}(\tau_h)\mathbf{q}_0 = \mathbb{P}(\tau_h) \times \mathbf{q}(\tau_h) = [\mathbb{P}(\tau_h, t_{h+1})]_{t_{h+1} \in \mathcal{U}_{h+1}},$$

which completes the verification of (2) in Definition 2.

We next show that a process admitting a B-representation is a PSR. Suppose we have a sequential decision process that admits a B-representation with respect to $(\mathcal{U}_h)_{h \in [H]}$ as in Definition 2. Fix $h \in [H]$. We first claim that, to construct vectors $(b_{t_h, h})_{t_h} \in \mathbb{R}^{\mathcal{U}_h}$ such that $\mathbb{P}(t_h|\tau_{h-1}) = \langle b_{t, h}, \mathbf{q}(\tau_{h-1}) \rangle$ for all test $t_h$ and history $\tau_{h-1}$ (Definition 1), we only need to construct such vectors for *full-length tests* $t_h = (o_{h:H+1}, a_{h:H})$. This is because, suppose we have assigned $b_{t_h, h} \in \mathbb{R}^{\mathcal{U}_h}$ for all full-length $t_h$'s. Then for any other $t_h = (o_{h:h+W-1}, a_{h:h+W-2})$ with $h + W - 1 < H + 1$ (non-full-length), take

$$b_{t_h, h} = \sum_{o_{h+W:H+1}} b_{t_h, (o_{h+W:H+1}, a'_{h+W-1:H}), h},$$

where $a'_{h+W-1:H} \in \mathcal{A}^{H-h-W+2}$ is an arbitrary and fixed action sequence. For this choice we have

$$\langle b_{t,h}, \mathbf{q}(\tau_{h-1}) \rangle = \sum_{o_{h+W:H+1}} \left\langle b_{t_h,(o_{h+W:H+1},a'_{h+W-1:H}),h}, \mathbf{q}(\tau_{h-1}) \right\rangle$$

$$= \sum_{o_{h+W:H+1}} \mathbb{P}(t_h, o_{h+W:H+1}, a'_{h+W-1:H}|\tau_{h-1}) = \mathbb{P}(t_h|\tau_{h-1})$$

as desired.

It remains to construct $b_{t_h,h}$ for all full-length tests. For any full-length test $t_h = (o_{h:H+1}, a_{h:H})$, take $b_{t_h,h} \in \mathbb{R}^{\mathcal{U}_h}$ with

$$b_{t_h,h}^\top = \mathbf{B}_H(o_H, a_H) \cdots \mathbf{B}_h(o_h, a_h) \in \mathbb{R}^{1 \times \mathcal{U}_h}.$$

By definition of the B-representation, for any history $\tau_h = (o_1, a_1, \cdots, o_h, a_h)$, and any test $t_{h+1} \in \mathcal{U}_{h+1}$, we have

$$\mathbb{P}(\tau_h) \times \mathbb{P}(t_{h+1}|\tau_h) = \mathbf{e}_{t_{h+1}}^\top \mathbf{B}_{h:1}(\tau_h) \times \mathbf{q}_0,$$

or in vector form,

$$\mathbb{P}(\tau_h) \times \mathbf{q}(\tau_h) = \mathbf{B}_{h:1}(\tau_h)\mathbf{q}_0, \tag{15}$$

where we recall $\mathbb{P}(\tau_h) = \mathbb{P}(o_1, \cdots, o_h|\text{do}(a_1, \cdots, a_h))$. Therefore, for the particular full history $\tau_H = (\tau_{h-1}, t_h)$, we have by applying (15) twice (for steps $H$ and $h-1$) that

$$\mathbb{P}(\tau_H) = \mathbf{B}_{H:1}(\tau_H)\mathbf{q}_0 = \mathbf{B}_{H:h}(o_{h:H}, a_{h:H})\mathbf{B}_{h-1:1}(\tau_{h-1})\mathbf{q}_0$$

$$= b_{t_h,h}^\top(\mathbb{P}(\tau_{h-1}) \times \mathbf{q}(\tau_{h-1})).$$

Dividing both sides by $\mathbb{P}(\tau_{h-1})$ (when it is nonzero), we get

$$\mathbb{P}(t_h|\tau_{h-1}) = \mathbb{P}(\tau_H|\tau_{h-1}) = \mathbb{P}(\tau_H)/\mathbb{P}(\tau_{h-1}) = b_{t_h,h}^\top \mathbf{q}(\tau_{h-1}). \tag{16}$$

This verifies (1) for all $\tau_{h-1}$ that are reachable. For $\tau_{h-1}$ that are not reachable, (16) also holds as both sides equal zero by our convention. This completes the verification of (1) in Definition 1. □

From the proof above, we can extract the following basic property of B-representation.

**Corollary D.2.** *Consider a PSR model with B-representation* $\{\{\mathbf{B}_h(o_h, a_h)\}_{h,o_h,a_h}, \mathbf{q}_0\}$. *For* $0 \leqslant h \leqslant H-1$, *it holds that*

$$\mathbb{P}(o_h|\tau_{h-1}) \times \mathbf{q}(\tau_{h-1}, o_h, a_h) = \mathbf{B}_h(o_h, a_h)\mathbf{q}(\tau_{h-1}).$$

*Furthermore, it holds that*

$$\mathbf{B}_{H:h}(\tau_{h:H})\mathbf{q}(\tau_{h-1}) = \mathbb{P}(\tau_{h:H}|\tau_{h-1}). \tag{17}$$

## D.2 WEAK B-STABILITY CONDITION

In this section, we define a weaker structural condition on PSRs, named the weak B-stability condition. In the remaining appendices, the proofs of our main sample complexity guarantees (Theorem 9, 10, H.4, H.6) will then assume the less-stringent weak B-stability condition of PSRs. Therefore, these main results will hold under both $\Lambda_B$-stablility (Definition 4) and weak $\Lambda_B$-stablility (Definition D.4) simultaneously.

To define weak B-stability, we first extend our definition of $\Pi$-norm to $\mathbb{R}^T$ for any set $T$ of tests. Recall that in (3), we have defined $\Pi$-norm on $\mathbb{R}^T$ with $T = (\mathcal{O} \times \mathcal{A})^{H-h}$ (and in (5), the $\Pi'$-norm for $T = \mathcal{U}_h$).

**Definition D.3** ($\Pi$-norm for general test set). *For $T \subset \mathfrak{T}$, we equip $\mathbb{R}^T$ with $\|\cdot\|_\Pi$ defined by*

$$\|v\|_\Pi := \max_{T' \subset T} \max_{\bar{\pi}} \sum_{t \in T'} \bar{\pi}(t) |v(t)|, \qquad v \in \mathbb{R}^T$$

*where $\max_{T' \subset T}$ is taken over all subsets $T'$ of $T$ such that $T'$ satisfies the prefix condition: there is no two $t \neq t' \in T'$ such that $t$ is a prefix of $t'$.*

It is straightforward to see that, for any $v \in \mathbb{R}^{\mathcal{U}_h}$, we have $\|v\|_1 \geqslant \|v\|_\Pi \geqslant \|v\|_{\Pi'}$.

**Definition D.4** (Weak B-stability)**.** *A PSR is weakly B-stable with parameter $\Lambda_\mathsf{B} \geqslant 1$ (henceforth weakly $\Lambda_\mathsf{B}$-stable) if it admits a B-representation and associated $\mathcal{B}$-operators $\{\mathcal{B}_{H:h}\}_{h \in [H]}$ such that, for any $h \in [H]$ and $\mathbf{p}, \mathbf{q} \in \mathbb{R}_{\geqslant 0}^{\mathcal{U}_h}$, we have[10]*

$$\|\mathcal{B}_{H:h}(\mathbf{p} - \mathbf{q})\|_\Pi \leqslant \Lambda_\mathsf{B} \sqrt{2(\|\mathbf{p}\|_\Pi + \|\mathbf{q}\|_\Pi)} \, \|\sqrt{\mathbf{p}} - \sqrt{\mathbf{q}}\|_2 \,, \tag{18}$$

Despite the seemingly different form, we can show that the weak B-stability condition is indeed weaker than the B-stability condition. Furthermore, the converse also holds: the B-stability can be implied by the weak B-stability condition, if we are willing to pay a $\sqrt{2U_A}$ factor. This is given by the proposition below.

**Proposition D.5.** *If a PSR is B-stable with parameter $\Lambda_\mathsf{B}$, then it is weakly B-stable with the same parameter $\Lambda_\mathsf{B}$. Conversely, if a PSR is weakly B-stable with parameter $\Lambda_\mathsf{B}$ (cf. Definition D.4), then it is B-stable with parameter $\sqrt{2U_A}\Lambda_\mathsf{B}$.*

*Proof of Proposition D.5.* We first show that B-stability implies weak B-stability. Fix a $h \in [H]$. We only need to show that, for $\mathbf{p}, \mathbf{q} \in \mathbb{R}_{\geqslant 0}^{\mathcal{U}_h}$, we have

$$\|\mathbf{p} - \mathbf{q}\|_* \leqslant \sqrt{2(\|\mathbf{p}\|_\Pi + \|\mathbf{q}\|_\Pi)} \, \|\sqrt{\mathbf{p}} - \sqrt{\mathbf{q}}\|_2 \,. \tag{19}$$

We show this inequality by showing the bound for the $(1, 2)$-norm and the $\Pi'$-norm separately. First, we have

$$\|\mathbf{p} - \mathbf{q}\|_{1,2}^2 = \sum_{\mathbf{a} \in \mathcal{U}_{A,h}} \left( \sum_{\mathbf{o}:(\mathbf{o},\mathbf{a}) \in \mathcal{U}_h} |\mathbf{p}(\mathbf{o},\mathbf{a}) - \mathbf{q}(\mathbf{o},\mathbf{a})| \right)^2$$

$$\leqslant \sum_{\mathbf{a} \in \mathcal{U}_{A,h}} \left( \sum_{\mathbf{o}:(\mathbf{o},\mathbf{a}) \in \mathcal{U}_h} \left| \sqrt{\mathbf{p}(\mathbf{o},\mathbf{a})} + \sqrt{\mathbf{q}(\mathbf{o},\mathbf{a})} \right|^2 \right) \left( \sum_{\mathbf{o}:(\mathbf{o},\mathbf{a}) \in \mathcal{U}_h} \left| \sqrt{\mathbf{p}(\mathbf{o},\mathbf{a})} - \sqrt{\mathbf{q}(\mathbf{o},\mathbf{a})} \right|^2 \right)$$

$$\leqslant 2 \sum_{\mathbf{a} \in \mathcal{U}_{A,h}} \left( \sum_{\mathbf{o}:(\mathbf{o},\mathbf{a}) \in \mathcal{U}_h} \mathbf{p}(\mathbf{o},\mathbf{a}) + \mathbf{q}(\mathbf{o},\mathbf{a}) \right) \left( \sum_{\mathbf{o}:(\mathbf{o},\mathbf{a}) \in \mathcal{U}_h} \left| \sqrt{\mathbf{p}(\mathbf{o},\mathbf{a})} - \sqrt{\mathbf{q}(\mathbf{o},\mathbf{a})} \right|^2 \right)$$

$$\leqslant 2(\|\mathbf{p}\|_\Pi + \|\mathbf{q}\|_\Pi) \sum_{\mathbf{a} \in \mathcal{U}_{A,h}} \sum_{\mathbf{o}:(\mathbf{o},\mathbf{a}) \in \mathcal{U}_h} \left| \sqrt{\mathbf{p}(\mathbf{o},\mathbf{a})} - \sqrt{\mathbf{q}(\mathbf{o},\mathbf{a})} \right|^2$$

$$= 2(\|\mathbf{p}\|_\Pi + \|\mathbf{q}\|_\Pi) \, \|\sqrt{\mathbf{p}} - \sqrt{\mathbf{q}}\|_2^2 \,,$$

where the first inequality is due to the Cauchy-Schwarz inequality; the second inequality is due to AM-GM inequality; the last inequality is because $\max_{\mathbf{a} \in \mathcal{U}_{A,h}} \sum_{\mathbf{o}:(\mathbf{o},\mathbf{a}) \in \mathcal{U}_h} v(\mathbf{o},\mathbf{a}) \leqslant \|v\|_\Pi$. Next, we have

$$\|\mathbf{p} - \mathbf{q}\|_{\Pi'}^2 = \max_\pi \left( \sum_{t \in \overline{\mathcal{U}}_h} \pi(t) \times |\mathbf{p}(t) - \mathbf{q}(t)| \right)^2$$

$$\leqslant 2 \max_\pi \left( \sum_{t \in \overline{\mathcal{U}}_h} \pi(t)(\mathbf{p}(t) + \mathbf{q}(t)) \right) \left( \sum_{t \in \overline{\mathcal{U}}_h} \pi(t) \left| \sqrt{\mathbf{p}(t)} - \sqrt{\mathbf{q}(t)} \right|^2 \right)$$

$$\leqslant 2 \max_\pi \left( \sum_{t \in \overline{\mathcal{U}}_h} \pi(t)(\mathbf{p}(t) + \mathbf{q}(t)) \right) \left( \sum_{t \in \mathcal{U}_h} \left| \sqrt{\mathbf{p}(t)} - \sqrt{\mathbf{q}(t)} \right|^2 \right)$$

$$\leqslant 2(\|\mathbf{p}\|_\Pi + \|\mathbf{q}\|_\Pi) \, \|\sqrt{\mathbf{p}} - \sqrt{\mathbf{q}}\|_2^2 \,.$$

Combining these two inequalities completes the proof of Eq. (19), which gives the first claim of Proposition D.5.

Next, we show that weak B-stability implies B-stability up to a $\sqrt{2U_A}$ factor. Fix a $h \in [H]$. For $x \in \mathbb{R}^{\mathcal{U}_h}$, we take $\mathbf{p} = [x]_+$, $\mathbf{q} = [x]_-$, then it suffices to show that

$$\sqrt{2(\|\mathbf{p}\|_\Pi + \|\mathbf{q}\|_\Pi)} \|\sqrt{\mathbf{p}} - \sqrt{\mathbf{q}}\|_2 \leqslant \sqrt{2U_A}\|x\|_* \,. \tag{20}$$

---

[10]Here we introduce the constant 2 in the square root in order for weak B-stability to be weaker than B-stability (Definition 4).

Indeed, we have

$$\left\|\sqrt{\mathbf{p}} - \sqrt{\mathbf{q}}\right\|_2 = \left\|\sqrt{|x|}\right\|_2 = \sqrt{\|x\|_1},$$

$$\|\mathbf{p}\|_\Pi + \|\mathbf{q}\|_\Pi \leqslant \left\|[x]_+\right\|_1 + \left\|[x]_-\right\|_1 = \|x\|_1.$$

This implies that

$$\sqrt{2(\|\mathbf{p}\|_\Pi + \|\mathbf{q}\|_\Pi)}\left\|\sqrt{\mathbf{p}} - \sqrt{\mathbf{q}}\right\|_2 \leqslant \sqrt{2}\|x\|_1. \tag{21}$$

Applying Lemma D.6 completes the proof of Eq. (20), and hence proves the second claim of Proposition D.5. □

**Lemma D.6.** *Consider the fused-norm as defined in Eq. (4). For any* $\mathbf{q} \in \mathbb{R}^{\mathcal{U}_h}$*, we have*

$$\|\mathbf{q}\|_* \leqslant \|\mathbf{q}\|_1 \leqslant |\mathcal{U}_{A,h}|^{1/2}\|\mathbf{q}\|_*.$$

*Proof of Lemma D.6.* By definition, we clearly have $\|\mathbf{q}\|_{1,2} \leqslant \|\mathbf{q}\|_1$ and $\|\mathbf{q}\|_{\Pi'} \leqslant \|\mathbf{q}\|_1$. On the other hand, by Cauchy-Schwarz inequality,

$$\|\mathbf{q}\|_1^2 = \left(\sum_{(\mathbf{o},\mathbf{a})\in\mathcal{U}_h} |\mathbf{q}(\mathbf{o},\mathbf{a})|\right)^2 \leqslant |\mathcal{U}_{A,h}| \sum_{\mathbf{a}\in\mathcal{U}_{A,h}} \left(\sum_{\mathbf{o}:(\mathbf{o},\mathbf{a})\in\mathcal{U}_h} |\mathbf{q}(\mathbf{o},\mathbf{a})|\right)^2 \leqslant |\mathcal{U}_{A,h}|\|\mathbf{q}\|_*^2.$$

Combining the inequalities above completes the proof. □

### D.3 Proofs for Section 3.2

#### D.3.1 Revealing POMDPs

$\ell_1$ **revealing condition** We first remark that, besides the revealing condition using the $\ell_2$ norms defined in Example 5, we also consider the $\ell_1$ **version of the revealing condition**, which measures the $\ell_1$-operator norm of *any left inverse* $\mathbb{M}_h^+$ of $\mathbb{M}_h$, instead of the $\ell_2$-operator norm of the pseudo-inverse $\mathbb{M}_h^\dagger$. Concretely, we say a POMDP satisfies the $m$-step $\alpha_{\mathsf{rev},\ell_1}$ $\ell_1$-revealing condition, if there exists a matrix $\mathbb{M}_h^+$ such that $\mathbb{M}_h^+\mathbb{M}_h = I$ and $\|\mathbb{M}_h^+\|_{1\to 1} \leqslant \alpha_{\mathsf{rev},\ell_1}^{-1}$. In Proposition D.7, we also show that any $m$-step $\alpha_{\mathsf{rev},\ell_1}$ $\ell_1$-revealing POMDP is a $\Lambda_B$-stable PSR with core test sets $\mathcal{U}_h = (\mathcal{O} \times \mathcal{A})^{\min\{m-1,H-h\}} \times \mathcal{O}$, and $\Lambda_B \leqslant \sqrt{A^{m-1}}\alpha_{\mathsf{rev},\ell_1}^{-1}$.

We consider Example 5, and show that any $m$-step revealing POMDP admit a B-representation that is B-stable. By definition, the initial predictive state is given by $\mathbf{q}_0 = \mathbb{M}_1\mu_1$. For $h \leqslant H - m$, we take

$$\mathbf{B}_h(o_h, a_h) = \mathbb{M}_{h+1}\mathbb{T}_{h,a_h}\operatorname{diag}(\mathbb{O}_h(o_h|\cdot))\mathbb{M}_h^+ \in \mathbb{R}^{\mathcal{U}_{h+1}\times\mathcal{U}_h}, \tag{22}$$

where $\mathbb{T}_{h,a} := \mathbb{T}_h(\cdot|\cdot, a) \in \mathbb{R}^{\mathcal{S}\times\mathcal{S}}$ is the transition matrix of action $a \in \mathcal{A}$, and $\mathbb{M}_h^+$ is any left inverse of $\mathbb{M}_h$. When $h > H - m$, we only need to take

$$\mathbf{B}_h(o_h, a_h) = [\mathbb{1}(t_h = (o_h, a_h, t_{h+1}))]_{(t_{h+1},t_h)\in\mathcal{U}_{h+1}\times\mathcal{U}_h} \in \mathbb{R}^{\mathcal{U}_{h+1}\times\mathcal{U}_h}, \tag{23}$$

where $\mathbb{1}(t_h = (o_h, a_h, t_{h+1}))$ is 1 if $t_h$ equals to $(o_h, a_h, t_{h+1})$, and 0 otherwise.

**Proposition D.7** (Weakly revealing POMDPs are B-stable). *For $m$-step revealing POMDP, (22) and (23) indeed give a B-representation, which is B-stable with $\Lambda_B \leqslant \max_h \left\|\mathbb{M}_h^+\right\|_{*\to 1}$, where*

$$\left\|\mathbb{M}_h^+\right\|_{*\to 1} := \max_{x\in\mathbb{R}^{U_h}:\|x\|_*\leqslant 1} \left\|\mathbb{M}_h^+ x\right\|_1.$$

*Therefore, any $m$-step $\alpha_{\mathsf{rev}}$-weakly revaling POMDP is B-stable with $\Lambda_B \leqslant \sqrt{S}\alpha_{\mathsf{rev}}^{-1}$ (by taking $+ = \dagger$, using $\|\cdot\|_2 \leqslant \|\cdot\|_*$, and $\|\cdot\|_1 \leqslant \sqrt{S}\|\cdot\|_2$). Similarly, any $m$-step $\alpha_{\mathsf{rev},\ell_1}$ $\ell_1$-revealing POMDP is B-stable with $\Lambda_B \leqslant \sqrt{A^{m-1}}\alpha_{\mathsf{rev},\ell_1}^{-1}$ (using $\|\cdot\|_1 \leqslant \sqrt{U_A}\|\cdot\|_*$ with $U_A = A^{m-1}$).*

For succinctness, we only provide the proof of a more general result (Proposition D.12). Besides, by a similar argument, we can also show that the parameter $R_B$ that appears in Theorem 9 can be bounded by $R_B \leqslant \alpha_{\mathsf{rev}}^{-1}A^m$ (for $\alpha_{\mathsf{rev}}$-weakly revealing POMDP) or $R_B \leqslant \alpha_{\mathsf{rev}}^{-1}A^m$ (for $\alpha_{\mathsf{rev},\ell_1}$ $\ell_1$-revealing POMDP, see e.g. Lemma D.13).

### D.3.2 LATENT MDPS

In this section, we follow Kwon et al. (2021) to show that latent MDPs as a sub-class of POMDPs, and then obtain the sample complexity for learning latent MDPs of our algorithms.

**Example D.8** (Latent MDP). A latent MDP $M$ is specified by a tuple $\{\mathcal{S}, \mathcal{A}, (M_m)_{m=1}^N, H, \nu\}$, where $M_1, \cdots, M_N$ are $N$ MDPs with joint state space $\mathcal{S}$, joint action space $\mathcal{A}$, horizon $H$, and $\nu \in \Delta([N])$ is the mixing distribution over $M_1, \cdots, M_N$. For $m \in [N]$, the transition dynamic of $M_m$ is specified by $(\mathbb{T}_{h,m} : \mathcal{S} \times \mathcal{A} \to \Delta(\mathcal{S}))_{h=1}^H$ along with the initial state distribution $\mu_m$, and at step $h$ the binary random reward $r_{h,m}$ is generated according to probability $R_{h,m} : \mathcal{S} \times \mathcal{A} \to [0,1]$.

Clearly, $M$ can be casted into a POMDP $M'$ with state space $\overline{\mathcal{S}} = [N] \times \mathcal{S} \times \{0,1\}$ and observation space $\mathcal{O} = \mathcal{S} \times \{0,1\}$ by considering the latent state being $\bar{s}_h = (s_h, r_h, m) \in \overline{\mathcal{S}}$ and observation being $o_h = (s_h, r_{h-1}) \in \mathcal{O}$. More specifically, at the start of each episode, the environment generates a $m \sim \nu$ and a state $s \sim \mu_m$, then the initial latent state is $\bar{s}_1 = (m, s, 0)$ and $o_1 = (s, 0)$; at each step $h$, the agent takes $a_h$ after receiving $o_h$, then the environment generates $r_h \in \{0, 1\}$, $\bar{s}_{h+1}$ and $o_{h+1}$ according to $(\bar{s}_h, a_h)$: $r_h = 1$ with probability $R_{h,m}(s_h, a_h)$[11], $s_{h+1} \sim \mathbb{T}_{h,m}(\cdot|s_h, a_h)$, $\bar{s}_{h+1} = (m, s_{h+1}, r_h)$ and $o_{h+1} = (s_{h+1}, r_h)$. [12]

In a latent MDP, we denote $T_h$ to be the set of all possible sequences of the form $(a_h, r_h, s_{h+1}, \cdots, a_{h+l-1}, r_{h+l-1}, s_{h+l})$ (called a *test* in (Kwon et al., 2021)). For $h \leqslant H - l + 1$, $t = (a_h, r_h, s_{h+1}, \cdots, a_{h+l-1}, r_{h+l-1}, s_{h+l}) \in T_h$ and $s \in \mathcal{S}$, we can define

$$\mathbb{P}_{h,m}(t|s) = \mathbb{P}_m(r_h, s_{h+1}, \cdots, r_{h+l-1}, s_{h+l}|s_h = s, \text{do}(a_h, \cdots, a_{h+l-1})),$$

where $\mathbb{P}_m$ stands for the probability distribution under MDP $M_m$.

**Definition D.9** (Sufficient tests for latent MDP). *A latent MDP $M$ is said to be $l$-step test-sufficient, if for $h \in [H - l + 1]$ and $s \in \mathcal{S}$, the matrix $L_h(s)$ given by*

$$L_h(s) := [\mathbb{P}_{h,m}(t|s)]_{(t,m) \in T_h \times [N]} \in \mathbb{R}^{T_h \times N}$$

*has rank $N$. $M$ is $l$-step $\sigma$-test-sufficient if $\sigma_N(L_h(s)) \geqslant \sigma$ for all $h \in [H - l + 1]$ and $s \in \mathcal{S}$.*

Under test sufficiency, the latent MDP is an $(l + 1)$-step $\sigma$-weakly revealing POMDP, as shown in (Zhan et al., 2022, Lemma 12). Hence, as a corollary of Proposition D.7, using the fact that $|\overline{\mathcal{S}}| = 2SN$, we have the following result.

**Proposition D.10** (Latent MDPs are B-stable). *For an $l$-step $\sigma$-test-sufficient latent MDP $M$, its equivalent POMDP $M'$ is $(l+1)$-step $\sigma$-weakly revealing, and thus B-stable with $\Lambda_{\mathsf{B}} \leqslant \sqrt{2SN}\sigma^{-1}$.*

Therefore, by a similar reasoning to $m$-step revealing POMDPs in Section 5 (and Appendix D.3.1), our algorithms OMLE/EXPLORATIVE E2D/MOPS can achieve a sample complexity of

$$\widetilde{\mathcal{O}}\left(\frac{S^2 N^2 A^{l+1} H^2 \log \mathcal{N}_\Theta}{\sigma^2 \varepsilon^2}\right)$$

for learning $\varepsilon$-optimal policy, where $\Theta$ is the class of all such latent MDPs. Further, the optimistic covering number of $\Theta$ can be bounded as (similar as (Liu et al., 2022a, Appendix B) and Appendix D.3.4)

$$\log \mathcal{N}_\Theta(\rho) \leqslant \widetilde{\mathcal{O}}\left(NS^2 AH\right).$$

Thus, we achieve a $\widetilde{\mathcal{O}}\left(S^4 N^3 A^{l+2} H^3 \sigma^{-2} \varepsilon^{-2}\right)$ sample complexity. This improves over the result of Kwon et al. (2021) who requires extra assumptions including reachability, a gap between the $N$ MDP transitions, and full rank condition of histories (Kwon et al., 2021, Condition 2.2). Besides, our result does not require extra assumptions on core histories—which is needed for deriving sample complexities from the $\alpha_{\mathsf{psr}}$-regularity of (Zhan et al., 2022)—which could be rather unnatural for latent MDPs.

We remark that the argument above can be generalized to low-rank latent MDPs[13] straightforwardly, achieving a sample complexity of $\widetilde{\mathcal{O}}\left(d_{\mathsf{trans}}^2 N^2 A^{l+2} H^2 \log \mathcal{N}_\Theta / \sigma^2 \varepsilon^2\right)$. For more details, see Appendix D.3.3.

---

[11]Note that under such formulation, $M'$ has deterministic rewards.

[12]The terminal state $s_{H+1}$ is a dummy state.

[13]A latent MDP $M$ has transition rank $d$ if each $M_m$ has rank $d$ as a linear MDP (Jin et al., 2020c).

*Proof of Proposition D.10.* As is pointed out by Zhan et al. (2022), the $(l+1)$-step emission matrix of $M'$ has a relatively simple form: notice that for $h \in [H - l + 1]$, $\bar{s} = (m, s, r) \in \overline{\mathcal{S}}$ and $t_h = (o_h, a_h, \cdots, o_{h+l}) \in \mathcal{U}_h$ (with $o_{h+1} = (s_{h+1}, r_h), \cdots, o_{h+l} = (s_{h+l}, r_{h+l-1})$), we have

$$\mathbb{M}_h(t, s) = \mathbb{1}(o_h = (s_h, r_{h-1}))\mathbb{P}_m(r_h, s_{h+1}, \cdots, r_{h+l-1}, s_{h+l} | s_h = s, \mathrm{do}(a_h, \cdots, a_{h+l-1})),$$

where $\mathbb{1}(o_h = (s_h, r_{h-1}))$ is 1 when $o_h = (s_h, r_{h-1})$ and 0 otherwise. Therefore, up to some permutation, $\mathbb{M}_h$ has the form

$$\mathbb{M}_h = \begin{bmatrix} L_h(s^{(1)}) & & & & & \\ & L_h(s^{(1)}) & & & & \\ & & L_h(s^{(2)}) & & & \\ & & & L_h(s^{(2)}) & & \\ & & & & \ddots & \\ & & & & & L_h(s^{(|\mathcal{S}|)}) \\ & & & & & & L_h(s^{(|\mathcal{S}|)}) \end{bmatrix},$$

where $\{s^{(1)}, s^{(2)}, \cdots, s^{(|\mathcal{S}|)}\}$ is an ordering of $\mathcal{S}$. Therefore, it follows from definition that $\|\mathbb{M}_h^\dagger\|_{2\to 2} \leq \max_{h,s} \|L_h(s)^\dagger\|_{2\to 2} \leq \sigma^{-1}$. Applying Proposition D.7 completes the proof. $\square$

### D.3.3 Low-rank POMDPs with future sufficiency

In this section, we provide a detailed discussion of low-rank POMDPs and $m$-step future sufficiency condition mentioned in Example 6. We present a slightly generalized version of the $m$-step future sufficiency condition defined in (Wang et al., 2022); see also (Cai et al., 2022).

For low-rank POMDPs, we now state a slightly more relaxed version of the future-sufficiency condition defined in (Wang et al., 2022). Recall the $m$-step emission-action matrices $\mathbb{M}_h \in \mathbb{R}^{\mathcal{U}_h \times \mathcal{S}}$ defined in (7).

**Definition D.11** ($m$-step $\nu$-future-sufficient POMDP). *We say a low-rank POMDP is $m$-step $\nu$-future-sufficient if for $h \in [H]$, $\min_{\mathbb{M}_h^\natural} \|\mathbb{M}_h^\natural\|_{1\to 1} \leq \nu$, where $\min_{\mathbb{M}_h^\natural}$ is taken over all possible $\mathbb{M}_h^\natural$'s such that $\mathbb{M}_h^\natural \mathbb{M}_h \mathbb{T}_{h-1} = \mathbb{T}_{h-1}$.*

Wang et al. (2022) consider a factorization of the latent transition: $\mathbb{T}_h = \Psi_h \Phi_h$ with $\Psi_h \in \mathbb{R}^{\mathcal{S} \times d_{\text{trans}}}, \Phi_h \in \mathbb{R}^{d_{\text{trans}} \times (\mathcal{S} \times \mathcal{A})}$ for $h \in [H]$, and assumes that $\|\mathbb{M}_h^\natural\|_{1\to 1} \leq \nu$ with the specific choice $\mathbb{M}_h^\natural = \Psi_{h-1}(\mathbb{M}_h \Psi_{h-1})^\dagger$ (note that it is taking an exact pseudo-inverse instead of any general left inverse). It is straightforward to check that this choice indeed satisfies $\mathbb{M}_h^\natural \mathbb{M}_h \mathbb{T}_h = \mathbb{T}_h$, using which Definition D.11 recovers the definition of Wang et al. (2022). It also encompasses the setting of Cai et al. (2022) ($m = 1$).

We show that the following (along with (23)) gives a B-representation for the POMDP:[14]

$$\mathbf{B}_h(o, a) = \mathbb{M}_{h+1} \mathbb{T}_{h,a} \, \mathrm{diag}\left(\mathbb{O}_h(o|\cdot)\right)\mathbb{M}_h^\natural, \qquad h \in [H - m]. \tag{24}$$

This generalizes the choice of B-representation in (22) for (tabular) revealing POMDPs, as the matrix $\mathbb{M}_h^\natural$ can be thought of as a "generalized pseudo-inverse" of $\mathbb{M}_h$ that is aware of the subspace spanned by $\mathbb{T}_{h-1}$. This choice is more suitable when $\mathcal{S}$ or $\mathcal{O}$ are extremely large, in which case the vanilla pseudo-inverse $\mathbb{M}_h^\dagger$ may not be bounded in $\|\cdot\|_{1\to 1}$ norm. In the tabular case, setting $\natural = \dagger$ in (24) recovers (22).

**Proposition D.12** (Future-sufficient low-rank POMDPs are B-stable). *The operators $(\mathbf{B}_h(o, a))_{h,o,a}$ given by (24) (with the case $h > H - m$ given by (23)) is indeed a B-representation, and it is B-stable with $\Lambda_B \leq \sqrt{A^{m-1}} \max_h \|\mathbb{M}_h^\natural\|_1$. As a corollary, any $m$-step $\nu$-future-sufficient low-rank POMDP admits a B-representation with $\Lambda_B \leq \sqrt{A^{m-1}}\nu$ (and also $R_B \leq A^m \nu$).*

Combining Proposition D.12 and Algorithm 2 gives the sample complexity guarantee of Algorithm 2 for future sufficient POMDP. For Algorithm OMLE, combining $R_B \leq A^m \nu$ with Theorem 9 establishes the sample complexity of OMLE, as claimed in Section 5.

---

[14]For simplicity, we write $\mathbb{T}_{h,a} := \mathbb{T}_h(\cdot|\cdot, a) \in \mathbb{R}^{\mathcal{S} \times \mathcal{S}}$ the transition matrix of action $a \in \mathcal{A}$.

*Proof of Proposition D.12.* First, we verify (2) for $0 \leqslant h \leqslant H - m$. In this case, for $t_{h+1} \in \mathcal{U}_{h+1}$, we have[15]

$$
\begin{aligned}
\mathbf{e}_{t_{h+1}}^\top \mathbf{B}_{h:1}(\tau_h)\mathbf{q}_0 &= \mathbf{e}_{t_{h+1}}^\top \prod_{l=1}^h \left[ \mathbb{M}_{l+1} \mathbb{T}_{l,a_l} \operatorname{diag}\left(\mathbb{O}_l(o_l|\cdot)\right) \mathbb{M}_l^\natural \right] \mathbb{M}_1 \mu_1 \\
&\overset{(i)}{=} \mathbf{e}_{t_{h+1}}^\top \mathbb{M}_{h+1} \mathbb{T}_{h,a_h} \operatorname{diag}\left(\mathbb{O}_h(o_h|\cdot)\right) \cdots \mathbb{T}_{1,a_1} \operatorname{diag}\left(\mathbb{O}_1(o_1|\cdot)\right) \mu_1 \\
&\overset{(ii)}{=} \sum_{s_1, s_2, \cdots, s_{h+1}} \mathbb{P}(t_{h+1}|s_{h+1}) \mathbb{T}_{h,a_h}(s_{h+1}|s_h) \mathbb{O}_h(o_h|s_h) \cdots \mathbb{T}_1(s_2|s_1) \mathbb{O}_1(o_1|s_1) \mu_1(s_1) \\
&= \mathbb{P}(\tau_h, t_{h+1}),
\end{aligned}
\tag{25}
$$

where (i) is due to $\mathbb{M}_l^\natural \mathbb{M}_l \mathbb{T}_l = \mathbb{T}_l$ for $1 \leqslant l \leqslant h$, in (ii) we use the definition (7) to deduce that the $(t_{h+1}, s_{h+1})$-entry of $\mathbb{M}_{h+1}$ is $\mathbb{P}(t_{h+1}|s_{h+1})$.

Finally, we verify (2) for $H - m < h < H$. In this case, $\mathcal{U}_{h+1} = \mathcal{O}^{H-h} \times \mathcal{A}^{H-h-1}$, and hence for $\tau_h = (o_1, a_1, \cdots, o_h, a_h)$, $t_{h+1} = (o_{h+1}, a_{h+1}, \cdots, o_H) \in \mathcal{U}_{h+1}$, we consider $t_{H-m+1} = (o_{H-m+1}, a_{H-m+1}, \cdot, o_H)$:

$$
\mathbf{e}_{t_{h+1}}^\top \mathbf{B}_{h:1}(\tau_h)\mathbf{q}_0 = \mathbf{e}_{t_{H-m+1}}^\top \mathbf{B}_{H-m:1}(\tau_{H-m})\mathbf{q}_0 = \mathbb{P}(t_{H-m+1}, \tau_{H-m}) = \mathbb{P}(t_{h+1}, \tau_h).
$$

It remains to verify that the B-representation is $\Lambda_B$-stable with $\Lambda_B \leqslant \sqrt{A^{m-1}}\nu$ and $R_B \leqslant A^m \nu$, we invoke the following lemma.

**Lemma D.13.** *For $1 \leqslant h \leqslant H$, $x \in \mathbb{R}^{|\mathcal{U}_h|}$, it holds that*

$$
\|\mathcal{B}_{H:h}x\|_\Pi = \max_\pi \sum_{\tau_{h:H}} \|\mathbf{B}_H(o_H, a_H) \cdots \mathbf{B}_h(o_h, a_h)x\|_1 \times \pi(\tau_{h:H}) \leqslant \max\left\{ \left\|\mathbb{M}_h^\natural x\right\|_1, \|x\|_\Pi \right\}.
$$

*Similarly, we have* $\sum_{o,a} \|\mathbf{B}_h(o,a)v\|_1 \leqslant \max\left\{ A^m \left\|\mathbb{M}_h^\natural v\right\|_1, A\|v\|_1 \right\}$.

By Lemma D.13, it holds that

$$
\|\mathcal{B}_{H:h}x\|_\Pi \leqslant \max\{\nu\|x\|_1, \|x\|_\Pi\} \leqslant \nu\|x\|_1 \leqslant \nu\sqrt{U_A}\|x\|_* = \nu\sqrt{A^{m-1}}\|x\|_*,
$$

where the second inequality is because $\|x\|_\Pi \leqslant \|x\|_1$ and $\nu \geqslant 1$, and the third inequality is due to Lemma D.6 and $\|x\|_* \geqslant \|x\|_{\Pi'}$ by definition. Similarly, we have $R_B \leqslant A^m \nu$. This concludes the proof of Proposition D.12. $\qquad\square$

*Proof of Lemma D.13.* We first consider the case $h > H - m$. Then for each $h \leqslant l \leqslant H$, $\mathbf{B}_l$ is given by (23), and hence for trajectory $\tau_{h:H} = (o_h, a_h, \cdots, o_H, a_H)$ and $x \in \mathbb{R}^{\mathcal{U}_h}$, it holds that

$$
\mathbf{B}_{h:H}(\tau_{h:H})x = x(o_h, a_h, \cdots, o_H).
$$

This implies that $\|\mathcal{B}_{H:h}x\|_\Pi = \|x\|_\Pi$ and $\sum_{o,a} \|\mathbf{B}_h(o,a)x\|_1 = A\|x\|_1$ directly.

We next consider the case $h \leqslant H - m$. Note that for $\tau_{h:H} = (o_h, a_h, \cdots, o_{H-m}, a_{H-m}, \cdots, o_H)$, we can denote $t_{H-m+1} = (o_{H-m+1}, a_{H-m+1}, \cdots, o_H)$, then similar to (25) we have

$$
\begin{aligned}
\mathbf{B}_{H:h}(\tau_{h:H}) &= \mathbf{e}_{t_{H-m+1}}^\top \mathbb{M}_{H-m+1} \left[ \prod_{l=h}^{H-m} \mathbb{T}_{l,a_l} \operatorname{diag}\left(\mathbb{O}_l(o_l|\cdot)\right) \right] \mathbb{M}_h^\natural \\
&= \sum_{s_h, \cdots, s_{H-m+1}} \mathbb{P}(t_{H-m+1}|s_{H-m+1}) \left[ \prod_{l=h}^{H-m} \mathbb{T}_{l,a_l}(s_{l+1}|s_l) \mathbb{O}_l(o_l|s_l) \right] \mathbf{e}_{s_h}^\top \mathbb{M}_h^\natural \\
&= \sum_{s \in \mathcal{S}} \mathbb{P}(\tau_{h:H}|s_h = s) \mathbf{e}_s^\top \mathbb{M}_h^\natural.
\end{aligned}
$$

Therefore, for policy $\pi$ and trajectory $\tau_{h:H}$, it holds that

$$
\pi(\tau_{h:H}) \times \mathbf{B}_{H:h}(\tau_{h:H})x = \sum_{s \in \mathcal{S}} \mathbb{P}^\pi(\tau_{h:H}|s_h = s) \times \mathbf{e}_s^\top \mathbb{M}_h^\natural x,
$$

---

[15]For the clarity of presentation, in this section we adopt the following notation: for operator $(\mathcal{L}_n)_{n \in \mathbb{N}}$, we write $\prod_{h=n}^m \mathcal{L}_h = \mathcal{L}_m \circ \cdots \circ \mathcal{L}_n$.

and this gives $\|\mathcal{B}_{H:h}x\|_{\Pi} \leqslant \left\|\mathbb{M}_h^{\natural}x\right\|_1$ directly.

Besides, we similarly have

$$\sum_{o,a} \|\mathbf{B}_h(o,a)v\|_1 = \sum_{o,a} \left\|\mathbb{M}_{h+1}\mathbb{T}_{h,a}\,\mathrm{diag}\left(\mathbb{O}_h(o|\cdot)\right)\mathbb{M}_h^{\natural}x\right\|_1 \leqslant A\left|\mathcal{U}_{A,h+1}\right|\left\|\mathbb{M}_h^{\natural}x\right\|_1.$$

The proof is completed by combining the two cases above. $\qquad\square$

### D.3.4 Linear POMDPs

Linear POMDPs (Zhan et al., 2022) is a subclass of low-rank POMDPs where the latent transition and emission dynamics are linear in certain known feature maps. In the following, we present a slightly more general version of the linear POMDP definition in Zhan et al. (2022, Definition 5).

**Definition D.14** (Linear POMDP). *A POMDP is linear with respect to the given set $\Psi$ of feature maps $(\psi_h : \mathcal{S} \to \mathbb{R}^{d_{s,1}}, \psi_h : \mathcal{S} \times \mathcal{A} \to \mathbb{R}^{d_{s,2}}, \varphi_h : \mathcal{O} \times \mathcal{S} \to R^{d_o})_h$ if there exists $A_h \in \mathbb{R}^{d_{s,1} \times d_{s,2}}, u_h \in \mathbb{R}^d, v \in \mathbb{R}^{d_{s,1}}$ such that*

$$\mathbb{T}_h(s'|s,a) = \phi_h(s')^{\top}A_h\psi_h(s,a), \qquad \mu_1(s) = \langle v, \phi_0(s)\rangle, \qquad \mathbb{O}_h(o|s) = \langle u_h, \varphi_h(o|s)\rangle.$$

*We further assume a standard normalization condition: For $R := \max\{d_{s,1}, d_{s,2}, d_o\}$,*

$$\sum_{s'}\left\|\phi_h(s')\right\|_1 \leqslant R, \qquad \|\psi_h(s,a)\|_1 \leqslant R, \qquad \sum_{o}\|\varphi_h(o|s)\|_1 \leqslant R,$$

$$\|A_h\|_{\infty,\infty} \leqslant R, \qquad \|v\|_{\infty} \leqslant R, \qquad \|u_h\|_{\infty} \leqslant R.$$

**Proposition D.15.** *Suppose that $\Theta$ is the set of models that are linear with respect to a given $\Psi$ and have parameters bounded by $R$. Then $\log\mathcal{N}_{\Theta}(\rho) = \mathcal{O}\left((d_s + d_o)H\log(d_s d_o H/\rho)\right)$, where we denote $d_s := d_{s,1}d_{s,2}$.*

It is direct to check that any linear POMDP is a low-rank POMDP (cf. Example 6) with $d_{\mathsf{PSR}} \leqslant d_{\mathsf{trans}} \leqslant \min\{d_{s,1}, d_{s,2}\}$. Therefore, by a similar reasoning to Appendix D.3.3, Theorem 9 & 10 both achieve a sample complexity of $\widetilde{\mathcal{O}}\left(\min\{d_{s,1}, d_{s,2}\}(d_{s,1}d_{s,2} + d_o)AU_A H^3\Lambda_{\mathsf{B}}^2\varepsilon^{-2}\right)$ for learning an $\varepsilon$-optimal policy in $\Lambda_{\mathsf{B}}$-stable linear POMDPs (which include e.g. revealing and decodable linear POMDPs).

This result significantly improves over the result extracted from (Zhan et al., 2022, Corollary 6.5): Assuming their $\alpha_{\mathsf{psr}}$-regularity, we have $\Lambda_{\mathsf{B}} \leqslant \sqrt{U_A}\alpha_{\mathsf{psr}}^{-1}$ (Example 8) and thus obtain a sample complexity of

$$\widetilde{\mathcal{O}}\left(\min\{d_{s,1}, d_{s,2}\}(d_{s,1}d_{s,2} + d_o)AU_A^2 H^3/(\alpha_{\mathsf{psr}}^2\varepsilon^2)\right).$$

This only scales with $d^3 AU_A^2$ (where $d \geqslant \max\{d_{s,1}, d_{s,2}, d_o\}$), whereas their results involve much larger polynomial factors of all three parameters. Further, apart from the dimension-dependence, their covering number scales with an additional $\log O$ (and thus their result does not handle extremely large observation spaces).

*Proof of Proposition D.15.* In the following, we generalize the construction of optimistic covering of $\Theta$ using the optimistic covering of $\left\{\mathbb{O}_h^{\theta}\right\}_{\theta\in\Theta}$ and $\left\{\mathbb{T}_h^{\theta}\right\}_{\theta\in\Theta}$ as in Liu et al. (2022a, Appendix B).

**Lemma D.16** (Bounding optimistic covering number for POMDPs). *For $\Theta$ a class of POMDPs, let us denote $\Theta_{h;o} = \left\{\mathbb{O}_h^{\theta}\right\}_{\theta\in\Theta}$ and $\Theta_{h;o} = \left\{\mathbb{T}_h^{\theta}\right\}_{\theta\in\Theta}$[16]. Then it holds that for $\rho \in (0,1]$,[17]*

$$\log\mathcal{N}_{\Theta}(\rho_1) \leqslant 2H\max_h\left\{\log\mathcal{N}_{\Theta_{h;o}}(\rho_1/3H), \log\mathcal{N}_{\Theta_{h;t}}(\rho_1/3H)\right\}.$$

By Lemma D.16, we only need to verify that for all $h \in [H]$,

$$\log\mathcal{N}_{\Theta_{h;o}}(\rho) = \mathcal{O}\left(d_o\log(Rd_o/\rho)\right), \qquad \log\mathcal{N}_{\Theta_{h;t}}(\rho) = \mathcal{O}\left(d_s\log(Rd_s/\rho)\right).$$

---

[16]Here, for $h = 0$, we take $\Theta_{0;t} = \left\{\mu_1^{\theta}\right\}_{\theta\in\Theta}$.

[17]The optimistic covers of the emission matrices $\Theta_{h;o}$ and transitions $\Theta_{h;t}$ are defined as in Chen et al. (2022, Definition C.5) with context $\pi$ being $s$ and $(s,a)$, and output being $o$ and $s$, respectively.

We demonstrate how to construct a $\rho$-optimistic covering for $\Theta_{h;o}$; the construction for $\Theta_{h;t}$ is essentially the same. In the following, we follow the idea of (Chen et al., 2022, Proposition H.15).

Fix a $h \in [H]$ and set $N = \lceil R/\rho \rceil$. Let $R' = N\rho$, for $u \in [-R', R']^{d_o}$, we define the $\rho$-neighborhood of $u$ as $\mathbb{B}_\infty(u, \rho) := \rho \lfloor u/\rho \rfloor + [0, \rho]^d$, and let

$$\widetilde{\mathbb{O}}_{h;u}(o|s) := \max_{u' \in \mathbb{B}_\infty(u,\rho)} \left\langle u', \varphi_h(o|s) \right\rangle.$$

Then, if $u$ induces a emission dynamic $\mathbb{O}_{h;v}$, then $\widetilde{\mathbb{O}}_{h;u}(o|s) \geqslant \mathbb{O}_{h;u}(o|s)$, and

$$\sum_o \left| \widetilde{\mathbb{O}}_{h;u}(o|s) - \mathbb{O}_{h;u}(o|s) \right| = \sum_o \max_{u' \in \mathbb{B}_\infty(u,\rho)} \left| \left\langle u' - u, \varphi_h(o|s) \right\rangle \right| \leqslant \rho \sum_o \|\varphi_h(o|s)\|_1 \leqslant R\rho.$$

Therefore, we can pick each $\widetilde{\mathbb{O}}_{h;u}(\cdot|\cdot)$ a representative $u$ such that $u$ induce a lawful emission dynamic; there are at most $(2N)^{d_o}$ many elements in the set $\left\{ \widetilde{\mathbb{O}}_{h;u}(\cdot|\cdot) \right\}_{u \in [-R',R']^{d_o}}$, and hence by doing this, we obtain a $R\rho$-optimistic covering $(\widetilde{\mathbb{O}}, \Theta'_{h;o})$ of $\Theta_{h;o}$ such that $\left| \Theta'_{h;o} \right| \leqslant (2\lceil R/\rho \rceil)^{d_o}$. This proves Proposition D.15. $\qquad\square$

*Proof of Lemma D.16.* Fix a $\rho_1 \in (0, 1]$ and let $\rho = \rho_1/3H$.

Note that given a tuple of parameters $(\widetilde{\mu}_1, \widetilde{\mathbb{T}}, \widetilde{\mathbb{O}})$ (not necessarily induce a POMDP model), we can define $\widetilde{\mathbb{P}}$ as

$$\widetilde{\mathbb{P}}(\tau_H) = \sum_{s_1, \cdots, s_H} \widetilde{\mu}_1(s_1) \widetilde{\mathbb{O}}_1(o_1|s_1) \widetilde{\mathbb{T}}_1(s_2|s_1, a_1) \cdots \widetilde{\mathbb{T}}_{H-1}(s_H|s_{H-1}, a_{H-1}) \widetilde{\mathbb{O}}(o_H|s_H),$$

and $\widetilde{\mathbb{P}}^\pi(\tau_H) = \pi(\tau_H) \times \widetilde{\mathbb{P}}(\tau_H)$. Then for a tuple of parameters $(\mu_1, \mathbb{T}, \mathbb{O})$ that induce a POMDP such that

$$\|\widetilde{\mu}_1 - \mu_1\|_1 \leqslant \rho^2, \qquad \max_{s,a,h} \left\| (\widetilde{\mathbb{T}}_h - \mathbb{T}_h)(\cdot|s, a) \right\|_1 \leqslant \rho^2, \qquad \max_{s,h} \left\| (\widetilde{\mathbb{O}}_h - \mathbb{O}_h)(\cdot|s) \right\|_1 \leqslant \rho^2,$$

it holds that

$$\left\| \widetilde{\mathbb{P}}^\pi(\cdot) - \mathbb{P}^\pi(\cdot) \right\|_1 = \sum_{\tau_H} \left| \widetilde{\mathbb{P}}^\pi(\tau_H) - \mathbb{P}^\pi(\tau_H) \right|$$

$$\leqslant \sum_{s_{1:H}, \tau_H} \left\{ \pi(\tau_H) |\widetilde{\mu}_1(s_1) - \mu_1(s_1)| \, \widetilde{\mathbb{O}}_1(o_1|s_1) \widetilde{\mathbb{T}}_1(s_2|s_1, a_1) \cdots \widetilde{\mathbb{O}}(o_H|s_H) \right.$$

$$+ \pi(\tau_H) \mu_1(s_1) \left| \widetilde{\mathbb{O}}_1(o_1|s_1) - \mathbb{O}_h(o_1|s_1) \right| \widetilde{\mathbb{T}}_1(s_2|s_1, a_1) \cdots \widetilde{\mathbb{O}}(o_H|s_H)$$

$$+ \pi(\tau_H) \mu_1(s_1) \mathbb{O}_1(o_1|s_1) \left| \widetilde{\mathbb{T}}_1(s_2|s_1, a_1) - \mathbb{T}_1(s_2|s_1, a_1) \right| \cdots \widetilde{\mathbb{O}}(o_H|s_H)$$

$$+ \cdots$$

$$\left. + \pi(\tau_H) \mu_1(s_1) \mathbb{O}_1(o_1|s_1) \mathbb{T}_1(s_2|s_1, a_1) \cdots \left| \widetilde{\mathbb{O}}(o_H|s_H) - \mathbb{O}(o_H|s_H) \right| \right\}$$

$$\overset{(*)}{\leqslant} 2H\rho^2 (1 + \rho^2)^{2H} \leqslant 4H\rho^2 \leqslant \rho_1^2,$$

where $(*)$ is because $\sum_{s_{h+1}} \widetilde{\mathbb{T}}_h(s_{h+1}|s_h, a_h) \leqslant 1 + \rho^2$ and $\sum_{o_h} \mathbb{O}_h(o_h|s_h) \leqslant 1 + \rho^2$ for all $h, s_h, a_h$.

Therefore, suppose that for each $h$, $(\widetilde{\mathbb{T}}_h, \Theta'_{h;t})$ is a $\rho$-optimistic covering of $\Theta_{h;t}$, and $(\widetilde{\mathbb{O}}_h, \Theta'_{h;o})$ is a $\rho$-optimistic covering of $\Theta_{h;o}$, then we can obtain a $\rho_1$-optimistic covering $(\widetilde{\mathbb{P}}, \Theta')$ of $\Theta$, where

$$\Theta' = \Theta'_{0;t} \times \Theta'_{1;o} \times \Theta'_{1;t} \times \cdots \times \Theta'_{H-1;t} \times \Theta'_{H;o}.$$

This completes the proof. $\qquad\square$

### D.3.5 DECODABLE POMDPS

To construct a B-representation for the decodable POMDP, we introduce the following notation. For $h \leq H - m$, we consider $t_h = (o_h, a_h, \cdots, o_{h+m-1}) \in \mathcal{U}_h$, $t_{h+1} = (o'_{h+1}, a'_{h+1}, \cdots, o'_{h+m}) \in \mathcal{U}_{h+1}$, and define

$$\mathbb{P}_h(t_{h+1}|t_h) = \begin{cases} \mathbb{P}(o_{h+m} = o'_{h+m}|s_{h+m-1} = \phi_{h+m-1}(t_h), a_{h+m-1}), & \text{if } o_{h+1:h+m-1} = o'_{h+1:h+m-1} \\ & \text{and } a_{h+1:h+m-2} = a'_{h+1:h+m-2}, \\ 0, & \text{otherwise}, \end{cases} \tag{26}$$

where $\phi_{h+m-1}$ is the decoder function that maps $t_h$ to a latent state $s_{h+m-1}$. Similarly, for $h > H - m$, $t_h \in \mathcal{U}_h$, $t_{h+1} \in \mathcal{U}_{h+1}$, we let $\mathbb{P}_h(t_{h+1}|t_h)$ be 1 if $t_h$ ends with $t_{h+1}$, and 0 otherwise.

Under such definition, for all $h \in [H]$, $t_h \in \mathcal{U}_h$, $t_{h+1} \in \mathcal{U}_{h+1}$, it is clear that

$$\mathbb{P}_h(t_{h+1}|t_h) = \mathbb{P}(t_{h+1}|t_h, \tau_{h-1}) \tag{27}$$

for any reachable $(\tau_{h-1}, t_h)$, because of decodability. Hence, we can interpret $\mathbb{P}_h(t_{h+1}|t_h)$ as the probability of observing $t_{h+1}$ conditional on observing $t_h$ on step $h$. [18] Then, for $h \in [H]$, we can take

$$\mathbf{B}_h(o, a) = \left[ \mathbb{1}((o, a) \to t_h) \mathbb{P}_h(t_{h+1}|t_h) \right]_{(t_{h+1}, t_h) \in \mathcal{U}_{h+1} \times \mathcal{U}_h}, \tag{28}$$

where $\mathbb{1}((o, a) \to t_h)$ is 1 if $t_h$ starts with $(o, a)$ and 0 otherwise[19].

We verify that (28) indeed gives a B-representation for decodable POMDPs:

**Proposition D.17** (Decodable POMDPs are B-stable). *(28) gives a B-stable B-representation of the m-step decodable POMDP, with $\Lambda_B = 1$.*

The results above already guarantee the sample complexity of EXPLORATIVE E2D for decodable POMDPs. For OMLE, we can similarly obtain that $\sum_{o,a} \|\mathbf{B}_h(o, a)x\|_1 = A \|x\|_1$, and thus we can take $R_B = A$. Combining this fact with Theorem 9 establishes the sample complexity of OMLE as claimed in Section 5.

*Proof of Proposition D.17.* We verify that (28) gives a B-representation for decodable POMDP: Note that for $h \in [H-1]$, $(o_h, a_h) \in \mathcal{O} \times \mathcal{A}$, $t_{h+1} \in \mathcal{U}_{h+1}$, there is a unique element $t_h \in \mathcal{U}_h$ such that $t_h$ is the prefix of the trajectory $(o_h, a_h, t_{h+1})$, and it holds that

$$\mathbf{e}_{t_{h+1}}^\top \mathbf{B}_h(o_h, a_h)x = \mathbb{P}_h(t_{h+1}|t_h) \times x(t_h).$$

Applying this equality recursively, we obtain the following fact: For trajectory $\tau_{h':h}$ and $t_{h+1} \in \mathcal{U}_{h+1}$, $(\tau_{h':h}, t_{h+1})$ has a prefix $t_{h'} \in \mathcal{U}_{h'}$, and

$$\mathbf{e}_{t_{h+1}}^\top \mathbf{B}_{h:h'}(\tau_{h':h})x = \mathbb{P}(\tau_{h':h}, t_{h+1}|t_{h'}) \times x(t_{h'}), \tag{29}$$

where $\mathbb{P}(\tau_{h':h}, t_{h+1}|t_{h'})$ stands for the probability of observing $(\tau_{h':h}, t_{h+1})$ conditional on observing $t_{h'}$ at step $h'$, which is well-defined due to decodability (similar to (27)).

Taking $h' = 1$ and $x = \mathbf{q}_0$ in (29), we have for any history $\tau_h$ and $t_{h+1} \in \mathcal{U}_{h+1}$ that

$$\mathbb{P}(\tau_h, t_{h+1}) = \mathbf{e}_{t_{h+1}}^\top \mathbf{B}_{h:1}(\tau_h)\mathbf{q}_0.$$

Therefore, (28) indeed gives a B-representation of the decodable POMDP.

Furthermore, we can take $h = H$ in (29) to obtain that: For any trajectory $\tau_{h:H} = (o_h, a_h, \cdots, o_H, a_H)$, it has a prefix $t_h \in \mathcal{U}_h$, and

$$\mathbf{B}_{H:h}(\tau_{h:H})x = \mathbb{P}(\tau_{h:H}|t_h) \times x(t_h).$$

Hence, for any policy $\pi$, it holds that

$$\sum_{\tau_{h:H}} \pi(\tau_{h:H}) \times |\mathbf{B}_{H:h}(\tau_{h:H})x| = \sum_{\tau_{h:H}} \mathbb{P}^\pi(\tau_{h:H}|t_h) \times |x(t_h)| = \sum_{t_h \in \mathcal{U}_h} \pi(t_h) \times |x(t_h)|.$$

Therefore, $\|\mathcal{B}_{H:h}x\|_\Pi \leq \|x\|_\Pi$ always. This completes the proof of Proposition D.17. $\square$

---

[18] It is worth noting that the $(\mathbb{P}_h)$ we define is exactly the transition dynamics of the associated *megastate MDP* (Efroni et al., 2022).

[19] For $h = H$, we understand $\mathbf{B}_H(o, a) = [\mathbb{1}(t = o)]_{t \in \mathcal{U}_H}$ because $o_{H+1} = o_{\text{dum}}$ always.

### D.3.6 REGULAR PSRs

**Proposition D.18** (Regular PSRs are B-stable). *Any $\alpha_{\mathsf{psr}}$-regular PSR admits a B-representation ($\mathbf{B}$) such that for all $1 \leqslant h \leqslant H$, $\|\mathcal{B}_{H:h}x\|_{\Pi} \leqslant \|K_h^{\dagger}x\|_1$, where $K_h$ is any core matrix of $D_h$ (cf. Example 8). Hence, any $\alpha_{\mathsf{psr}}$-regular PSR is B-stable with $\Lambda_{\mathsf{B}} \leqslant \sqrt{U_A}\alpha_{\mathsf{psr}}^{-1}$. As a byproduct, we show that the B-representation also has $R_{\mathsf{B}} \leqslant \alpha_{\mathsf{psr}}^{-1}AU_A$.*

*Proof of Proposition D.18.* By (Zhan et al., 2022, Lemma 6), the PSR admits a B-representation such that $\mathrm{rowspan}(\mathbf{B}_h(o,a)) \subset \mathrm{colspan}(D_{h-1})$. In the following, we show that such a B-representation is indeed what we want.

Fix a core matrix $K_{h-1}$ of $D_{h-1}$, and suppose that $K_{h-1} = \left[\mathbf{q}(\tau_{h-1}^1), \cdots, \mathbf{q}(\tau_{h-1}^d)\right]$ with $d = \mathrm{rank}(D_h)$. Then it holds that

$$
\begin{aligned}
\|\mathcal{B}_{H:h}x\|_{\Pi} &= \max_{\pi} \sum_{\tau_{h:H}} \pi(\tau_{h:H}) \times |\mathbf{B}_{H:h}(\tau_{h:H})x| \\
&= \max_{\pi} \sum_{\tau_{h:H}} \pi(\tau_{h:H}) \times \left|\mathbf{B}_{H:h}(\tau_{h:H})K_{h-1}K_{h-1}^{\dagger}x\right| \\
&\leqslant \max_{\pi} \sum_{\tau_{h:H}} \pi(\tau_{h:H}) \times \sum_{j=1}^{d} |\mathbf{B}_{H:h}(\tau_{h:H})K_{h-1}\mathbf{e}_j| \cdot \left|\mathbf{e}_j^{\top}K_{h-1}^{\dagger}x\right| \\
&= \max_{\pi} \sum_{j=1}^{d} \left|\mathbf{e}_j^{\top}K_{h-1}^{\dagger}x\right| \times \sum_{\tau_{h:H}} \pi(\tau_{h:H}) \times \sum_{j=1}^{d} \left|\mathbf{B}_{H:h}(\tau_{h:H})\mathbf{q}(\tau_{h-1}^j)\right|.
\end{aligned}
$$

Notice that $B_{H:h}(\tau_{h:H})\mathbf{q}(\tau_{h-1}^j) = \mathbb{P}(\tau_{h:H}|\tau_{h-1}^j)$ by Corollary D.2, and hence for any policy $\pi$, we have

$$
\sum_{\tau_{h:H}} \pi(\tau_{h:H}) \times \left|\mathbf{B}_{H:h}(\tau_{h:H})\mathbf{q}(\tau_{h-1}^j)\right| = \sum_{\tau_{h:H}} \mathbb{P}^{\pi}(\tau_{h:H}|\tau_{h-1}^j) = 1
$$

Therefore, it holds that $\|\mathcal{B}_{H:h}x\|_{\Pi} \leqslant \left\|K_{h-1}^{\dagger}x\right\|_1$ for $h \in [H]$ and any core matrix $K_{h-1}$ of $D_{h-1}$.

Similarly, we can pick a core matrix $K_{h-1}$ such that $\|K_{h-1}^{\dagger}\|_1 \leqslant \alpha_{\mathsf{psr}}^{-1}$, then

$$
\sum_{o,a} \|\mathbf{B}_h(o,a)x\|_1 = \sum_{o,a} \left\|\mathbf{B}_h(o,a)K_{h-1}K_{h-1}^{\dagger}x\right\|_1 \leqslant A\left|\mathcal{U}_{A,h+1}\right|\left\|K_{h-1}^{\dagger}x\right\|_1 \leqslant \alpha_{\mathsf{psr}}^{-1}AU_A\|x\|_1.
$$

This completes the proof. $\qquad\square$

### D.4 COMPARISON WITH WELL-CONDITIONED PSRs

Concurrent work by Liu et al. (2022b) defines the following class of well-conditioned PSRs.

**Definition D.19.** *A PSR is $\gamma$-well-conditioned if it admits a B-representation such that for all $h \in [H]$, policy $\pi$ (that starts at step $h$), vector $x \in \mathbb{R}^{\mathcal{U}_h}$, the following holds:*

$$
\sum_{\tau_{h:H}} \pi(\tau_{h:H}) \times |\mathbf{B}_H(o_H, a_H)\cdots\mathbf{B}_h(o_h, a_h)x| \leqslant \frac{1}{\gamma}\|x\|_1, \tag{30}
$$

$$
\sum_{o_h,a_h} \pi(a_h|o_h) \times \|\mathbf{B}_h(o_h, a_h)x\|_1 \leqslant \frac{1}{\gamma}\|x\|_1. \tag{31}
$$

By (30) and the inequality $\|x\|_1 \leqslant \sqrt{U_A}\|x\|_*$ (Lemma D.6), any $\gamma$-well-conditioned PSR is a B-stable PSR with $\Lambda_{\mathsf{B}} \leqslant \sqrt{U_A}\gamma^{-1}$. Plugging this into our main results shows that, for well-conditioned PSRs, OMLE, EXPLORATIVE E2D and MOPS all achieve sample complexity

$$
\widetilde{\mathcal{O}}\left(\frac{dAU_A^2H^2\log\mathcal{N}_{\Theta}}{\gamma^2\varepsilon^2}\right),
$$

which is better than the sample complexity[20] $\tilde{\mathcal{O}}\left(d^2 A^5 U_A^3 H^4 \log \mathcal{N}_\Theta / \gamma^4 \varepsilon^2\right)$ achieved by the analysis of OMLE in Liu et al. (2022b). Also, being well-conditioned imposes the extra restriction (31) on the structure of the PSR, while our B-stability condition does not.

# E    DECORRELATION ARGUMENTS

In this section, we present two decorrelation propositions: the generalized $\ell_2$-Eluder argument (Proposition E.1), and the decoupling argument (Proposition E.6). These two propositions are important steps in the proof of main theorems (Theorem 9, 10, H.4, H.6). These two Propositions are parallel: Proposition E.1 is the triangular-to-diagonal version of the decorrelation used in the proof of Theorem 9 (see Appendix G for its proof), whereas Proposition E.6 is the expectation-to-expectation version of the decorrelation used in the proof of Theorem 10 (see Appendix I for its proof).

## E.1    GENERALIZED $\ell_2$-ELUDER ARGUMENT

We first present the triangular-to-diagonal version of the decorrelation argument, the generalized $\ell_2$-Eluder argument.

**Proposition E.1** (Generalized $\ell_2$-Eluder argument). *Suppose we have sequences of vectors*

$$\{x_{k,i}\}_{(k,i)\in[K]\times\mathcal{I}} \subset \mathbb{R}^d, \qquad \{y_{k,j,r}\}_{(k,j,r)\in[K]\times[J]\times\mathcal{R}} \subset \mathbb{R}^d$$

*where $\mathcal{I}, \mathcal{R}$ are arbitrary (abstract) index sets. Consider functions $\{f_k : \mathbb{R}^d \to \mathbb{R}\}_{k\in[K]}$:*

$$f_k(x) := \max_{r\in\mathcal{R}} \sum_{j=1}^J |\langle x, y_{k,j,r}\rangle|.$$

*Assume that the following condition holds:*

$$\sum_{t=1}^{k-1} \mathbb{E}_{i\sim q_t}\left[f_k(x_{t,i})^2\right] \le \beta_k, \qquad \forall k \in [K],$$

*where $(q_k \in \Delta(\mathcal{I}))_{k\in[K]}$ is a family of distributions over $\mathcal{I}$.*

*Then for any $M > 0$, it holds that*

$$\sum_{t=1}^k M \wedge \mathbb{E}_{i\sim q_t}[f_t(x_{t,i})] \le \sqrt{2d\left(M^2 k + \sum_{t=1}^k \beta_t\right) \log\left(1 + \frac{k}{d}\frac{R_x^2 R_y^2}{M^2}\right)}, \qquad \forall k \in [K],$$

*where $R_x^2 = \max_k \mathbb{E}_{i\sim q_k}\left[\|x_{k,i}\|_2^2\right]$, $R_y = \max_{k,r} \sum_j \|y_{k,j,r}\|_2$.*

We call this proposition "generalized $\ell_2$-Eluder argument" because, when $\mathcal{I}$ is a single element set and $\beta_k = \beta$, the result reduces to

$$\text{if } \sum_{t<k} f_k(x_t)^2 \le \beta, \text{ for all } k \in [K], \text{ then } \sum_{t=1}^k |f_t(x_t)| \le \tilde{\mathcal{O}}\left(\sqrt{d\beta k}\right), \tag{32}$$

as long as $\max_t |f_t(x_t)| \le 1$, which implies that the function class $\{f_t\}_t$ has Eluder dimension $\tilde{\mathcal{O}}(d)$. In particular, when $\{f_k\}_{k\in[K]}$ is given by $f_k(x) = |\langle y_k, x\rangle|$, (32) is equivalent to the standard $\ell_2$-Eluder argument for linear functions, which can be proved using the elliptical potential lemma (Lattimore & Szepesvári, 2020, Lemma 19.4).

In the following, we present a corollary of Proposition E.1 that is more suitable for our applications.

---

[20]Liu et al. (2022b) only asserts a polynomial rate without spelling out the concrete powers of the problem parameters. This rate is extracted from Liu et al. (2022b, Proposition C.5 & Lemma C.6).

**Corollary E.2.** *Suppose we have a sequence of functions* $\{f_k : \mathbb{R}^n \to \mathbb{R}\}_{k \in [K]}$:

$$f_k(x) := \max_{r \in \mathcal{R}} \sum_{j=1}^{J} |\langle x, y_{k,j,r} \rangle|,$$

*which is given by the family of vectors* $\{y_{k,j,r}\}_{(k,j,r) \in [K] \times [J] \times \mathcal{R}} \subset \mathbb{R}^n$. *Further assume that there exists* $L > 0$ *such that* $f_k(x) \leqslant L \|x\|_1$.

*Consider further a sequence of vector* $(x_i)_{i \in \mathcal{I}}$, *satisfying the following condition*

$$\sum_{t=1}^{k-1} \mathbb{E}_{i \sim q_t}\left[ f_k^2(x_i) \right] \leqslant \beta_k, \qquad \forall k \in [K],$$

*and the subspace spanned by* $(x_i)_{i \in \mathcal{I}}$ *has dimension at most* $d$. *Then it holds that*

$$\sum_{t=1}^{k} 1 \wedge \mathbb{E}_{i \sim q_t}[f_t(x_i)] \leqslant \sqrt{4d\Big(k + \sum_{t=1}^{k} \beta_t\Big) \log\Big(1 + kdL \max_i \|x_i\|_1\Big)}, \qquad \forall k \in [K].$$

We prove Proposition E.1 and Corollary E.2 in the following subsections.

**Remark E.3.** In the initial version of this paper, the statement of Corollary E.2 was slightly different from above, which states that under the same precondition,

$$\sum_{t=1}^{k} 1 \wedge \mathbb{E}_{i \sim q_t}[f_t(x_i)] \leqslant \sqrt{4d\Big(k + \sum_{t=1}^{k} \beta_t\Big) \log\left(1 + kdL\kappa_d(X)\right)}, \qquad \forall k \in [K],$$

where matrix $X := [x_i]_{i \in \mathcal{I}} \in \mathbb{R}^{n \times \mathcal{I}}$ and

$$\kappa_d(X) = \min\left\{ \|F_1\|_1 \|F_2\|_1 : X = F_1 F_2, F_1 \in \mathbb{R}^{n \times d}, F_2 \in \mathbb{R}^{d \times \mathcal{I}} \right\}.$$

After our initial version, we noted the concurrent work Liu et al. (2022b, Lemma G.3) which essentially shows that $\kappa_d(X) \leqslant d \max_i \|x_i\|_1$ by an elegant argument using the Barycentric spanner. For the sake of simplicity, we have applied their result (cf. Lemma E.5) to make Corollary E.2 slightly more convenient to use.

We also note that, in the initial version of this paper, in the statement of Theorem 9, the sample complexity involved a log factor $\iota := \log(1 + Kd\Lambda_B R_B \kappa_d)$, where $\kappa_d := \max_h \kappa_d(D_h)$, which we then tightly bounded for all concrete problem classes in terms of the corresponding problem parameter. The above change makes the statement slightly cleaner (though the result slightly looser) by always using the bound $\kappa_d \leqslant dU_A$. The effect on the final result is however minor, as the sample complexity of OMLE only depends on $\kappa_d$ logarithmically through $\iota$, and the sample complexity of MOPS or EXPLORATIVE E2D does not involve this factor.

### E.1.1 PROOF OF PROPOSITION E.1

To prove this proposition, we first show that the proposition can be reduced to the case when $n = 1$, extending the idea of the proof of (Liu et al., 2022a, Proposition 22). After that, we invoke a certain variant of the elliptical potential lemma to derive the desired inequality.

We first transform and reduce the problem. For every pair of $(k, i) \in [K] \times \mathcal{I}$, we take $r^*(k, i) := \arg\max_r \sum_j |\langle x_{k,i}, y_{k,j,r} \rangle|$, and consider

$$\widetilde{y}_{k,i,j} := y_{k,j,r^*(k,i)} \quad \forall (k, i, j) \in [K] \times \mathcal{I} \times [n].$$

We then define

$$\widetilde{y}_{k,i} := \sum_j \widetilde{y}_{k,i,j} \operatorname{sign} \langle \widetilde{y}_{k,i,j}, x_{k,i} \rangle \quad \forall (k, i) \in [K] \times \mathcal{I}.$$

Under such a transformation, it holds that for all $t, k, i, i'$,

$$|\langle x_{t,i}, \widetilde{y}_{t,i} \rangle| = \sum_j |\langle x_{t,i}, y_{t,j,r^*(t,i)} \rangle| = \max_r \sum_j |\langle x_{t,i}, y_{t,j,r} \rangle| = f_t(x_{t,i}),$$

$$|\langle x_{t,i}, \widetilde{y}_{k,i'} \rangle| \leqslant \max_r \sum_j |\langle x_{t,i}, y_{k,j,r} \rangle| = f_k(x_{t,i}), \qquad \|\widetilde{y}_{k,i}\|_2 \leqslant R_y.$$

Therefore, it remains to bound $\sum_{t=1}^k M \wedge \mathbb{E}_{i \sim q_t} |\langle x_{t,i}, \widetilde{y}_{t,i} \rangle|$, under the condition that for all $k \in [K]$, $\sum_{t<k} \mathbb{E}_{i \sim q_t}[\max_{i'} |\langle x_{t,i}, \widetilde{y}_{k,i'} \rangle|^2] \leqslant \beta_k$.

To show this, we define $\Phi_t := \mathbb{E}_{i \sim q_t}[x_{t,i} x_{t,i}^\top]$, and take $\lambda_0 = \frac{M^2}{R_y^2}$, $V_k := \lambda_0 I + \sum_{t<k} \Phi_t$. Then

$$
\begin{aligned}
\sum_{t=1}^k M \wedge \mathbb{E}_{i \sim q_t} |\langle x_{t,i}, \widetilde{y}_{t,i} \rangle| &\leqslant \sum_{t=1}^k \min \left\{ M, \mathbb{E}_{i \sim q_t} \left[ \|x_{t,i}\|_{V_t^{-1}} \|\widetilde{y}_{t,i}\|_{V_t} \right] \right\} \\
&\leqslant \sum_{t=1}^k \min \left\{ M, \sqrt{(M^2 + \beta_t) \mathbb{E}_{i \sim q_t} \left[ \|x_{t,i}\|_{V_t^{-1}}^2 \right]} \right\} \\
&\leqslant \sum_{t=1}^k \sqrt{(M^2 + \beta_t) \min \left\{ 1, \mathbb{E}_{i \sim q_t} \left[ \|x_{t,i}\|_{V_t^{-1}}^2 \right] \right\}} \\
&\leqslant \left( kM^2 + \sum_{t=1}^k \beta_t \right)^{\frac{1}{2}} \left( \sum_{t=1}^k \min \left\{ 1, \mathbb{E}_{i \sim q_t} \left[ \|x_{t,i}\|_{V_t^{-1}}^2 \right] \right\} \right)^{\frac{1}{2}},
\end{aligned}
$$

where the second inequality is due to the fact that for all $(t, i)$,

$$\|\widetilde{y}_{t,i}\|_{V_t}^2 = \lambda_0 \|\widetilde{y}_{t,i}\|^2 + \sum_{s<t} \mathbb{E}_{i' \sim q_s} |\langle x_{s,i'}, \widetilde{y}_{t,i} \rangle|^2 \leqslant M^2 + \beta_t.$$

Note that

$$\mathbb{E}_{i \sim q_t} \left[ \|x_{t,i}\|_{V_t^{-1}}^2 \right] = \mathbb{E}_{i \sim q_t} \left[ \mathrm{tr} \left( V_k^{-\frac{1}{2}} x_{k,i} x_{k,i}^\top V_k^{-\frac{1}{2}} \right) \right] = \mathrm{tr} \left( V_k^{-\frac{1}{2}} \Phi_k V_k^{-\frac{1}{2}} \right).$$

In order to bound the term $\sum_{t=1}^k \min\{1, \mathrm{tr}(V_k^{-1/2} \Phi_k V_k^{-1/2})\}$, we invoke the following standard lemma, which generalizes Lattimore & Szepesvári (2020, Lemma 19.4).

**Lemma E.4** (Generalized elliptical potential lemma). *Let $\{\Phi_k \in \mathbb{R}^{d \times d}\}_{k \in [K]}$ be a sequence of symmetric semi-positive definite matrix, and $V_k := \lambda_0 I + \sum_{t<k} \Phi_t$, where $\lambda_0 > 0$ is a fixed real. Then it holds that*

$$\sum_{k=1}^K \min \left\{ 1, \mathrm{tr} \left( V_k^{-\frac{1}{2}} \Phi_k V_k^{-\frac{1}{2}} \right) \right\} \leqslant 2d \log \left( 1 + \frac{\sum_{k=1}^K \mathrm{tr}(\Phi_k)}{d \lambda_0} \right).$$

Applying Lemma E.4 and noticing $\mathrm{tr}(\Phi_t) = \mathbb{E}_{i \sim q_t}[\|x_{t,i}\|_2^2] \leqslant R_x^2$, the proof of Proposition E.1 is completed. $\qquad \square$

*Proof of Lemma E.4.* By definition and by linear algebra, we have

$$V_{k+1} = V_k^{\frac{1}{2}} \left( I + V_k^{-\frac{1}{2}} \Phi_k V_k^{-\frac{1}{2}} \right) V_k^{\frac{1}{2}},$$

and hence $\det(V_{k+1}) = \det(V_k) \det(I + V_k^{-\frac{1}{2}} \Phi_k V_k^{-\frac{1}{2}})$. Therefore, we have

$$
\begin{aligned}
\sum_{k=1}^K \min \left\{ 1, \mathrm{tr} \left( V_k^{-\frac{1}{2}} \Phi_k V_k^{-\frac{1}{2}} \right) \right\} &\leqslant \sum_{k=1}^K 2 \log \left( 1 + \mathrm{tr} \left( V_k^{-\frac{1}{2}} \Phi_k V_k^{-\frac{1}{2}} \right) \right) \\
&\leqslant 2 \sum_{k=1}^K \log \det \left( 1 + V_k^{-\frac{1}{2}} \Phi_k V_k^{-\frac{1}{2}} \right) \\
&= 2 \sum_{k=1}^K \left[ \log \det(V_{k+1}) - \log \det(V_k) \right] \\
&= 2 \log \frac{\det(V_{K+1})}{\det(V_0)},
\end{aligned}
$$

where the first inequality is due to the fact that $\min\{1, u\} \leqslant 2\log(1 + u)$, $\forall u \geqslant 0$, and the second inequality is because for any positive semi-definite matrix $X$, it holds $\det(I + X) \geqslant 1 + \operatorname{tr}(X)$. Now, we have

$$\log\det(V_{K+1}) \leqslant \log\left(\frac{\operatorname{tr}(V_{K+1})}{d}\right)^d = d\log\left(\lambda_0 + \frac{\sum_{k=1}^{K}\operatorname{tr}(\Phi_k)}{d}\right),$$

which completes the proof of Lemma E.4. $\qquad\square$

### E.1.2 Proof of Corollary E.2

Let us take a decomposition $x_i = Fv_i \forall i \in \mathcal{I}$, such that $\|v_i\|_\infty \leqslant 1$ and $\|F\|_{1\to 1} \leqslant \max_i \|x_i\|_1$ (the existence of such a decomposition is guaranteed by Lemma E.5). We define $\widetilde{f}_k : \mathbb{R}^d \to \mathbb{R}$ as follows:

$$\widetilde{f}_k(v) := f_k(Fv) = \max_r \sum_j \left|\langle v, F^\top y_{k,j,r}\rangle\right|.$$

By definition, $\widetilde{f}_k(v_i) = f_k(x_i)$, and hence our condition becomes

$$\sum_{t<k} \mathbb{E}_{i\sim q_t}\left[\widetilde{f}_k^2(v_i)\right] \leqslant \beta_k, \qquad \forall k \in [K],$$

Then applying Proposition E.1 gives for all $k \in [K]$,

$$\sum_{t=1}^{k} 1 \wedge \mathbb{E}_{i\sim q_t}[f_t(x_i)] = \sum_{t=1}^{k} 1 \wedge \mathbb{E}_{i\sim q_t}\left[\widetilde{f}_t(v_i)\right] \leqslant \sqrt{2d\Big(k + \sum_{t=1}^{k}\beta_t\Big)\log\left(1 + kd^{-1}\cdot R_2^2 R_1^2\right)},$$

where $R_2 \leqslant \max_i \|v_i\|_2 \leqslant \sqrt{d}$, and

$$R_1 = \max_{k,r}\sum_j \left\|F^\top y_{k,j,r}\right\|_2 \leqslant \max_{k,r}\sum_j \left\|F^\top y_{k,j,r}\right\|_1 \leqslant \max_{k,r}\sum_j \sum_{m=1}^{d} \left|\mathbf{e}_m^\top F^\top y_{k,j,r}\right|$$

$$= \max_{k,r}\sum_j \sum_{m=1}^{d} \left|\langle F\mathbf{e}_m, y_{k,j,r}\rangle\right| \leqslant \max_k \sum_{m=1}^{d} f_k(F\mathbf{e}_m) \leqslant \sum_{m=1}^{d} L\|F\mathbf{e}_m\|_1 \leqslant dL\|F\|_1 \leqslant dL\max_i \|x_i\|_1.$$

Therefore, we have

$$\log\left(1 + kd^{-1}\cdot R_1^2 R_2^2\right) \leqslant \log\left(1 + kd^2 L^2 \max_i \|x_i\|_1^2\right) \leqslant 2\log(1 + kdL\max_i \|x_i\|_1),$$

which completes the proof of Corollary E.2. $\qquad\square$

The following lemma is an immediate consequence of Liu et al. (2022b, Lemma G.3).

**Lemma E.5.** *Assume that a sequence of vectors $\{x_i\}_{i\in\mathcal{I}} \subset \mathbb{R}^n$ satisfies that $\operatorname{span}(x_i : i \in \mathcal{I})$ has dimension at most $d$ and $R = \max_i \|x\|_1 < \infty$. Then, there exists a sequence of vectors $\{v_i\}_{i\in\mathcal{I}} \subset \mathbb{R}^d$ and a matrix $F \in \mathbb{R}^{n\times d}$, such that $x_i = Fv_i \,\forall i \in \mathcal{I}$, and $\|v_i\|_\infty \leqslant 1, \|F\|_{1\to 1} \leqslant R$.*

*Proof.* Without loss of generality, we assume that $\mathcal{X} = \operatorname{span}(x_i : i \in \mathcal{I})$ has dimension at most $d$. Then $\mathcal{X}$ is a $d$-dimensional compact subset of $\mathbb{R}^n$, and we take a Barycentric spanner of $\mathcal{X}$ to be $\{w_1, \cdots, w_d\}$. By definition, for each $i \in \mathcal{I}$, there exists weights $(\alpha_{ij})_{1\leqslant j\leqslant d}$ such that $\alpha_{ij} \in [-1, 1]$ and $x_i = \sum_{j=1}^{d} \alpha_{ij} w_j$. Therefore, we can take $v_i = [\alpha_{ij}]_{1\leqslant j\leqslant d}^\top \in \mathbb{R}^d$ and $F = [w_1, \cdots, w_d] \in \mathbb{R}^{n\times d}$, and they clearly fulfill the statement of Lemma E.5. $\qquad\square$

### E.2 Decoupling argument

Proposition E.1 can be regarded a triangular-to-diagonal decorrelation result. In this section, we present its expectation-to-expectation analog, which is central for bounding Explorative DEC.

**Proposition E.6** (Decoupling argument). *Suppose we have vectors and functions*

$$\{x_i\}_{i \in \mathcal{I}} \subset \mathbb{R}^n, \qquad \{f_\theta : \mathbb{R}^n \to \mathbb{R}\}_{\theta \in \Theta}$$

*where $\Theta, \mathcal{I}$ are arbitrary abstract index sets, with functions $f_\theta$ given by*

$$f_\theta(x) := \max_{r \in \mathcal{R}} \sum_{j=1}^{J} |\langle x, y_{\theta,j,r} \rangle|, \qquad \forall x \in \mathbb{R}^n,$$

*where $\{y_{\theta,j,r}\}_{(\theta,j,r) \in \Theta \times [J] \times \mathcal{R}} \subset \mathbb{R}^n$ is a family of bounded vectors in $\mathbb{R}^n$. Then for any distribution $\mu$ over $\Theta$ and probability family $\{q_\theta\}_{\theta \in \Theta} \subset \Delta(\mathcal{I})$,*

$$\mathbb{E}_{\theta \sim \mu} \mathbb{E}_{i \sim q_\theta} [f_\theta(x_i)] \leqslant \sqrt{d_X \mathbb{E}_{\theta, \theta' \sim \mu} \mathbb{E}_{i \sim q_{\theta'}} [f_\theta(x_i)^2]},$$

*where $d_X$ is the dimension of the subspace of $\mathbb{R}^n$ spanned by $(x_i)_{i \in \mathcal{I}}$.*

*Proof of Proposition E.6.* By the assumption that $\{y_{\theta,j,r}\}_{(\theta,j,r)}$ is a family of bounded vectors in $\mathbb{R}^d$, there exists $R_y < \infty$ such that $\sup_{\theta,r} \sum_{j=1}^n \|y_{\theta,j,r}\| \leqslant R_y$. We follow the same two steps as the proof of Proposition E.1.

First, we reduce the problem. We consider $r^*(\theta, i) = \arg\max_{r \in \mathcal{R}} \sum_j |\langle x_i, y_{\theta,j,r} \rangle|$, and define the vectors

$$\widetilde{y}_{\theta,i,j} = y_{\theta,j,r^*(\theta,i)},$$
$$\widetilde{y}_{\theta,i} = \sum_j \mathrm{sign}\,\langle x_i, \widetilde{y}_{\theta,i,j} \rangle \, \widetilde{y}_{\theta,i,j}.$$

Then for all $i \in \mathcal{I}$, $\theta \in \Theta$,

$$\langle x_i, \widetilde{y}_{\theta,i} \rangle = \sum_j |\langle x_i, \widetilde{y}_{\theta,i,j} \rangle| = \sum_j |\langle x_i, y_{\theta,j,r^*(\theta,i)} \rangle| = f_\theta(x_i),$$
$$|\langle x_i, \widetilde{y}_{\theta',i'} \rangle| \leqslant \sum_j |\langle x_i, \widetilde{y}_{\theta',i',j} \rangle| = \sum_j |\langle x_i, y_{\theta',j,r^*(\theta',i')} \rangle| \leqslant f_{\theta'}(x_i),$$

(33)

and $\|\widetilde{y}_{\theta,i}\|_2 \leqslant \sum_j \|y_{\theta,j,r^*(\theta,i)}\|_2 \leqslant R_y$. Therefore, it suffices to bound $\mathbb{E}_{\theta \sim \mu} \mathbb{E}_{i \sim q_\theta}[|\langle x_i, \widetilde{y}_{\theta,i} \rangle|]$.

Next, we define $\Phi_\lambda := \lambda + \mathbb{E}_{\theta \sim \mu} \mathbb{E}_{i \sim q_\theta} [x_i x_i^\top]$ with $\lambda > 0$. Then we can bound the target as

$$\mathbb{E}_{\theta \sim \mu} \mathbb{E}_{i \sim q_\theta}[|\langle x_i, \widetilde{y}_{\theta,i} \rangle|] \leqslant \mathbb{E}_{\theta \sim \mu} \mathbb{E}_{i \sim q_\theta} \left[ \|x_i\|_{\Phi_\lambda^{-1}} \|\widetilde{y}_{\theta,i}\|_{\Phi_\lambda} \right]$$
$$\leqslant \left[ \mathbb{E}_{\theta \sim \mu} \mathbb{E}_{i \sim q_\theta} \|x_i\|_{\Phi_\lambda^{-1}}^2 \right]^{1/2} \left[ \mathbb{E}_{\theta \sim \mu} \mathbb{E}_{i \sim q_\theta} \|\widetilde{y}_{\theta,i}\|_{\Phi_\lambda}^2 \right]^{1/2}.$$

The first term can be rewritten as

$$\mathbb{E}_{\theta \sim \mu} \mathbb{E}_{i \sim q_\theta} \left[ \|x_i\|_{\Phi_\lambda^{-1}}^2 \right] = \mathbb{E}_{\theta \sim \mu} \mathbb{E}_{i \sim q_\theta} \left[ \mathrm{tr} \left( \Phi_\lambda^{-1/2} x_i x_i^\top \Phi_\lambda^{-1/2} \right) \right]$$
$$= \mathrm{tr} \left( \Phi_\lambda^{-1/2} \mathbb{E}_{\theta \sim \mu} \mathbb{E}_{i \sim q_\theta} \left[ x_i x_i^\top \right] \Phi_\lambda^{-1/2} \right)$$
$$= \mathrm{tr} \left( \Phi_\lambda^{-1/2} \Phi_0 \Phi_\lambda^{-1/2} \right) \leqslant \mathrm{rank}(\Phi_0) \leqslant d_X.$$

The second term can be bounded as

$$\mathbb{E}_{\theta \sim \mu} \mathbb{E}_{i \sim q_\theta} \|\widetilde{y}_{\theta,i}\|_{\Phi_\lambda}^2 = \mathbb{E}_{\theta' \sim \mu} \mathbb{E}_{i' \sim q_{\theta'}} \|\widetilde{y}_{\theta',i'}\|_{\Phi_\lambda}^2$$
$$= \mathbb{E}_{\theta' \sim \mu} \mathbb{E}_{i' \sim q_{\theta'}} \left\{ \mathbb{E}_{\theta \sim \mu} \mathbb{E}_{i \sim q_\theta} \left[ |\langle x_i, \widetilde{y}_{\theta',i'} \rangle|^2 \right] + \lambda \|\widetilde{y}_{\theta',i'}\|^2 \right\}$$
$$= \mathbb{E}_{\theta' \sim \mu} \mathbb{E}_{\theta \sim \mu} \mathbb{E}_{i \sim q_\theta} \left[ |\langle x_i, \widetilde{y}_{\theta',i'} \rangle|^2 \right] + \lambda \mathbb{E}_{\theta' \sim \mu} \mathbb{E}_{i' \sim q_{\theta'}} \|\widetilde{y}_{\theta',i'}\|^2$$
$$\leqslant \mathbb{E}_{\theta' \sim \mu} \mathbb{E}_{\theta \sim \mu} \mathbb{E}_{i \sim q_\theta} \left[ |f_{\theta'}(x_i)|^2 \right] + \lambda R_y^2,$$

where the last inequality is due to (33). Letting $\lambda \to 0^+$ completes the proof of Proposition E.6. $\square$

# F STRUCTURAL PROPERTIES OF B-STABLE PSRS

In this section, we present two important propositions that are used in the proofs of all the main theorems (Theorem 9, 10, H.4, H.6). The first proposition bounds the performance difference of two PSR models by B-errors. The second proposition bounds the squared B-errors by the Hellinger distance of observation probabilities between two models.

## F.1 PERFORMANCE DECOMPOSITION

We first present the performance decomposition proposition.

**Proposition F.1** (Performance decomposition). *Suppose that two PSR models $\theta, \bar{\theta}$ admit $\{\{\mathbf{B}_h^\theta(o_h, a_h)\}_{h,o_h,a_h}, \mathbf{q}_0^\theta\}$ and $\{\{\mathbf{B}_h^{\bar{\theta}}(o_h, a_h)\}_{h,o_h,a_h}, \mathbf{q}_0^{\bar{\theta}}\}$ as B-representation respectively, and suppose that $\{\mathcal{B}_{H:h}^\theta\}_{h\in[H]}$ and $\{\mathcal{B}_{H:h}^{\bar{\theta}}\}_{h\in[H]}$ are the associated $\mathcal{B}$-operators respectively. Define*

$$\mathcal{E}_{\theta,h}^{\bar{\theta}}(\tau_{h-1}) := \frac{1}{2}\max_\pi \sum_{o_h,a_h} \pi(a_h|o_h)\left\|\mathcal{B}_{H:h+1}^\theta\left(\mathbf{B}_h^\theta(o_h,a_h) - \mathbf{B}_h^{\bar{\theta}}(o_h,a_h)\right)\mathbf{q}^{\bar{\theta}}(\tau_{h-1})\right\|_\Pi,$$

$$\mathcal{E}_{\theta,0}^{\bar{\theta}} := \frac{1}{2}\left\|\mathcal{B}_{H:1}^\theta\left(\mathbf{q}_0^\theta - \mathbf{q}_0^{\bar{\theta}}\right)\right\|_\Pi.$$

*Then it holds that*

$$D_{\mathrm{TV}}\left(\mathbb{P}_\theta^\pi, \mathbb{P}_{\bar{\theta}}^\pi\right) \leqslant \mathcal{E}_{\theta,0}^{\bar{\theta}} + \sum_{h=1}^H \mathbb{E}_{\bar{\theta},\pi}\left[\mathcal{E}_{\theta,h}^{\bar{\theta}}(\tau_{h-1})\right],$$

*where for $h \in [H]$, the expectation $\mathbb{E}_{\bar{\theta},\pi}$ is taking over $\tau_{h-1}$ under model $\bar{\theta}$ and policy $\pi$.*

*Proof of Proposition F.1.* By the definition of B-representation, we have $\mathbb{P}_\theta^\pi(\tau_H) = \pi(\tau_H) \times \mathbf{B}_{H:1}^\theta(\tau_H)\mathbf{q}_0^\theta$ for PSR model $\theta$. Then for two different PSR models $\theta, \bar{\theta}$, we have

$$\mathbb{P}_\theta^\pi(\tau_H) - \mathbb{P}_{\bar{\theta}}^\pi(\tau_H)$$

$$= \pi(\tau_H) \times \left[\mathbf{B}_{H:1}^\theta(\tau_{1:H})\mathbf{q}_0^\theta - \mathbf{B}_{H:1}^{\bar{\theta}}(\tau_{1:H})\mathbf{q}_0^{\bar{\theta}}\right]$$

$$= \pi(\tau_H) \times \mathbf{B}_{H:1}^\theta(\tau_{1:H})\left(\mathbf{q}_0^\theta - \mathbf{q}_0^{\bar{\theta}}\right)$$

$$+ \pi(\tau_H) \times \sum_{h=1}^H \mathbf{B}_{H:h+1}^\theta(\tau_{h+1:H})\left(\mathbf{B}_h^\theta(o_h,a_h) - \mathbf{B}_h^{\bar{\theta}}(o_h,a_h)\right)\mathbf{B}_{1:h-1}^{\bar{\theta}}(\tau_{h-1})\mathbf{q}_0^{\bar{\theta}}$$

$$= \pi(\tau_H) \times \mathbf{B}_{H:1}^\theta(\tau_H)\left(\mathbf{q}_0^\theta - \mathbf{q}_0^{\bar{\theta}}\right)$$

$$+ \sum_{h=1}^H \pi(\tau_{h:H}) \times \mathbf{B}_{H:h+1}^\theta(\tau_{h+1:H})\left(\mathbf{B}_h^\theta(o_h,a_h) - \mathbf{B}_h^{\bar{\theta}}(o_h,a_h)\right)\mathbf{q}^{\bar{\theta}}(\tau_{h-1}) \times \mathbb{P}_{\bar{\theta}}^\pi(\tau_{h-1}),$$

where the last equality is due to the definition of B-representation (see e.g. (15)). Therefore, we have

$$\frac{1}{2}\sum_{\tau_H}\left|\mathbb{P}_\theta^\pi(\tau_H) - \mathbb{P}_{\theta^\star}^\pi(\tau_H)\right|$$

$$\leqslant \frac{1}{2}\sum_{\tau_H}\pi(\tau_H) \times \left|\mathbf{B}_{H:1}^\theta(\tau_H)\left(\mathbf{q}_0^\theta - \mathbf{q}_0^{\bar{\theta}}\right)\right| + \frac{1}{2}\sum_{\tau_H}\sum_{h=1}^H \pi(\tau_{h:H})$$

$$\times \left|\mathbf{B}_{H:h+1}^\theta(\tau_{h+1:H})\left(\mathbf{B}_h^\theta(o_h,a_h) - \mathbf{B}_h^{\bar{\theta}}(o_h,a_h)\right)\mathbf{q}^{\bar{\theta}}(\tau_{h-1})\right| \times \mathbb{P}_{\bar{\theta}}^\pi(\tau_{h-1})$$

$$\leqslant \frac{1}{2}\left\|\mathcal{B}_{H:1}^\theta\left(\mathbf{q}_0^\theta - \mathbf{q}_0^{\bar{\theta}}\right)\right\|_\Pi + \frac{1}{2}\sum_{h=1}^H\sum_{\tau_{h-1}}\mathbb{P}_{\bar{\theta}}^\pi(\tau_{h-1})$$

$$\times \max_\pi \sum_{o_h,a_h}\pi(a_h|o_h)\left\|\mathcal{B}_{H:h+1}^\theta(\tau_{h+1:H})\left(\mathbf{B}_h^\theta(o_h,a_h) - \mathbf{B}_h^{\bar{\theta}}(o_h,a_h)\right)\mathbf{q}^{\bar{\theta}}(\tau_{h-1})\right\|_\Pi$$

$$= \mathcal{E}_{\theta,0}^{\bar{\theta}} + \sum_{h=1}^{H} \mathbb{E}_{\bar{\theta},\pi}\Big[ \mathcal{E}_{\theta,h}^{\bar{\theta}}(\tau_{h-1}) \Big],$$

where the last inequality is due to the definition of $\mathbb{E}_{\bar{\theta},\pi}$ and $\mathcal{E}_{\theta,h}^{\bar{\theta}}(\tau_{h-1})$. $\qquad \square$

## F.2 Bounding the squared B-errors by Hellinger distance

In the following proposition, we show that under B-stability or weak B-stability, the squared B-errors can be bounded by the Hellinger distance between $\mathbb{P}_{\theta}^{\pi_{h},\mathrm{exp}}$ and $\mathbb{P}_{\bar{\theta}}^{\pi_{h},\mathrm{exp}}$. Here, for a policy $\pi \in \Pi$ and $h \in [H]$, $\pi_{h,\mathrm{exp}}$ is defined as

$$\pi_{h,\mathrm{exp}} := \pi \circ_h \mathrm{Unif}(\mathcal{A}) \circ_{h+1} \mathrm{Unif}(\mathcal{U}_{A,h+1}), \tag{34}$$

which is the policy that follows $\pi$ for the first $h-1$ steps, takes $\mathrm{Unif}(\mathcal{A})$ at step $h$, takes an action sequence sampled from $\mathrm{Unif}(\mathcal{U}_{A,h+1})$ at step $h+1$, and behaves arbitrarily afterwards. This notation is consistent with the exploration policy in the OMLE algorithm (Algorithm 1).

**Proposition F.2** (Bounding squared B-errors by squared Hellinger distance). *Suppose that the B-representation of $\theta$ is $\Lambda_{\mathrm{B}}$-stable (cf. Definition 4) or weakly $\Lambda_{\mathrm{B}}$-stable (cf. Definition D.4), then we have for $h \in [H-1]$*

$$\mathbb{E}_{\bar{\theta},\pi}\Big[ \mathcal{E}_{\theta,h}^{\bar{\theta}}(\tau_{h-1})^2 \Big] \leqslant 4\Lambda_{\mathrm{B}}^2 A U_A \left[ D_{\mathrm{H}}^2\left( \mathbb{P}_{\theta}^{\pi_{h},\mathrm{exp}}, \mathbb{P}_{\bar{\theta}}^{\pi_{h},\mathrm{exp}} \right) + D_{\mathrm{H}}^2\left( \mathbb{P}_{\theta}^{\pi_{h-1},\mathrm{exp}}, \mathbb{P}_{\bar{\theta}}^{\pi_{h-1},\mathrm{exp}} \right) \right],$$

*and*

$$\mathbb{E}_{\bar{\theta},\pi}\Big[ \mathcal{E}_{\theta,H}^{\bar{\theta}}(\tau_{H-1})^2 \Big] \leqslant 2(\Lambda_{\mathrm{B}}+1)^2 D_{\mathrm{H}}^2\left( \mathbb{P}_{\theta}^{\pi_{H-1},\mathrm{exp}}, \mathbb{P}_{\bar{\theta}}^{\pi_{H-1},\mathrm{exp}} \right),$$

$$\left( \mathcal{E}_{\theta,0}^{\bar{\theta}} \right)^2 \leqslant \Lambda_{\mathrm{B}}^2 U_A D_{\mathrm{H}}^2\left( \mathbb{P}_{\theta}^{\pi_{0},\mathrm{exp}}, \mathbb{P}_{\bar{\theta}}^{\pi_{0},\mathrm{exp}} \right),$$

*where $\mathcal{E}_{\theta,h}^{\bar{\theta}}(\tau_{h-1})$ and $\mathcal{E}_{\theta,0}^{\bar{\theta}}$ are as defined in Proposition F.1.*

*Proof of Proposition F.2.* We first deal with the case $h \in [H]$. By taking the difference, we have

$$2\mathcal{E}_{\theta,h}^{\bar{\theta}}(\tau_{h-1}) = \max_{\pi} \sum_{\tau_{h:H}} \pi(\tau_{h:H}) \times \left| \mathbf{B}_{H:h+1}^{\theta}(\tau_{h+1:H}) \left( \mathbf{B}_{h}^{\theta}(o_h, a_h) - \mathbf{B}_{h}^{\bar{\theta}}(o_h, a_h) \right) \mathbf{q}^{\bar{\theta}}(\tau_{h-1}) \right|$$

$$\leqslant \max_{\pi} \sum_{\tau_{h:H}} \pi(\tau_{h:H}) \times \left| \mathbf{B}_{H:h}^{\theta}(\tau_{h:H}) \left( \mathbf{q}^{\theta}(\tau_{h-1}) - \mathbf{q}^{\bar{\theta}}(\tau_{h-1}) \right) \right|$$

$$+ \max_{\pi} \sum_{\tau_{h:H}} \pi(\tau_{h:H}) \times \left| \mathbf{B}_{H:h+1}^{\theta}(\tau_{h+1:H}) \left( \mathbf{B}_{h}^{\theta}(o_h, a_h)\mathbf{q}^{\theta}(\tau_{h-1}) - \mathbf{B}_{h}^{\bar{\theta}}(o_h, a_h)\mathbf{q}^{\bar{\theta}}(\tau_{h-1}) \right) \right|$$

$$= \left\| \mathcal{B}_{H:h}^{\theta}\left( \mathbf{q}^{\theta}(\tau_{h-1}) - \mathbf{q}^{\bar{\theta}}(\tau_{h-1}) \right) \right\|_{\Pi}$$

$$+ \max_{\pi_h} \sum_{o_h, a_h} \pi_h(a_h|o_h) \left\| \mathcal{B}_{H:h+1}^{\theta}\left( \mathbf{B}_{h}^{\theta}(o_h, a_h)\mathbf{q}^{\theta}(\tau_{h-1}) - \mathbf{B}_{h}^{\bar{\theta}}(o_h, a_h)\mathbf{q}^{\bar{\theta}}(\tau_{h-1}) \right) \right\|_{\Pi}.$$

We now introduce several notations for the convenience of the proof.

1. For an action sequence $\mathbf{a}$ of length $l(\mathbf{a})$, $\mathbb{P}(\cdot|\tau_{h-1}, \mathrm{do}(\mathbf{a}))$ stands for the distribution of $o_{h:h+l(\mathbf{a})}$ conditional on $\tau_{h-1}$ and taking action $\mathbf{a}$ for step $h$ to step $h + l(\mathbf{a}) - 1$.

2. Given a set $\mathscr{A}$ of action sequences (possibly of different length), $\mathbb{P}^{\mathrm{Unif}(\mathscr{A})}(\cdot|\tau_{h-1})$ stands for the distribution of observation generated by: conditional on $\tau_{h-1}$, first sample a $\mathbf{a} \sim \mathrm{Unif}(\mathcal{U}_{A,h})$, then take $\mathbf{a}$ and then observe $\mathbf{o}$ (of length $l(\mathbf{a}) + 1$).

By the definition of Hellinger distances and by the notations above, we have

$$D_{\mathrm{H}}^2\left( \mathbb{P}_{\theta}^{\mathrm{Unif}(\mathscr{A})}(\cdot|\tau_{h-1}), \mathbb{P}_{\bar{\theta}}^{\mathrm{Unif}(\mathscr{A})}(\cdot|\tau_{h-1}) \right) = \frac{1}{|\mathscr{A}|} \sum_{\mathbf{a} \in \mathscr{A}} D_{\mathrm{H}}^2\left( \mathbb{P}_{\theta}(\cdot|\tau_{h-1}, \mathrm{do}(\mathbf{a})), \mathbb{P}_{\bar{\theta}}(\cdot|\tau_{h-1}, \mathrm{do}(\mathbf{a})) \right).$$
$$\tag{35}$$

Next, we present two lemmas whose proof will be deferred after the proof of the proposition.

**Lemma F.3.** *Suppose that* $\mathbf{B}$ *is weakly* $\Lambda_{\mathsf{B}}$*-stable* ($\Lambda_{\mathsf{B}}$*-stable is a sufficient condition), then it holds that*

$$\left\|\mathcal{B}_{H:h}^{\theta}\Big(\mathbf{q}^{\theta}(\tau_{h-1}) - \mathbf{q}^{\bar{\theta}}(\tau_{h-1})\Big)\right\|_{\Pi} \leqslant 2\Lambda_{\mathsf{B}}\sqrt{|\mathcal{U}_{A,h}|}D_{\mathsf{H}}\left(\mathbb{P}_{\theta}^{\mathrm{Unif}(\mathcal{U}_{A,h})}(\cdot|\tau_{h-1}), \mathbb{P}_{\bar{\theta}}^{\mathrm{Unif}(\mathcal{U}_{A,h})}(\cdot|\tau_{h-1})\right).$$

**Lemma F.4.** *Suppose that* $\mathbf{B}$ *is weakly* $\Lambda_{\mathsf{B}}$*-stable* ($\Lambda_{\mathsf{B}}$*-stable is a sufficient condition), then it holds that*

$$\max_{\pi_h}\sum_{o_h,a_h}\pi_h(a_h|o_h)\left\|\mathcal{B}_{H:h+1}^{\theta}\Big(\mathbf{B}_h^{\theta}(o_h,a_h)\mathbf{q}^{\theta}(\tau_{h-1}) - \mathbf{B}_h^{\bar{\theta}}(o_h,a_h)\mathbf{q}^{\bar{\theta}}(\tau_{h-1})\Big)\right\|_{\Pi}$$
$$\leqslant 2\Lambda_{\mathsf{B}}\sqrt{A|\mathcal{U}_{A,h+1}|}D_{\mathsf{H}}\left(\mathbb{P}_{\theta}^{\mathrm{Unif}(\mathcal{A})\circ\mathrm{Unif}(\mathcal{U}_{A,h+1})}(\cdot|\tau_{h-1}), \mathbb{P}_{\bar{\theta}}^{\mathrm{Unif}(\mathcal{A})\circ\mathrm{Unif}(\mathcal{U}_{A,h+1})}(\cdot|\tau_{h-1})\right).$$

Therefore, we first consider the case $h \in [H-1]$. Applying Lemma F.3 and taking expectation with respect to $\tau_{h-1}$, we obtain

$$\mathbb{E}_{\bar{\theta},\pi}\left[\left\|\mathcal{B}_{H:h}^{\theta}\Big(\mathbf{q}^{\theta}(\tau_{h-1}) - \mathbf{q}^{\bar{\theta}}(\tau_{h-1})\Big)\right\|_{\Pi}^2\right]$$
$$\leqslant 4\Lambda_{\mathsf{B}}^2|\mathcal{U}_{A,h}|\mathbb{E}_{\bar{\theta},\pi}\left[D_{\mathsf{H}}^2\left(\mathbb{P}_{\theta}^{\mathrm{Unif}(\mathcal{U}_{A,h})}(\cdot|\tau_{h-1}), \mathbb{P}_{\bar{\theta}}^{\mathrm{Unif}(\mathcal{U}_{A,h})}(\cdot|\tau_{h-1})\right)\right] \tag{36}$$
$$\leqslant 8\Lambda_{\mathsf{B}}^2|\mathcal{U}_{A,h}|D_{\mathsf{H}}^2\left(\mathbb{P}_{\theta}^{\pi\circ_h\mathrm{Unif}(\mathcal{U}_{A,h})}, \mathbb{P}_{\bar{\theta}}^{\pi\circ_h\mathrm{Unif}(\mathcal{U}_{A,h})}\right)$$
$$\leqslant 8\Lambda_{\mathsf{B}}^2 A|\mathcal{U}_{A,h}|D_{\mathsf{H}}^2\left(\mathbb{P}_{\theta}^{\pi_{h-1},\exp}, \mathbb{P}_{\bar{\theta}}^{\pi_{h-1},\exp}\right),$$

where the second inequality is due to Lemma C.1, and the last inequality is due to importance sampling. Similarly, applying Lemma F.4 and taking expectation with respect to $\tau_{h-1}$, we have

$$\mathbb{E}_{\bar{\theta},\pi}\left[\left(\max_{\pi_h}\sum_{o_h,a_h}\pi_h(a_h|o_h)\left\|\mathcal{B}_{H:h+1}^{\theta}\Big(\mathbf{B}_h^{\theta}(o_h,a_h)\mathbf{q}^{\theta}(\tau_{h-1}) - \mathbf{B}_h^{\bar{\theta}}(o_h,a_h)\mathbf{q}^{\bar{\theta}}(\tau_{h-1})\Big)\right\|_{\Pi}\right)^2\right]$$
$$\leqslant 8\Lambda_{\mathsf{B}}^2 A|\mathcal{U}_{A,h+1}|D_{\mathsf{H}}^2\left(\mathbb{P}_{\theta}^{\pi\circ_h\mathrm{Unif}(\mathcal{A})\circ\mathrm{Unif}(\mathcal{U}_{A,h})}, \mathbb{P}_{\bar{\theta}}^{\pi\circ_h\mathrm{Unif}(\mathcal{A})\circ\mathrm{Unif}(\mathcal{U}_{A,h})}\right)$$
$$= 8\Lambda_{\mathsf{B}}^2 A|\mathcal{U}_{A,h+1}|D_{\mathsf{H}}^2\left(\mathbb{P}_{\theta}^{\pi_h,\exp}, \mathbb{P}_{\bar{\theta}}^{\pi_h,\exp}\right).$$

The proof for $h \in [H-1]$ is completed by noting that $(x+y)^2 \leqslant 2x^2 + 2y^2$ and $U_A = \max_h |\mathcal{U}_{A,h}|$.

For the case $h = H$, note that by Corollary D.2,

$$\max_{\pi}\sum_{o_H,a_H}\pi(a_H|o_H)\left|\Big(\mathbf{B}_H^{\theta}(o_H,a_H)\mathbf{q}^{\theta}(\tau_{H-1}) - \mathbf{B}_H^{\bar{\theta}}(o_H,a_H)\mathbf{q}^{\bar{\theta}}(\tau_{H-1})\Big)\right|$$
$$= \sum_{o_H}|\mathbb{P}_{\theta}(o_H|\tau_{H-1}) - \mathbb{P}_{\bar{\theta}}(o_H|\tau_{H-1})| \leqslant 2D_{\mathsf{H}}\left(\mathbb{P}_{\theta}(\cdot|\tau_{H-1}), \mathbb{P}_{\bar{\theta}}(\cdot|\tau_{H-1})\right),$$

and by Lemma F.3 it holds that

$$\left\|\mathcal{B}_{H:H}^{\theta}\Big(\mathbf{q}^{\theta}(\tau_{H-1}) - \mathbf{q}^{\bar{\theta}}(\tau_{H-1})\Big)\right\|_{\Pi}$$
$$\leqslant 2\Lambda_{\mathsf{B}}\sqrt{|\mathcal{U}_{A,H}|}D_{\mathsf{H}}\left(\mathbb{P}_{\theta}^{\mathrm{Unif}(\mathcal{U}_{A,H})}(\cdot|\tau_{H-1}), \mathbb{P}_{\bar{\theta}}^{\mathrm{Unif}(\mathcal{U}_{A,H})}(\cdot|\tau_{H-1})\right)$$
$$= 2\Lambda_{\mathsf{B}}D_{\mathsf{H}}\left(\mathbb{P}_{\theta}(\cdot|\tau_{H-1}), \mathbb{P}_{\bar{\theta}}(\cdot|\tau_{H-1})\right),$$

where the equality is due to $\mathcal{U}_{A,H}$ only containing the null action sequence. Therefore,

$$\mathcal{E}_{\theta,H}^{\bar{\theta}}(\tau_{H-1}) \leqslant (\Lambda_{\mathsf{B}}+1)D_{\mathsf{H}}\left(\mathbb{P}_{\theta}^{\pi_{H-1},\exp}, \mathbb{P}_{\bar{\theta}}^{\pi_{H-1},\exp}\right),$$

and applying Lemma C.1 completes the proof of the case $h = H$.

The case $h = 0$ is directly implied by Lemma F.3:

$$\left\|\mathcal{B}_{H:1}^{\theta}\Big(\mathbf{q}_0^{\theta} - \mathbf{q}_0^{\bar{\theta}}\Big)\right\|_{\Pi}^2 \leqslant 4\Lambda_{\mathsf{B}}^2|\mathcal{U}_{A,1}|D_{\mathsf{H}}^2\left(\mathbb{P}_{\theta}^{\pi\circ_1\mathrm{Unif}(\mathcal{U}_{A,1})}, \mathbb{P}_{\bar{\theta}}^{\pi\circ_1\mathrm{Unif}(\mathcal{U}_{A,1})}\right)$$
$$= 4\Lambda_{\mathsf{B}}^2|\mathcal{U}_{A,1}|D_{\mathsf{H}}^2\left(\mathbb{P}_{\theta}^{\pi_0,\exp}, \mathbb{P}_{\bar{\theta}}^{\pi_0,\exp}\right).$$

Combining all these cases finishes the proof of Proposition F.2. $\qquad\square$

We next prove Lemma F.3 and F.4 that were used in the proof of Proposition F.2.

*Proof of Lemma F.3.* By the weak B-stability as in Definition D.4 (B-stability is also sufficient, see Eq. (19)), we have

$$\left\|\mathcal{B}^\theta_{H:h}\Big(\mathbf{q}^\theta(\tau_{h-1})-\mathbf{q}^{\bar\theta}(\tau_{h-1})\Big)\right\|^2_\Pi \leqslant 2\Lambda^2_\mathsf{B}\Big(\left\|\mathbf{q}^\theta(\tau_{h-1})\right\|_\Pi + \left\|\mathbf{q}^{\bar\theta}(\tau_{h-1})\right\|_\Pi\Big)\left\|\sqrt{\mathbf{q}^\theta(\tau_{h-1})}-\sqrt{\mathbf{q}^{\bar\theta}(\tau_{h-1})}\right\|^2_2,$$

where $\|\cdot\|_\Pi$ is defined in Definition D.3. By the definition of $\mathbf{q}^\theta(\tau_{h-1})$, for $t_h = (\mathbf{o}, \mathbf{a}) \in \mathcal{U}_h$, we have

$$\mathbf{q}^\theta(\tau_{h-1})(\mathbf{o}, \mathbf{a}) = \mathbb{P}_\theta(t_h|\tau_{h-1}) = \mathbb{P}_\theta(o_{h:h+l(\mathbf{a})-1} = \mathbf{o}|\tau_{h-1}, \mathrm{do}(\mathbf{a})).$$

Hence, we have

$$\left\|\mathbf{q}^\theta(\tau_{h-1})\right\|_\Pi = \max_{T'\subset\mathcal{U}_h}\max_\pi \sum_{(\mathbf{o},\mathbf{a})\in\mathcal{U}_h}\pi(\mathbf{o},\mathbf{a}) \times \mathbb{P}_\theta(o_{h:h+l(\mathbf{a})-1} = \mathbf{o}|\tau_{h-1}, \mathrm{do}(\mathbf{a}))$$

$$= \max_{T'\subset\mathcal{U}_h}\max_\pi \mathbb{P}^\pi_\theta(T'|\tau_{h-1}) \leqslant 1,$$

where $\mathbb{P}^\pi_\theta(T'|\tau_{h-1})$ stands for the probability that some test in $T'$ is observed under $\bar\theta, \pi$ conditional on $\tau_{h-1}$. Similarly, we have $\left\|\mathbf{q}^{\bar\theta}(\tau_{h-1})\right\|_\Pi \leqslant 1$. Therefore, we have

$$\frac{1}{4}\Lambda^{-2}_\mathsf{B}\left\|\mathcal{B}^\theta_{H:h}\Big(\mathbf{q}^\theta(\tau_{h-1})-\mathbf{q}^{\bar\theta}(\tau_{h-1})\Big)\right\|^2_\Pi \leqslant \left\|\sqrt{\mathbf{q}^\theta(\tau_{h-1})}-\sqrt{\mathbf{q}^{\bar\theta}(\tau_{h-1})}\right\|^2_2$$

$$= \sum_{\mathbf{a}\in\mathcal{U}_{A,h}}\sum_{\mathbf{o}:(\mathbf{o},\mathbf{a})\in\mathcal{U}_h}\left|[\sqrt{\mathbb{P}_\theta} - \sqrt{\mathbb{P}_{\bar\theta}}](\mathbf{o}|\tau_{h-1}, \mathrm{do}(\mathbf{a}))\right|^2$$

$$\overset{(i)}{\leqslant} \sum_{\mathbf{a}\in\mathcal{U}_{A,h}}\sum_{\mathbf{o}\in\mathcal{O}^{l(\mathbf{a})+1}}\left|[\sqrt{\mathbb{P}_\theta} - \sqrt{\mathbb{P}_{\bar\theta}}](\mathbf{o}|\tau_{h-1}, \mathrm{do}(\mathbf{a}))\right|^2$$

$$\overset{(ii)}{=} \sum_{\mathbf{a}\in\mathcal{U}_{A,h}}D^2_\mathrm{H}\left(\mathbb{P}_\theta(\cdot|\tau_{h-1}, \mathrm{do}(\mathbf{a})), \mathbb{P}_{\bar\theta}(\cdot|\tau_{h-1}, \mathrm{do}(\mathbf{a}))\right)$$

$$\overset{(iii)}{=} |\mathcal{U}_{A,h}|\, D^2_\mathrm{H}\left(\mathbb{P}^{\mathrm{Unif}(\mathcal{U}_{A,h})}_\theta(\cdot|\tau_{h-1}), \mathbb{P}^{\mathrm{Unif}(\mathcal{U}_{A,h})}_{\bar\theta}(\cdot|\tau_{h-1})\right),$$

where in (i) we include those $\mathbf{o}$ such that $(\mathbf{o}, \mathbf{a})$ may not belong to $\mathcal{U}_{h+1}$ into summation, (ii) is due to the definition of $\mathbb{P}(\cdot|\tau_{h-1}, \mathrm{do}(\mathbf{a}))$, and (iii) follows from importance sampling (35). This completes the proof of Lemma F.3. $\square$

*Proof of Lemma F.4.* Similar to the proof of Lemma F.3, we only need to work under the weak B-stability condition. By Corollary D.2, for $t_{h+1} = (\mathbf{o}, \mathbf{a}) \in \mathcal{U}_{h+1}$, it holds that

$$\left[\mathbf{B}^\theta_h(o, a)\mathbf{q}^\theta(\tau_{h-1})\right](\mathbf{o}, \mathbf{a}) = \mathbb{P}_\theta(t_{h+1}|\tau_{h-1}, o, a) \times \mathbb{P}_\theta(o|\tau_{h-1}) = \mathbb{P}_\theta(o, a, t_{h+1}|\tau_{h-1}),$$

and hence

$$\left\|\mathbf{B}^\theta_h(o, a)\mathbf{q}^\theta(\tau_{h-1})\right\|_\Pi = \max_{T'\subset\mathcal{U}_{h+1}}\max_\pi \sum_{t_{h+1}\in T'}\pi(t_{h+1}) \times \mathbb{P}_\theta(t_{h+1}|\tau_{h-1}, o, a) \times \mathbb{P}_\theta(o|\tau_{h-1})$$

$$= \max_{T'\subset\mathcal{U}_{h+1}}\max_\pi \mathbb{P}^\pi_\theta(T'|\tau_{h-1}, o, a) \times \mathbb{P}_\theta(o|\tau_{h-1}) \leqslant \mathbb{P}_\theta(o|\tau_{h-1}),$$

where $\mathbb{P}^\pi_\theta(T'|\tau_{h-1}, o, a)$ stands for the probability that some test in $T'$ is observed under $\theta, \pi$ conditional on observing $\tau_h = (\tau_{h-1}, o, a)$. Similarly, we have $\left\|\mathbf{B}^{\bar\theta}_h(o, a)\mathbf{q}^{\bar\theta}(\tau_{h-1})\right\|_\Pi \leqslant \mathbb{P}_{\bar\theta}(o|\tau_{h-1})$. Therefore, by the weak B-stability as in Definition D.4 and combining with the inequalities above, it holds that

$$\left\|\mathcal{B}^\theta_{H:h+1}\Big(\mathbf{B}^\theta_h(o, a)\mathbf{q}^\theta(\tau_{h-1}) - \mathbf{B}^{\bar\theta}_h(o, a)\mathbf{q}^{\bar\theta}(\tau_{h-1})\Big)\right\|_\Pi$$

$$\leqslant \Lambda_\mathsf{B}\sqrt{2[\mathbb{P}_\theta + \mathbb{P}_{\bar\theta}](o_h = o|\tau_{h-1})} \cdot \Big[\sum_{t\in\mathcal{U}_{h+1}}\left|\big[\sqrt{\mathbb{P}_\theta} - \sqrt{\mathbb{P}_{\bar\theta}}\big](o, a, t|\tau_{h-1})\right|^2\Big]^{1/2}.$$

Hence, we have

$$
\Lambda_{\mathsf{B}}^{-1} \max_{\pi_h} \sum_{o_h, a_h} \pi_h(a_h|o_h) \left\| \mathcal{B}_{H:h+1}^{\theta} \Big( \mathbf{B}_h^{\theta}(o_h, a_h) \mathbf{q}^{\theta}(\tau_{h-1}) - \mathbf{B}_h^{\bar{\theta}}(o_h, a_h) \mathbf{q}^{\bar{\theta}}(\tau_{h-1}) \Big) \right\|_{\Pi}
$$

$$
\leqslant \max_{\pi} \sum_{o,a} \pi(a|o) \sqrt{2[\mathbb{P}_{\theta} + \mathbb{P}_{\bar{\theta}}](o_h = o|\tau_{h-1})} \Big[ \sum_{t \in \mathcal{U}_{h+1}} \left| \left[ \sqrt{\mathbb{P}_{\theta}} - \sqrt{\mathbb{P}_{\bar{\theta}}} \right](o, a, t|\tau_{h-1}) \right|^2 \Big]^{1/2}
$$

$$
\overset{(i)}{\leqslant} \sum_{o} \sqrt{2[\mathbb{P}_{\theta} + \mathbb{P}_{\bar{\theta}}](o_h = o|\tau_{h-1})} \Big[ \sum_{a} \sum_{t \in \mathcal{U}_{h+1}} \left| \left[ \sqrt{\mathbb{P}_{\theta}} - \sqrt{\mathbb{P}_{\bar{\theta}}} \right](o, a, t|\tau_{h-1}) \right|^2 \Big]^{1/2}
$$

$$
\overset{(ii)}{\leqslant} 2 \Big[ \sum_{o,a} \sum_{t \in \mathcal{U}_{h+1}} \left| \left[ \sqrt{\mathbb{P}_{\theta}} - \sqrt{\mathbb{P}_{\bar{\theta}}} \right](o, a, t|\tau_{h-1}) \right|^2 \Big]^{1/2}
$$

$$
= 2 \Big[ \sum_{o,a} \sum_{(\mathbf{o},\mathbf{a}) \in \mathcal{U}_{h+1}} \left| [\sqrt{\mathbb{P}_{\theta}} - \sqrt{\mathbb{P}_{\bar{\theta}}}](o_{h:h+l(\mathbf{a})+1} = (o, \mathbf{o})|\tau_{h-1}, \mathrm{do}(a, \mathbf{a})) \right|^2 \Big]^{1/2}
$$

$$
\overset{(iii)}{\leqslant} 2 \Big[ \sum_{(a,\mathbf{a}) \in \mathcal{A} \times \mathcal{U}_{A,h+1}} \sum_{o,\mathbf{o}} \left| [\sqrt{\mathbb{P}_{\theta}} - \sqrt{\mathbb{P}_{\bar{\theta}}}](o_{h:h+l(\mathbf{a})+1} = (o, \mathbf{o})|\tau_{h-1}, \mathrm{do}(a, \mathbf{a})) \right|^2 \Big]^{1/2}
$$

$$
\overset{(iv)}{\leqslant} 2\sqrt{A |\mathcal{U}_{A,h+1}|} D_{\mathrm{H}} \left( \mathbb{P}_{\theta}^{\mathrm{Unif}(\mathcal{A}) \circ \mathrm{Unif}(\mathcal{U}_{A,h+1})}(\cdot|\tau_{h-1}), \mathbb{P}_{\bar{\theta}}^{\mathrm{Unif}(\mathcal{A}) \circ \mathrm{Unif}(\mathcal{U}_{A,h+1})}(\cdot|\tau_{h-1}) \right),
$$

where (i) is due to the fact that $\max_{\pi \in \Delta(\mathcal{A})} \sum_{a \in \mathcal{A}} \pi(a)x(a) \leqslant \left( \sum_a x(a)^2 \right)^{1/2}$, (ii) is due to Cauchy-Schwarz inequality, in (iii) we include those $\mathbf{o}$ such that $(\mathbf{o}, \mathbf{a})$ may not belong to $\mathcal{U}_{h+1}$ into summation, (iv) is due to (35): $\mathrm{Unif}(\mathcal{A}) \circ \mathrm{Unif}(\mathcal{U}_{A,h+1})$ is simply the uniform policy over $\mathcal{A} \times \mathcal{U}_{A,h+1}$. This concludes the proof of Lemma F.4. $\qquad\square$

## G  PROOF OF THEOREM 9

We first restate Theorem 9 as follows in terms of the (more relaxed) weak B-stability condition.

**Theorem G.1** (Restatement of Theorem 9). *Suppose every $\theta \in \Theta$ is $\Lambda_{\mathsf{B}}$-stable (Definition 4) or weakly $\Lambda_{\mathsf{B}}$-stable (Definition D.4), and the true model $\theta^{\star} \in \Theta$ with rank $d_{\mathsf{PSR}} \leqslant d$. Then, choosing $\beta = C \log(\mathcal{N}_{\Theta}(1/KH))/\delta)$ for some absolute constant $C > 0$, with probability at least $1 - \delta$, Algorithm 1 outputs a policy $\widehat{\pi}_{\mathrm{out}} \in \Delta(\Pi)$ such that $V_{\star} - V_{\theta^{\star}}(\widehat{\pi}_{\mathrm{out}}) \leqslant \varepsilon$, as long as the number of episodes*

$$
T = KH \geqslant \mathcal{O}\Big( dAU_A \Lambda_{\mathsf{B}}^2 H^2 \log(\mathcal{N}_{\Theta}(1/T)/\delta)\iota/\varepsilon^2 \Big), \tag{37}
$$

*where $\iota := \log\left(1 + KdU_A \Lambda_{\mathsf{B}} R_{\mathsf{B}}\right)$ with $R_{\mathsf{B}} := \max_h \{1, \max_{\|v\|_1 = 1} \sum_{o,a} \|\mathbf{B}_h(o, a)v\|_1\}$.*

The proof of Theorem G.1 uses the following fast rate guarantee for the OMLE algorithm, which is standard (e.g. Van de Geer (2000); Agarwal et al. (2020)). For completeness, we present its proof in Appendix G.1.

**Proposition G.2** (Guarantee of MLE). *Suppose that we choose $\beta \geqslant 2 \log \mathcal{N}_{\Theta}(1/T) + 2\log(1/\delta) + 2$ in Algorithm 1. Then with probability at least $1 - \delta$, the following holds:*

*(a) For all $k \in [K]$, $\theta^{\star} \in \Theta^k$;*

*(b) For all $k \in [K]$ and any $\theta \in \Theta^k$, it holds that*

$$
\sum_{t=1}^{k-1} \sum_{h=0}^{H-1} D_{\mathrm{H}}^2 \left( \mathbb{P}_{\theta}^{\pi_{h,\mathrm{exp}}^t}, \mathbb{P}_{\theta^{\star}}^{\pi_{h,\mathrm{exp}}^t} \right) \leqslant 2\beta.
$$

We next prove Theorem G.1. We adopt the definitions of $\mathcal{E}_{\theta,h}^{\bar{\theta}}(\tau_{h-1})$ as in Proposition F.1 and abbreviate $\mathcal{E}_{k,h}^{\star} = \mathcal{E}_{\theta^k,h}^{\theta^{\star}}$. We also condition on the success of the event in Proposition G.2.

**Step 1.** By Proposition G.2, it holds that $\theta^\star \in \Theta$. Therefore, $V_{\theta^k}(\pi^k) \geqslant V_\star$, and by Proposition F.1, we have

$$
\sum_{t=1}^k \left( V_\star - V_{\theta^\star}(\pi^t) \right) \leqslant \sum_{t=1}^k \left( V_{\theta^t}(\pi^t) - V_{\theta^\star}(\pi^t) \right) \leqslant \sum_{t=1}^k D_{\mathrm{TV}} \left( \mathbb{P}_{\theta^t}^{\pi^t}, \mathbb{P}_{\theta^\star}^{\pi^t} \right)
$$

$$
\leqslant \sum_{t=1}^k 1 \wedge \left( \mathcal{E}_{t,0}^\star + \sum_{h=1}^H \mathbb{E}_{\pi^t} \left[ \mathcal{E}_{t,h}^\star(\tau_{h-1}) \right] \right) \tag{38}
$$

$$
\leqslant \sum_{t=1}^k \left( 1 \wedge \mathcal{E}_{t,0}^\star + \sum_{h=1}^H 1 \wedge \mathbb{E}_{\pi^t} \left[ \mathcal{E}_{t,h}^\star(\tau_{h-1}) \right] \right).
$$

On the other hand, by Proposition F.2, we have

$$
(\mathcal{E}_{t,0}^\star)^2 + \sum_{h=1}^H \mathbb{E}_{\pi^t} \left[ \mathcal{E}_{k,h}^\star(\tau_{h-1})^2 \right] \leqslant 12 \Lambda_{\mathsf{B}}^2 A U_A \sum_{h=0}^{H-1} D_{\mathrm{H}}^2 \left( \mathbb{P}_{\theta^k}^{\pi_{h,\exp}}, \mathbb{P}_{\theta^\star}^{\pi_{h,\exp}} \right).
$$

Furthermore, by Proposition G.2 we have

$$
\sum_{t=1}^{k-1} \sum_{h=0}^{H-1} D_{\mathrm{H}}^2 \left( \mathbb{P}_{\theta^k}^{\pi_{h,\exp}^t}, \mathbb{P}_{\theta^\star}^{\pi_{h,\exp}^t} \right) \leqslant 2\beta.
$$

Therefore, defining $\beta_{k,h} := \sum_{t<k} \mathbb{E}_{\pi^t} [ \mathcal{E}_{k,h}^\star(\tau_{h-1})^2 ]$, combining the two equations above gives

$$
\sum_{h=0}^H \beta_{k,h} = \sum_{h=0}^H \sum_{t<k} \mathbb{E}_{\pi^t} [ \mathcal{E}_{k,h}^\star(\tau_{h-1})^2 ] \leqslant 24 \Lambda_{\mathsf{B}}^2 A U_A \beta, \qquad \forall k \in [K]. \tag{39}
$$

**Step 2.** We would like to bridge the performance decomposition (38) and the squared B-errors bound (39) using the generalized $\ell_2$-Eluder argument. We consider separately the case for $h = 0$ and $h \in [H]$.

**Case 1:** $h = 0$. This case follows directly from Cauchy-Schwarz inequality:

$$
\sum_{t=1}^k 1 \wedge \mathcal{E}_{t,0}^\star \leqslant \left( k \sum_{t=1}^k 1 \wedge \left( \mathcal{E}_{t,0}^\star \right)^2 \right)^{1/2} \leqslant \sqrt{k(\beta_{k,0} + 1)}. \tag{40}
$$

**Case 2:** $h \in [H]$. We invoke the generalized $\ell_2$-Eluder argument (actually, its corollary) as in Appendix E.1, restated as follows for convenience.

**Corollary E.2.** *Suppose we have a sequence of functions* $\{ f_k : \mathbb{R}^n \to \mathbb{R} \}_{k \in [K]}$*:*

$$
f_k(x) := \max_{r \in \mathcal{R}} \sum_{j=1}^J |\langle x, y_{k,j,r} \rangle|,
$$

*which is given by the family of vectors* $\{ y_{k,j,r} \}_{(k,j,r) \in [K] \times [J] \times \mathcal{R}} \subset \mathbb{R}^n$*. Further assume that there exists* $L > 0$ *such that* $f_k(x) \leqslant L \|x\|_1$*.*

*Consider further a sequence of vector* $(x_i)_{i \in \mathcal{I}}$*, satisfying the following condition*

$$
\sum_{t=1}^{k-1} \mathbb{E}_{i \sim q_t} [ f_k^2(x_i) ] \leqslant \beta_k, \qquad \forall k \in [K],
$$

*and the subspace spanned by* $(x_i)_{i \in \mathcal{I}}$ *has dimension at most* $d$*. Then it holds that*

$$
\sum_{t=1}^k 1 \wedge \mathbb{E}_{i \sim q_t} [ f_t(x_i) ] \leqslant \sqrt{ 4d \left( k + \sum_{t=1}^k \beta_t \right) \log \left( 1 + kdL \max_i \|x_i\|_1 \right) }, \qquad \forall k \in [K].
$$

We have the following three preparation steps to apply Corollary E.2.

1. Recall the definition of $\mathcal{E}^\star_{t,h}(\tau_{h-1})$ as in Proposition F.1 (in short $\mathcal{E}^\star_{k,h}(\tau_{h-1}) := \mathcal{E}^{\theta^\star}_{\theta^k,h}(\tau_{h-1})$),

$$\mathcal{E}^\star_{k,h}(\tau_{h-1}) := \frac{1}{2} \max_\pi \sum_{\tau_{h:H}} \pi(\tau_{h:H}) \times \left| \mathbf{B}^k_{H:h+1}(\tau_{h+1:H}) \left( \mathbf{B}^k_h(o_h, a_h) - \mathbf{B}^\star_h(o_h, a_h) \right) \mathbf{q}^\star(\tau_{h-1}) \right|,$$

where we replace superscript $\theta^k$ of $\mathbf{B}$ by $k$ for simplicity. Let us define

$$y_{k,j,\pi} := \frac{1}{2} \pi(\tau^j_{h:H}) \times \left[ \mathbf{B}^k_{H:h+1}(\tau^j_{h+1:H}) \left( \mathbf{B}^k_h(o^j_h, a^j_h) - \mathbf{B}^\star_h(o^j_h, a^j_h) \right) \right]^\top \in \mathbb{R}^{|\mathcal{U}_h|},$$

where $\{\tau^j_{h:H} = (o_h, a_h, \cdots, o_H, a_H)\}^n_{j=1}$ is an ordering of all possible $\tau_{h:H}$ (and hence $n = (OA)^{H-h+1}$), $\pi$ is any policy that starts at step $h$. We then define

$$f_k(x) = \max_\pi \sum_j |\langle y_{k,j,\pi}, x \rangle|, \qquad x \in \mathbb{R}^{\mathcal{U}_h}.$$

It follows from definition that $\mathcal{E}^\star_{k,h}(\tau_{h-1}) = f_k(\mathbf{q}^\star(\tau_{h-1}))$.

2. We define $x_i = \mathbf{q}^\star(\tau^i_{h-1}) \in \mathbb{R}^{|\mathcal{U}_h|}$, where $\{\tau^i_{h-1}\}_i$ is an ordering of all possible $\tau_{h-1} \in (\mathcal{O} \times \mathcal{A})^{h-1}$. Then by the assumption that $\theta^\star$ has PSR rank less than or equal to $d$, we have $\dim \text{span}(x_i : i \in \mathcal{I}) \leqslant d$. Furthermore, we have $\|x_i\|_1 \leqslant U_A$ by definition.

3. It remains to verify that $f_k$ is Lipschitz with respect to 1-norm. We only need to verify it under the weak $\Lambda_B$-stability condition. We have

$$f_k(\mathbf{q}) \leqslant \frac{1}{2} \left[ \left\| \mathcal{B}^k_{H:h} \mathbf{q} \right\|_\Pi + \max_\pi \sum_{o,a} \pi(a|o) \left\| \mathcal{B}^k_{H:h+1} \mathbf{B}^\star_h(o, a) \mathbf{q} \right\|_\Pi \right]$$

$$\leqslant 2 \Lambda_B \|\mathbf{q}\|_1 + 2 \Lambda_B \max_\pi \sum_{o,a} \pi(a|o) \|\mathbf{B}^\star_h(o, a) \mathbf{q}\|_1$$

$$\leqslant 2 \Lambda_B \|\mathbf{q}\|_1 + 2 \Lambda_B \sum_{o,a} \|\mathbf{B}^\star_h(o, a)\|_1 \|\mathbf{q}\|_1 \leqslant 2 \Lambda_B (R_B + 1) \|\mathbf{q}\|_1,$$

where the first inequality follows the same argument as (35); the second inequality is due to B-stability (or weak B-stability and (21)); the last inequality is due to the definition of $B$. Hence we can take $L = 2 \Lambda_B (R_B + 1)$ to ensure that $f_k(x) \leqslant L \|x\|_1$.

Therefore, applying Corollary E.2 yields

$$\sum_{t=1}^k 1 \wedge \mathbb{E}_{\pi^t} \left[ \mathcal{E}^\star_{t,h}(\tau_{h-1}) \right] \leqslant \sqrt{4\iota \left( kd + d \sum_{t=1}^k \beta_{t,h} \right)}, \tag{41}$$

where $\iota = \log \left( 1 + 2kd U_A \Lambda_B (R_B + 1) \right)$. This completes case 2.

Combining these two cases, we obtain

$$\sum_{t=1}^k \left( V_\star - V_{\theta^\star}(\pi^t) \right) \overset{(i)}{\leqslant} \sum_{t=1}^k 1 \wedge \mathcal{E}^\star_{t,0} + \sum_{h=1}^H \left( \sum_{t=1}^k 1 \wedge \mathbb{E}_{\pi^t} \left[ \mathcal{E}^\star_{t,h}(\tau_{h-1}) \right] \right)$$

$$\overset{(ii)}{\leqslant} \sqrt{k(\beta_{k,0} + 1)} + 2\sqrt{\iota} \cdot \sum_{h=1}^H \left( kd + d \sum_{t=1}^k \beta_{t,h} \right)^{1/2}$$

$$\leqslant \sqrt{(4H\iota + 1) \cdot \left( k(Hd + 1) + d \sum_{t=1}^k \sum_{h=0}^H \beta_{t,h} \right)}$$

$$\overset{(iii)}{=} \mathcal{O} \left( \sqrt{\Lambda_B^2 d A U_A H \cdot k\beta\iota} \right).$$

where (i) used (38); (ii) used the above two cases (40) and (41); (iii) used (39). As a consequence, whenever $k \geqslant \mathcal{O}(\Lambda_B^2 d A U_A H \cdot \beta\iota / \varepsilon^2)$, we have $\frac{1}{k} \sum_{t=1}^k \left( V_\star - V_{\theta^\star}(\pi^t) \right) \leqslant \varepsilon$. This completes the proof of Theorem G.1 (and hence Theorem 9). $\qquad \square$

## G.1 PROOF OF PROPOSITION G.2

For the simplicity of presentation, we consider the following general interaction process: For $t = 1, \cdots, T$, the learner determines a $\bar{\pi}^t$, then executes $\bar{\pi}^t$ and collects a trajectory $\bar{\tau}^t \sim \mathbb{P}_{\theta^\star}^{\bar{\pi}^t}(\cdot)$. We show that, with probability at least $1 - \delta$, the following holds for all $t \in [T]$ and $\theta \in \Theta$:

$$\sum_{s=1}^{t-1} D_{\mathrm{H}}^2 \left( \mathbb{P}_\theta^{\bar{\pi}^s}, \mathbb{P}_{\theta^\star}^{\bar{\pi}^s} \right) \leqslant \mathcal{L}_t(\theta^\star) - \mathcal{L}_t(\theta) + 2 \log \mathcal{N}_\Theta(1/T) + 2 \log(1/\delta) + 2, \qquad (42)$$

where $\mathcal{L}_t$ is the total log-likelihood (at step $t$) defined as

$$\mathcal{L}_t(\theta) := \sum_{s=1}^{t-1} \log \mathbb{P}_\theta^{\bar{\pi}^s}(\bar{\tau}^s).$$

Proposition G.2 is implied by (42) directly: Suppose we choose $\beta \geqslant 2 \log \mathcal{N}_\Theta(1/T) + 2 \log(1/\delta) + 2$. On the event (42), by the non-negativity of squared Hellinger distances, we have for all $k \in [K]$ and $\theta \in \Theta$ that

$$\sum_{(\pi, \tau) \in \mathcal{D}^k} \log \mathbb{P}_{\theta^\star}^\pi(\tau) \geqslant \sum_{(\pi, \tau) \in \mathcal{D}^k} \log \mathbb{P}_\theta^\pi(\tau) - \beta,$$

where $\mathcal{D}^k$ is the dataset of all histories before the outer loop of Algorithm 1 enters step $k$. Taking max over $\theta \in \Theta$ on the right-hand side, we obtain $\theta^\star \in \Theta^k$, which gives Proposition G.2(1). Furthermore, for $k \in [K]$ and $\theta \in \Theta^k$, (42) implies that

$$\sum_{t=1}^{k-1} \sum_{h=0}^{H-1} D_{\mathrm{H}}^2 \left( \mathbb{P}_\theta^{\pi_{h,\exp}^t}, \mathbb{P}_{\theta^\star}^{\pi_{h,\exp}^t} \right) \leqslant \sum_{(\pi,\tau) \in \mathcal{D}^k} \log \mathbb{P}_{\theta^\star}^\pi(\tau) - \sum_{(\pi,\tau) \in \mathcal{D}^k} \log \mathbb{P}_\theta^\pi(\tau) + \beta$$

$$\leqslant \max_{\hat{\theta}} \sum_{(\pi,\tau) \in \mathcal{D}^k} \log \mathbb{P}_{\hat{\theta}}^\pi(\tau) - \sum_{(\pi,\tau) \in \mathcal{D}^k} \log \mathbb{P}_\theta^\pi(\tau) + \beta \leqslant 2\beta,$$

which gives Proposition G.2(2).

In the following, we establish (42). Let us fix a $1/T$-optimistic covering $(\widetilde{\mathbb{P}}, \Theta_0)$ of $\Theta$, such that $n := |\Theta_0| = \mathcal{N}_\Theta(1/T)$. We label $(\widetilde{\mathbb{P}}_{\theta_0})_{\theta_0 \in \Theta_0}$ by $\widetilde{\mathbb{P}}_1, \cdots, \widetilde{\mathbb{P}}_n$. By the definition of optimistic covering, it is clear that for any $\theta \in \Theta$, there exists $i \in [n]$ such that for all $\pi, \tau$, it holds that $\widetilde{\mathbb{P}}_i^\pi(\tau) \geqslant \mathbb{P}_\theta^\pi(\tau)$ and $\|\widetilde{\mathbb{P}}_i^\pi(\cdot) - \mathbb{P}_\theta^\pi(\cdot)\|_1 \leqslant 1/T^2$. We say $\theta$ is covered by this $i \in [n]$.

Then, we consider

$$\ell_i^t = \log \frac{\mathbb{P}_{\theta^\star}^{\bar{\pi}^t}(\bar{\tau}^t)}{\widetilde{\mathbb{P}}_i^{\bar{\pi}^t}(\bar{\tau}^t)}, \qquad t \in [T], \ i \in [n].$$

By Lemma C.3, the following holds with probability at least $1 - \delta$: for all $t \in [T]$, $i \in [n]$,

$$\frac{1}{2} \sum_{s=1}^{t-1} \ell_i^s + \log(n/\delta) \geqslant \sum_{s=1}^{t-1} -\mathbb{E}_s \left[ \exp \left( -\frac{1}{2} \ell_i^s \right) \right],$$

where $\mathbb{E}_s$ denotes the conditional expectation over all randomness after $\bar{\pi}^s$ has been determined. By definition,

$$\mathbb{E}_t \left[ \exp \left( -\frac{1}{2} \ell_i^t \right) \right] = \mathbb{E}_t \left[ \sqrt{\frac{\widetilde{\mathbb{P}}_i^{\bar{\pi}^t}(\bar{\tau}^t)}{\mathbb{P}_{\theta^\star}^{\bar{\pi}^t}(\bar{\tau}^t)}} \right] = \mathbb{E}_{\tau \sim \bar{\pi}^t} \left[ \sqrt{\frac{\widetilde{\mathbb{P}}_i^{\bar{\pi}^t}(\tau)}{\mathbb{P}_{\theta^\star}^{\bar{\pi}^t}(\tau)}} \right] = \sum_\tau \sqrt{\mathbb{P}_{\theta^\star}^{\bar{\pi}^t}(\tau) \widetilde{\mathbb{P}}_i^{\bar{\pi}^t}(\tau)}$$

Therefore, for any $\theta \in \Theta$ that is covered by $i \in [n]$, we have

$$-\log \mathbb{E}_t \left[ \exp \left( -\frac{1}{2} \ell_i^t \right) \right] \geqslant 1 - \sum_\tau \sqrt{\mathbb{P}_{\theta^\star}^{\bar{\pi}^t}(\tau) \widetilde{\mathbb{P}}_i^{\bar{\pi}^t}(\tau)}$$

$$= 1 - \sum_\tau \sqrt{\mathbb{P}_{\theta^\star}^{\bar{\pi}^t}(\tau) \mathbb{P}_\theta^{\bar{\pi}^t}(\tau)} - \sum_\tau \sqrt{\mathbb{P}_{\theta^\star}^{\bar{\pi}^t}(\tau)} \left( \sqrt{\widetilde{\mathbb{P}}_i^{\bar{\pi}^t}(\tau)} - \sqrt{\mathbb{P}_\theta^{\bar{\pi}^t}(\tau)} \right)$$

---

**Algorithm 2** EXPLORATIVE E2D (Chen et al., 2022)

---

**Input:** Model class $\Theta$, parameters $\gamma > 0$, $\eta \in (0, 1/2)$. An $1/T$-optimistic cover $(\widetilde{\mathbb{P}}, \Theta_0)$.
 1: Initialize $\mu^1 = \mathrm{Unif}(\Theta_0)$.
 2: **for** $t = 1, \ldots, T$ **do**
 3:     Set $(p_{\exp}^t, p_{\mathrm{out}}^t) = \arg\min_{(p_{\exp}, p_{\mathrm{out}}) \in \Delta(\Pi)^2} \widehat{V}_\gamma^{\mu^t}(p_{\exp}, p_{\mathrm{out}})$, where $\widehat{V}_\gamma^{\mu^t}$ is defined by

$$\widehat{V}_\gamma^{\mu^t}(p_{\exp}, p_{\mathrm{out}}) := \sup_{\theta \in \Theta} \mathbb{E}_{\pi \sim p_{\mathrm{out}}}[V_\theta(\pi_\theta) - V_\theta(\pi)] - \gamma \mathbb{E}_{\pi \sim p_{\exp}} \mathbb{E}_{\theta^t \sim \mu^t}\big[D_{\mathrm{H}}^2(\mathbb{P}_\theta^\pi, \mathbb{P}_{\theta^t}^\pi)\big].$$

 4:     Sample $\pi^t \sim p_{\exp}^t$. Execute $\pi^t$ and observe $\tau^t$.
 5:     Compute $\mu^{t+1} \in \Delta(\Theta_0)$ by

$$\mu^{t+1}(\theta) \propto_\theta \mu^t(\theta) \cdot \exp\left(\eta \log \widetilde{\mathbb{P}}_\theta^{\pi^t}(\tau^t)\right).$$

**Output:** Policy $\widehat{\pi}_{\mathrm{out}} := \frac{1}{T} \sum_{t=1}^T p_{\mathrm{out}}^t$.

---

$$\geqslant \frac{1}{2} D_{\mathrm{H}}^2\left(\mathbb{P}_\theta^{\bar{\pi}^t}(\tau = \cdot), \mathbb{P}_{\theta^\star}^{\bar{\pi}^t}(\tau = \cdot)\right) - \left(\sum_\tau \left|\sqrt{\widetilde{\mathbb{P}}_i^{\bar{\pi}^t}(\tau)} - \sqrt{\mathbb{P}_\theta^{\bar{\pi}^t}(\tau)}\right|^2\right)^{1/2}$$

$$\geqslant \frac{1}{2} D_{\mathrm{H}}^2\left(\mathbb{P}_\theta^{\bar{\pi}^t}(\tau = \cdot), \mathbb{P}_{\theta^\star}^{\bar{\pi}^t}(\tau = \cdot)\right) - \left\|\widetilde{\mathbb{P}}_i^{\bar{\pi}^t}(\cdot) - \mathbb{P}_\theta^{\bar{\pi}^t}(\cdot)\right\|_1^{1/2}$$

$$\geqslant \frac{1}{2} D_{\mathrm{H}}^2\left(\mathbb{P}_\theta^{\bar{\pi}^t}(\tau = \cdot), \mathbb{P}_{\theta^\star}^{\bar{\pi}^t}(\tau = \cdot)\right) - \frac{1}{T},$$

where the first inequality is due to $-\log x \geqslant 1 - x$; in the second inequality we use the definition of Hellinger distance and Cauchy inequality; the third inequality is because $(\sqrt{x} - \sqrt{y})^2 \leqslant |x - y|$ for all $x, y \in \mathbb{R}_{\geqslant 0}$; the last inequality is due to our assumption that $\theta$ is covered by $i$. Notice that every $\theta \in \Theta$ is covered by some $i \in [n]$, and for such $i$, $\sum_{s=1}^{t-1} \ell_i^s \leqslant \mathcal{L}_t(\theta^\star) - \mathcal{L}_t(\theta)$; therefore, it holds with probability $1 - \delta$ that, for all $\theta \in \Theta$, $t \in [T]$,

$$\frac{1}{2}(\mathcal{L}_t(\theta^\star) - \mathcal{L}_t(\theta)) + \log(n/\delta) + \frac{t-1}{T} \geqslant \frac{1}{2} \sum_{s=1}^{t-1} D_{\mathrm{H}}^2\left(\mathbb{P}_\theta^{\bar{\pi}^s}, \mathbb{P}_{\theta^\star}^{\bar{\pi}^s}\right).$$

Plugging in $n = \mathcal{N}_\Theta(1/T)$ and scaling the above inequality by 2 gives (42).                    □

## H  EXPLORATIVE E2D, ALL-POLICY MODEL-ESTIMATION E2D, AND MOPS

In this section, we present the detailed algorithms of EXPLORATIVE E2D, ALL-POLICY MODEL-ESTIMATION E2D, and MOPS introduced in Section 4. We also state the theorems for their sample complexity bounds of learning $\varepsilon$-optimal policy of B-stable PSRs.

### H.1  EXPLORATIVE E2D ALGORITHM

In this section, we provide more details about the EXPLORATIVE E2D algorithm as discussed in Section 4.2. The full algorithm of EXPLORATIVE E2D is given in Algorithm 2, equivalent to Chen et al. (2022, Algorithm 2) in the known reward setting ($D_{\mathrm{RL}}^2$ becomes $D_H^2$ since we assumed that the reward is deterministic and known, so that the contribution from reward distance in $D_{\mathrm{RL}}^2$ becomes 0). Chen et al. (2022, Theorem F.1) showed that EXPLORATIVE E2D achieves the following estimation bound.

**Theorem H.1** (Chen et al. (2022), Theorem F.1)**.** *Given an $1/T$-optimistic cover $(\widetilde{\mathbb{P}}, \Theta_0)$ (c.f. Definition C.4) of the model class $\Theta$, Algorithm 2 with $\eta = 1/3$ achieves the following with probability at least $1 - \delta$:*

$$V_\star - V_{\theta^\star}(\widehat{\pi}_{\mathrm{out}}) \leqslant \overline{\mathrm{edec}}_\gamma(\Theta) + \frac{10\gamma}{T}[\log|\Theta_0| + 2\log(1/\delta) + 3],$$

*where $\overline{\mathrm{edec}}_\gamma$ is the Explorative DEC as defined in Section 4.2.*

---

**Algorithm 3** ALL-POLICY MODEL-ESTIMATION E2D(Chen et al., 2022)

1: **Input:** Model class $\Theta$, parameters $\gamma > 0$, $\eta \in (0, 1/2]$. An $1/T$-optimistic cover $(\widetilde{\mathbb{P}}, \Theta_0)$.
2: Initialize $\mu^1 = \mathrm{Unif}(\Theta_0)$.
3: **for** $t = 1, \dots, T$ **do**
4:   Set $(p_{\exp}^t, \mu_{\mathrm{out}}^t) = \arg\min_{(p_{\exp}, \mu_{\mathrm{out}}) \in \Delta(\Pi) \times \Delta(\Theta)} \widehat{V}_{\mathrm{me}, \gamma}^{\mu^t}(p_{\exp}, \mu_{\mathrm{out}})$, where

$$\widehat{V}_{\mathrm{me}, \gamma}^{\mu^t}(p_{\exp}, \mu_{\mathrm{out}}) := \sup_{\theta \in \Theta} \sup_{\bar{\pi} \in \Pi} \mathbb{E}_{\bar{\theta} \sim \mu_{\mathrm{out}}}\left[D_{\mathrm{TV}}\left(\mathbb{P}_\theta^{\bar{\pi}}, \mathbb{P}_{\bar{\theta}}^{\bar{\pi}}\right)\right] - \gamma \mathbb{E}_{\pi \sim p_{\exp}} \mathbb{E}_{\widehat{\theta}^t \sim \mu^t}\left[D_{\mathrm{H}}^2\left(\mathbb{P}_\theta^\pi, \mathbb{P}_{\widehat{\theta}^t}^\pi\right)\right].$$

5:   Sample $\pi^t \sim p_{\exp}^t$. Execute $\pi^t$ and observe $\tau^t$.
6:   Compute $\mu^{t+1} \in \Delta(\Theta_0)$ by

$$\mu^{t+1}(\theta) \propto_\theta \mu^t(\theta) \cdot \exp\left(\eta \log \widetilde{\mathbb{P}}_\theta^{\pi^t}(\tau^t)\right).$$

7: Compute $\overline{\mu}_{\mathrm{out}} = \frac{1}{T} \sum_{t=1}^T \mu_{\mathrm{out}}^t \in \Delta(\Theta)$.
8: **Output:** $\widehat{\theta} = \arg\min_{\theta \in \Theta} \sup_{\pi \in \Pi} \mathbb{E}_{\bar{\theta} \sim \overline{\mu}_{\mathrm{out}}}\left[D_{\mathrm{TV}}\left(\mathbb{P}_\theta^{\bar{\pi}}, \mathbb{P}_{\bar{\theta}}^{\bar{\pi}}\right)\right]$.

---

As we can see from the theorem above, as long as we can bound $\overline{\mathrm{edec}}_\gamma(\Theta)$, we can get a sample complexity bound for the EXPLORATIVE E2D algorithm. This gives Theorem 10 in the main text, which we restate as below.

**Theorem H.2** (Restatement of Theorem 10). *Suppose $\Theta$ is a PSR class with the same core test sets $\{\mathcal{U}_h\}_{h \in [H]}$, and each $\theta \in \Theta$ admits a B-representation that is $\Lambda_{\mathsf{B}}$-stable (c.f. Definition 4) or weakly $\Lambda_{\mathsf{B}}$-stable (c.f. Definition D.4), and has PSR rank $d_{\mathsf{PSR}} \leqslant d$. Then*

$$\overline{\mathrm{edec}}_\gamma(\Theta) \leqslant 9dAU_A\Lambda_{\mathsf{B}}^2 H^2/\gamma.$$

*Therefore, we can choose a suitable parameter $\gamma$ and an $1/T$-optimistic cover $(\widetilde{\mathbb{P}}, \Theta_0)$, such that with probability at least $1 - \delta$, Algorithm 2 outputs a policy $\widehat{\pi}_{\mathrm{out}} \in \Delta(\Pi)$ such that $V_\star - V_{\theta^\star}(\widehat{\pi}_{\mathrm{out}}) \leqslant \varepsilon$, as long as the number of episodes*

$$T \geqslant \mathcal{O}\left(dAU_A\Lambda_{\mathsf{B}}^2 H^2 \log(\mathcal{N}_\Theta(1/T)/\delta)/\varepsilon^2\right).$$

The proof of Theorem H.2 and hence Theorem 10 is contained in Appendix I.2.

## H.2 ALL-POLICY MODEL-ESTIMATION E2D FOR MODEL-ESTIMATION

In this section, we provide more details about model-estimation learning in PSRs as discussed in Section 4.2. In reward-free RL (Jin et al., 2020b), the goal is to optimally explore the environment without observing reward information, so that after the exploration phase, a near-optimal policy of any given reward can be computed using the collected trajectory data alone without further interacting with the environment.

Chen et al. (2022) developed ALL-POLICY MODEL-ESTIMATION E2D as a unified algorithm for reward-free/model-estimation learning in RL, and showed that its sample complexity scales with a complexity measure named All-policy Model-Estimation DEC (AMDEC). The AMDEC is defined as $\overline{\mathrm{amdec}}_\gamma(\Theta) := \sup_{\widehat{\mu} \in \Delta(\Theta)} \mathrm{amdec}_\gamma(\Theta, \widehat{\mu})$, where

$$\mathrm{amdec}_\gamma(\Theta, \widehat{\mu}) := \inf_{p_{\exp} \in \Delta(\Pi), \mu_{\mathrm{out}} \in \Delta(\Pi)} \sup_{\theta \in \Theta} \sup_{\bar{\pi} \in \Pi} \mathbb{E}_{\bar{\theta} \sim \mu_{\mathrm{out}}}\left[D_{\mathrm{TV}}\left(\mathbb{P}_\theta^{\bar{\pi}}, \mathbb{P}_{\bar{\theta}}^{\bar{\pi}}\right)\right] - \gamma \mathbb{E}_{\pi \sim p_{\exp}} \mathbb{E}_{\widehat{\theta} \sim \widehat{\mu}}\left[D_{\mathrm{H}}^2\left(\mathbb{P}_\theta^\pi, \mathbb{P}_{\widehat{\theta}}^\pi\right)\right].$$

(43)

The ALL-POLICY MODEL-ESTIMATION E2D algorithm (Algorithm 3) for a PSR class $\Theta$ is given as follows: In each episode $t \in [T]$, we maintain a distribution $\mu^t \in \Delta(\Theta_0)$ over an $1/T$-optimistic cover $(\widetilde{\mathbb{P}}, \Theta_0)$ of $\Theta$ (c.f. Definition C.4), which we use to compute an exploration policy distribution $p_{\exp}^t$ by minimizing the following risk:

$$(p_{\exp}^t, \mu_{\mathrm{out}}^t) = \arg\min_{(p_{\exp}, \mu_{\mathrm{out}}) \in \Delta(\Pi) \times \Delta(\Theta)} \sup_{\theta \in \Theta} \sup_{\bar{\pi} \in \Pi} \mathbb{E}_{\bar{\theta} \sim \mu_{\mathrm{out}}}\left[D_{\mathrm{TV}}\left(\mathbb{P}_\theta^{\bar{\pi}}, \mathbb{P}_{\bar{\theta}}^{\bar{\pi}}\right)\right] - \gamma \mathbb{E}_{\pi \sim p_{\exp}} \mathbb{E}_{\widehat{\theta}^t \sim \mu^t}\left[D_{\mathrm{H}}^2\left(\mathbb{P}_\theta^\pi, \mathbb{P}_{\widehat{\theta}^t}^\pi\right)\right].$$

Then, we execute policy $\pi^t \sim p_{\exp}^t$, collect trajectory $\tau^t$, and update the model distribution using the same *Tempered Aggregation* scheme as in EXPLORATIVE E2D. After $T$ episodes, we output the emipirical model $\widehat{\theta}$ by computing $\overline{\mu}_{\mathrm{out}} = \frac{1}{T} \sum_{t=1}^{T} \mu_{\mathrm{out}}^t \in \Delta(\Theta)$ and then projecting it into $\Theta$, i.e.

$$\widehat{\theta} = \arg\min_{\theta \in \Theta} \sup_{\pi \in \Pi} \mathbb{E}_{\bar{\theta} \sim \overline{\mu}_{\mathrm{out}}} \big[ D_{\mathrm{TV}} \left( \mathbb{P}_{\theta}^{\pi}, \mathbb{P}_{\bar{\theta}}^{\pi} \right) \big].$$

Chen et al. (2022, Theorem H.2) show that the output model $\widehat{\theta}$ of ALL-POLICY MODEL-ESTIMATION E2D has an estimation error (measured in terms of the TV distance) that scales as $\overline{\mathrm{amdec}}_\gamma$.

**Theorem H.3.** *Given an $1/T$-optimistic cover $(\widetilde{\mathbb{P}}, \Theta_0)$ (c.f. Definition C.4) of the class of transition dynamics $\Theta$, Algorithm 3 with $\eta = 1/2$ achieves the following with probability at least $1 - \delta$:*

$$\sup_{\pi} D_{\mathrm{TV}} \left( \mathbb{P}_{\widehat{\theta}}^{\pi}, \mathbb{P}_{\theta^\star}^{\pi} \right) \leqslant 6 \overline{\mathrm{amdec}}_\gamma(\Theta) + \frac{60\gamma}{T} [\log|\Theta_0| + 2\log(1/\delta) + 3],$$

*where $\overline{\mathrm{amdec}}_\gamma$ is the All-policy Model-Estimation DEC as defined in (43).*

We provide a sharp bound on the AMEDEC for B-stable PSRs, which implies that ALL-POLICY MODEL-ESTIMATION E2D can also learn them sample-efficient efficiently in a model-estimation manner.

**Theorem H.4.** *Suppose $\Theta$ is a PSR class with the same core test sets $\{\mathcal{U}_h\}_{h \in [H]}$, and each $\theta \in \Theta$ admits a B-representation that is $\Lambda_\mathsf{B}$-stable (c.f. Definition 4) or weakly $\Lambda_\mathsf{B}$-stable (c.f. Definition D.4), and has PSR rank $d_{\mathsf{PSR}} \leqslant d$. Then*

$$\overline{\mathrm{amdec}}_\gamma(\Theta) \leqslant 6 d A U_A \Lambda_\mathsf{B}^2 H^2 / \gamma. \tag{44}$$

*Therefore, we can choose a suitable parameter $\gamma$ and an $1/T$-optimistic cover $(\widetilde{\mathbb{P}}, \Theta_0)$, such that with probability at least $1 - \delta$, Algorithm 3 outputs a model $\widehat{\theta} \in \Theta$ such that $\sup_\pi D_{\mathrm{TV}} \left( \mathbb{P}_{\widehat{\theta}}^{\pi}, \mathbb{P}_{\theta^\star}^{\pi} \right) \leqslant \varepsilon$, as long as the number of episodes*

$$T \geqslant \mathcal{O} \big( d A U_A \Lambda_\mathsf{B}^2 H^2 \log(\mathcal{N}_\Theta(1/T)/\delta) / \varepsilon^2 \big).$$

The proof of Theorem H.4 is contained in Appendix I.3.

## H.3   MODEL-BASED OPTIMISTIC POSTERIOR SAMPLING (MOPS)

In this section, we provide more details about the MOPS algorithm as discussed in Section 4.3.

We consider the following version of the MOPS algorithm of Agarwal & Zhang (2022); Chen et al. (2022). Similar to EXPLORATIVE E2D, MOPS also maintains a posterior $\mu^t \in \Delta(\Theta_0)$ over an $1/T$ optimistic cover $(\widetilde{\mathbb{P}}, \Theta_0)$, initialized at a suitable prior $\mu^1$. The exploration policy in the $t$-th episode is obtained by posterior sampling: $\pi^t = \pi_{\theta^t} \circ_{h^t} \mathrm{Unif}(\mathcal{A}) \circ_{h^t+1} \mathrm{Unif}(\mathcal{U}_{A,h^t+1})$, where $\theta^t \sim \mu^t$ and $h^t \sim \mathrm{Unif}(\{0, 1, \ldots, H-1\})$. After executing $\pi^t$ and observing $\tau^t$, the algorithm updates the posterior as

$$\mu^{t+1}(\theta) \propto_\theta \mu^1(\theta) \exp \Big( \sum_{s=1}^{t} \big( \gamma^{-1} V_\theta(\pi_\theta) + \eta \log \widetilde{\mathbb{P}}_\theta^{\pi^s}(\tau^s) \big) \Big).$$

Finally, the algorithm output $\widehat{\pi}_{\mathrm{out}} := \frac{1}{T} \sum_{t=1}^{T} p_{\mathrm{out}}(\mu^t)$, where $p_{\mathrm{out}}(\mu^t) \in \Delta(\Pi)$ is defined as

$$p_{\mathrm{out}}(\mu)(\pi) = \mu(\{\theta : \pi_\theta = \pi\}), \qquad \forall \pi \in \Pi. \tag{45}$$

We further consider the following Explorative PSC (EPSC), which is a modification of the PSC proposed in Chen et al. (2022, Definition 4):

$$\mathrm{psc}_\gamma^{\mathrm{est}}(\Theta, \bar{\theta}) = \sup_{\mu \in \Delta_0(\Theta)} \mathbb{E}_{\theta \sim \mu} \big[ V_\theta(\pi_\theta) - V_{\bar{\theta}}(\pi_\theta) - \gamma \mathbb{E}_{\pi \sim \mu} \big[ D_{\mathrm{H}}^2 \left( \mathbb{P}_{\bar{\theta}}^{\pi_{\exp}}, \mathbb{P}_{\bar{\theta}}^{\pi_{\exp}} \right) \big] \big], \tag{46}$$

---

**Algorithm 4** MODEL-BASED OPTIMISTIC POSTERIOR SAMPLING (Agarwal & Zhang, 2022)

1: **Input:** Parameters $\gamma > 0$, $\eta \in (0, 1/2)$. An $1/T$-optimistic cover $(\widetilde{\mathbb{P}}, \Theta_0)$
2: **Initialize:** $\mu^1 = \text{Unif}(\Theta_0)$
3: **for** $t = 1, \dots, T$ **do**
4:     Sample $\theta^t \sim \mu^t$ and $h^t \sim \text{Unif}(\{0, 1, \cdots, H-1\})$.
5:     Set $\pi^t = \pi_{\theta^t} \circ_{h^t} \text{Unif}(\mathcal{A}) \circ_{h^t+1} \text{Unif}(\mathcal{U}_{A,h+1})$, execute $\pi^t$ and observe $\tau^t$.
6:     Compute $\mu^{t+1} \in \Delta(\Theta_0)$ by

$$\mu^{t+1}(\theta) \propto_\theta \mu^1(\theta) \exp\Big(\sum_{s=1}^{t}\big(\gamma^{-1}V_\theta(\pi_\theta) + \eta \log \widetilde{\mathbb{P}}_\theta^{\pi^s}(\tau^s)\big)\Big).$$

**Output:** Policy $\widehat{\pi}_{\text{out}} := \frac{1}{T}\sum_{t=1}^{T} p_{\text{out}}(\mu^t)$, where $p_{\text{out}}(\cdot)$ is defined in (45).

---

where $\Delta_0(\Theta)$ is the set of all finitely supported distributions on $\Theta$, $\pi_{\text{exp}}$ is defined as

$$\pi_{\text{exp}} = \frac{1}{H}\sum_{h=0}^{H-1} \pi \circ_h \text{Unif}(\mathcal{A}) \circ_{h+1} \text{Unif}(\mathcal{U}_{A,h+1}),$$

and we abbreviate $\pi \sim p_{\text{out}}(\mu)$ to $\pi \sim \mu$.

Adapting the proof for the MOPS algorithm in Chen et al. (2022, Corollary D.3 & Theorem D.1) to the explorative version, we can show that the output policy $\widehat{\pi}_{\text{out}}$ of MOPS has a sub-optimality gap that scales as $\text{psc}^{\text{est}}$.

**Theorem H.5.** *Given an $1/T$-optimistic cover $(\widetilde{\mathbb{P}}, \Theta_0)$ (c.f. Definition C.4) of the class of PSR models $\Theta$, Algorithm 4 with $\eta = 1/6$ and $\gamma \geq 1$ achieves the following with probability at least $1 - \delta$:*

$$V_\star - V_{\theta^\star}(\widehat{\pi}_{\text{out}}) \leq \text{psc}^{\text{est}}_{\gamma/6}(\Theta, \theta^\star) + \frac{2}{\gamma} + \frac{\gamma}{T}[\log|\Theta_0| + 2\log(1/\delta) + 5],$$

*where $\text{psc}^{\text{est}}_\gamma$ is the Explorative PSC as defined in (46).*

We provide a sharp bound on the EPSC for B-stable PSRs, which implies that MOPS can also learn them sample-efficient efficiently.

**Theorem H.6.** *Suppose $\Theta$ is a PSR class with the same core test sets $\{\mathcal{U}_h\}_{h\in[H]}$, and each $\theta \in \Theta$ admits a B-representation that is $\Lambda_{\text{B}}$-stable (c.f. Definition 4) or weakly $\Lambda_{\text{B}}$-stable (c.f. Definition D.4), and the ground truth model $\theta^\star$ has PSR rank at most $d$. Then*

$$\text{psc}^{\text{est}}_\gamma(\Theta, \theta^\star) \leq 6\Lambda_{\text{B}}^2 dAU_A H^2/\gamma.$$

*Therefore, we can choose a suitable parameter $\gamma$ and an $1/T$-optimistic cover $(\widetilde{\mathbb{P}}, \Theta_0)$, such that with probability at least $1 - \delta$, Algorithm 4 outputs a policy $\widehat{\pi}_{\text{out}} \in \Delta(\Pi)$ such that $V_\star - V_{\theta^\star}(\widehat{\pi}_{\text{out}}) \leq \varepsilon$, as long as the number of episodes*

$$T \geq \mathcal{O}\big(dAU_A\Lambda_{\text{B}}^2 H^2 \log(\mathcal{N}_\Theta(1/T)/\delta)/\varepsilon^2\big).$$

The proof of Theorem H.6 is contained in Appendix I.2. We remark here that EPSC provides an upper bound of EDEC (c.f. Eq. (55)), So Theorem H.2 (and hence Theorem 10) directly follows from Theorem H.6.

# I  PROOFS FOR APPENDIX H

For the clarity of discussion, we introduce the following notation in this section: for policy $\pi$, we denote $\varphi_h$ to be a policy modification such that

$$\varphi_h \diamond \pi = \pi \circ_h \text{Unif}(\mathcal{A}) \circ_{h+1} \text{Unif}(\mathcal{U}_{A,h+1}).$$

Again, here $\varphi_h \diamond \pi$ means that we follow $\pi$ for the first $h-1$ steps, takes $\text{Unif}(\mathcal{A})$ at step $h$, takes an action sequence sampled from $\text{Unif}(\mathcal{U}_{A,h+1})$ at step $h+1$, and behaves arbitrarily afterwards. Such definition agrees with (47). We further define the $\varphi$ policy modification as

$$\varphi \diamond \pi = \frac{1}{H}\sum_{h=0}^{H-1} \varphi_h \diamond \pi = \frac{1}{H}\sum_{h=0}^{H-1} \pi \circ_h \text{Unif}(\mathcal{A}) \circ_{h+1} \text{Unif}(\mathcal{U}_{A,h+1}). \tag{47}$$

We call $\varphi \diamond \pi$ the exploration policy of $\pi$.

## I.1 PROOF OF THEOREM H.6

To prove Theorem H.6, due to Theorem H.5, we only need to bound the coefficients $\mathrm{psc}_\gamma^{\mathrm{est}}(\Theta, \theta^\star)$. By its definition, we have

$$
\begin{aligned}
\mathrm{psc}_\gamma^{\mathrm{est}}(\Theta, \theta^\star) &= \sup_{\mu \in \Delta_0(\Theta)} \mathbb{E}_{\theta \sim \mu}\big[V_\theta(\pi_\theta) - V_{\theta^\star}(\pi_\theta) - \gamma \mathbb{E}_{\pi \sim \mu}\big[D_{\mathrm{H}}^2\big(\mathbb{P}_\theta^{\varphi \diamond \pi}, \mathbb{P}_{\theta^\star}^{\varphi \diamond \pi}\big)\big]\big] \\
&\leqslant \sup_{\mu \in \Delta_0(\Theta)} \mathbb{E}_{\theta \sim \mu}[D_{\mathrm{TV}}\left(\mathbb{P}_\theta^{\pi_\theta}, \mathbb{P}_{\theta^\star}^{\pi_\theta}\right)] - \gamma \mathbb{E}_{\theta \sim \mu} \mathbb{E}_{\pi \sim \mu}\big[D_{\mathrm{H}}^2\left(\mathbb{P}_\theta^{\varphi \diamond \pi}, \mathbb{P}_{\theta^\star}^{\varphi \diamond \pi}\right)\big].
\end{aligned}
\tag{48}
$$

We then invoke the following error decorrelation result, which follows from the decoupling argument in Appendix E.2 and Proposition F.1.

**Proposition I.1** (Error decorrelation). *Under the condition of Theorem H.2 (the same condition as Theorem H.6), for any $\mu \in \Delta_0(\Theta)$ and any reference model $\bar\theta \in \Theta$, we have*

$$
\mathbb{E}_{\theta \sim \mu}\big[D_{\mathrm{TV}}\left(\mathbb{P}_\theta^{\pi_\theta}, \mathbb{P}_{\bar\theta}^{\pi_\theta}\right)\big] \leqslant \sqrt{24 \Lambda_{\mathrm{B}}^2 d_{\bar\theta} A U_A H^2 \cdot \mathbb{E}_{\theta, \theta' \sim \mu}\big[D_{\mathrm{H}}^2\left(\mathbb{P}_\theta^{\varphi \diamond \pi_{\theta'}}, \mathbb{P}_{\bar\theta}^{\varphi \diamond \pi_{\theta'}}\right)\big]},
$$

*where $d_{\bar\theta}$ is the PSR rank of $\bar\theta$, $\varphi \diamond \pi$ defined in (47) is the exploration policy of $\pi$.*

Combining Proposition I.1 with (48) immediately gives the desired upper bound of $\mathrm{psc}_\gamma^{\mathrm{est}}(\Theta, \theta^\star)$, and thus completes the proof of Theorem H.6. □

We next turn to prove the Proposition I.1 above. We consider the following generalized version of Proposition I.1.

**Proposition I.2** (Generalized error decorrelation). *Under the condition of Theorem H.4, for any $\bar\theta \in \Theta$ $\nu \in \Delta_0(\Theta \times \Pi)$, we have*

$$
\mathbb{E}_{(\theta, \pi) \sim \nu}\big[D_{\mathrm{TV}}\left(\mathbb{P}_\theta^\pi, \mathbb{P}_{\bar\theta}^\pi\right)\big] \leqslant \sqrt{24 \Lambda_{\mathrm{B}}^2 d_{\bar\theta} A U_A H^2 \cdot \mathbb{E}_{\theta \sim \nu} \mathbb{E}_{\pi \sim \nu}\big[D_{\mathrm{H}}^2\left(\mathbb{P}_\theta^{\varphi \diamond \pi}, \mathbb{P}_{\bar\theta}^{\varphi \diamond \pi}\right)\big]},
$$

*where $\varphi \diamond \pi$ defined in (47) is the exploration policy of $\pi$.*

*Proof of Proposition I.2.* In the following, we fix a $\bar\theta \in \Theta$ and abbreviate $\overline{\mathcal{E}} = \mathcal{E}^{\bar\theta}, \overline{\mathbf{q}} = \mathbf{q}^{\bar\theta}$. Then, by Proposition F.1, we have

$$
\begin{aligned}
\mathbb{E}_{(\theta, \pi) \sim \mu}\big[D_{\mathrm{TV}}\left(\mathbb{P}_\theta^\pi, \mathbb{P}_{\bar\theta}^\pi\right)\big] &\leqslant \mathbb{E}_{(\theta, \pi) \sim \mu}\left[\overline{\mathcal{E}}_{\theta,0} + \sum_{h=1}^H \mathbb{E}_{\bar\theta, \pi}\big[\overline{\mathcal{E}}_{\theta, h}(\tau_{h-1})\big]\right] \\
&= \mathbb{E}_{\theta \sim \mu}\big[\overline{\mathcal{E}}_{\theta,0}\big] + \sum_{h=1}^H \mathbb{E}_{(\theta, \pi) \sim \mu} \mathbb{E}_{\bar\theta, \pi}\big[\overline{\mathcal{E}}_{\theta, h}(\tau_{h-1})\big].
\end{aligned}
\tag{49}
$$

Note that for the term $\mathbb{E}_{\theta \sim \mu}\big[\overline{\mathcal{E}}_{\theta,0}\big]$, we have

$$
\mathbb{E}_{\theta \sim \mu}\big[\overline{\mathcal{E}}_{\theta,0}\big] \leqslant \sqrt{\mathbb{E}_{\theta \sim \mu}\big[\overline{\mathcal{E}}_{\theta,0}^2\big]}.
\tag{50}
$$

We next consider the case for $h \in [H]$, and upper bound the corresponding terms in the right-hand-side of (49) using the decoupling argument introduced in Appendix E.2, restated as follows for convenience.

**Proposition E.6** (Decoupling argument). *Suppose we have vectors and functions*

$$
\{x_i\}_{i \in \mathcal{I}} \subset \mathbb{R}^n, \qquad \{f_\theta : \mathbb{R}^n \to \mathbb{R}\}_{\theta \in \Theta}
$$

*where $\Theta, \mathcal{I}$ are arbitrary abstract index sets, with functions $f_\theta$ given by*

$$
f_\theta(x) := \max_{r \in \mathcal{R}} \sum_{j=1}^J |\langle x, y_{\theta, j, r} \rangle|, \qquad \forall x \in \mathbb{R}^n,
$$

*where $\{y_{\theta, j, r}\}_{(\theta, j, r) \in \Theta \times [J] \times \mathcal{R}} \subset \mathbb{R}^n$ is a family of bounded vectors in $\mathbb{R}^n$. Then for any distribution $\mu$ over $\Theta$ and probability family $\{q_\theta\}_{\theta \in \Theta} \subset \Delta(\mathcal{I})$,*

$$
\mathbb{E}_{\theta \sim \mu} \mathbb{E}_{i \sim q_\theta} [f_\theta(x_i)] \leqslant \sqrt{d_X \mathbb{E}_{\theta, \theta' \sim \mu} \mathbb{E}_{i \sim q_{\theta'}} [f_\theta(x_i)^2]},
$$

*where $d_X$ is the dimension of the subspace of $\mathbb{R}^n$ spanned by $(x_i)_{i \in \mathcal{I}}$.*

We have the following three preparation steps to apply Proposition E.6:

1. Recall that $\overline{\mathcal{E}} = \mathcal{E}^{\bar{\theta}}$ is defined in Proposition F.1. Let us define

$$y_{\theta,j,\pi} := \frac{1}{2}\pi(\tau_{h:H}^j) \times \left[\mathbf{B}_{H:h+1}^\theta(\tau_{h+1:H}^j)\left(\mathbf{B}_h^\theta(o_h^j, a_h^j) - \mathbf{B}_h^{\bar{\theta}}(o_h^j, a_h^j)\right)\right]^\top \in \mathbb{R}^d,$$

where $\{\tau_{h:H}^j = (o_h^j, a_h^j, \cdots, o_H^j, a_H^j)\}_{j=1}^{n_y}$ is an ordering of all possible $\tau_{h:H}$ (and hence $n_y = (OA)^{H-h+1}$), $\pi$ is any policy (that starts at step $h$). We then define

$$f_\theta(x) = \max_\pi \sum_j |\langle y_{\theta,j,\pi}, x\rangle|, \qquad x \in \mathbb{R}^{|\mathcal{U}_h|}.$$

Then it follows from definition (c.f. Proposition F.1) that $\overline{\mathcal{E}}_{\theta,h}(\tau_{h-1}) = f_\theta(\overline{\mathbf{q}}(\tau_{h-1}))$.

2. We define $x_i = \mathbf{q}^\star(\tau_{h-1}^i) \in \mathbb{R}^{|\mathcal{U}_h|}$ for $i \in \mathcal{I} = (\mathcal{O} \times \mathcal{A})^{h-1}$ where $\{\tau_{h-1}^i\}_{i\in\mathcal{I}}$ is an ordering of all possible $\tau_{h-1} \in (\mathcal{O} \times \mathcal{A})^{h-1}$. Then by our definition of PSR rank (c.f. Definition 3), the subspace of $\mathbb{R}^{|\mathcal{U}_h|}$ spanned by $\{x_i\}_{i\in\mathcal{I}}$ has dimension less than or equal to $d_{\bar{\theta}}$.

3. We take $q_\theta \in \Delta(\mathcal{I})$ as

$$q_\theta(i) = \mathbb{E}_{\pi\sim\mu(\cdot|\theta)}\left[\mathbb{P}_{\bar{\theta}}^\pi(\tau_{h-1} = \tau_{h-1}^i)\right], \qquad i \in \mathcal{I} = (\mathcal{O} \times \mathcal{A})^{h-1}. \tag{51}$$

Therefore, applying Proposition E.6 to function family $\{f_\theta\}_{\theta\in\Theta}$, vector family $\{x_i\}_{i\in\mathcal{I}}$, and distribution family $\{q_\theta\}_{\theta\in\Theta}$ gives [21]

$$\mathbb{E}_{(\theta,\pi)\sim\mu}\left[\mathbb{E}_{\bar{\theta},\pi}\left[\overline{\mathcal{E}}_{\theta,h}(\tau_{h-1})\right]\right] = \mathbb{E}_{\theta\sim\mu}\mathbb{E}_{i\sim q_\theta}\left[f_\theta(x_i)\right]$$

$$\leqslant \sqrt{d_{\bar{\theta}}\mathbb{E}_{\theta,\theta'\sim\mu}\mathbb{E}_{i\sim q_{\theta'}}[f_\theta(x_i)^2]} = \sqrt{d_{\bar{\theta}}\mathbb{E}_{\theta\sim\mu}\mathbb{E}_{\pi\sim\mu}\left[\mathbb{E}_{\bar{\theta},\pi}\left[\overline{\mathcal{E}}_{\theta,h}^2(\tau_{h-1})\right]\right]}. \tag{52}$$

Combining Eq. (50), (52), and (49) yields

$$\mathbb{E}_{(\theta,\pi)\sim\mu}\left[D_{\mathrm{TV}}\left(\mathbb{P}_\theta^\pi, \mathbb{P}_{\bar{\theta}}^\pi\right)\right] \leqslant \mathbb{E}_{(\theta,\pi)\sim\mu}\left[\overline{\mathcal{E}}_{\theta,0}\right] + \sum_{h=1}^H \mathbb{E}_{(\theta,\pi)\sim\mu}\mathbb{E}_{\bar{\theta},\pi}\left[\overline{\mathcal{E}}_{\theta,h}(\tau_{h-1})\right]$$

$$\leqslant \sqrt{\mathbb{E}_{\theta\sim\mu}\left[\overline{\mathcal{E}}_{\theta,0}^2\right]} + \sum_{h=1}^H \sqrt{d_{\bar{\theta}}\,\mathbb{E}_{\theta,\pi\sim\mu}\left[\mathbb{E}_{\bar{\theta},\pi}\left[\overline{\mathcal{E}}_{\theta,h}^2(\tau_{h-1})\right]\right]}$$

$$\leqslant \sqrt{(Hd_{\bar{\theta}}+1)\left(\mathbb{E}_{\theta\sim\mu}\left[\overline{\mathcal{E}}_{\theta,0}^2\right] + \sum_{h=1}^H \mathbb{E}_{\theta,\pi\sim\mu}\left[\mathbb{E}_{\bar{\theta},\pi}\left[\overline{\mathcal{E}}_{\theta,h}^2(\tau_{h-1})\right]\right]\right)}$$

$$\leqslant \sqrt{(Hd_{\bar{\theta}}+1)\left(\mathbb{E}_{\theta,\pi\sim\mu}\left[\sum_{h=0}^{H-1} 12\Lambda_{\mathsf{B}}^2 A U_A \cdot D_{\mathrm{H}}^2\left(\mathbb{P}_\theta^{\varphi_h\diamond\pi}, \mathbb{P}_{\bar{\theta}}^{\varphi_h\diamond\pi}\right)\right]\right)}$$

$$= \sqrt{12(Hd_{\bar{\theta}}+1)H \cdot \Lambda_{\mathsf{B}}^2 A U_A \mathbb{E}_{\theta,\pi\sim\mu}\left[D_{\mathrm{H}}^2\left(\mathbb{P}_\theta^{\varphi\diamond\pi}, \mathbb{P}_{\bar{\theta}}^{\varphi\diamond\pi}\right)\right]},$$

where the third inequality is due to Cauchy-Schwarz inequality, and the fourth inequality is due to Proposition F.2. This completes the proof of Proposition I.1. $\qquad\square$

## I.2 PROOF OF THEOREM H.2 (THEOREM 10)

According to Theorem H.1, in order to prove Theorem H.2 (Theorem 10), we only need to bound the coefficients $\overline{\mathrm{edec}}_\gamma(\Theta)$ for $\gamma > 0$.

In the following, we bound $\overline{\mathrm{edec}}$ by $\mathrm{psc}^{\mathrm{est}}$ using the idea of Chen et al. (2022, Proposition 6). Recall that $\overline{\mathrm{edec}}$ is defined in (4.2). By strong duality (c.f. Theorem C.2), we have

$$\mathrm{edec}_\gamma(\Theta, \overline{\mu})$$

---

[21]The boundedness of $\{y_{\theta,j,\pi}\}$ is trivially satisfied, because $\mu_0$ is finitely supported.

$$\begin{aligned}
&:= \inf_{\substack{p_{\exp}\in\Delta(\Pi) \\ p_{\text{out}}\in\Delta(\Pi)}} \sup_{\theta\in\Theta} \mathbb{E}_{\pi\sim p_{\text{out}}}\left[V_\theta(\pi_\theta) - V_\theta(\pi)\right] - \gamma\mathbb{E}_{\bar{\theta}\sim\overline{\mu}}\mathbb{E}_{\pi\sim p_{\exp}}\left[D_H^2(\mathbb{P}_\theta^\pi,\mathbb{P}_{\bar{\theta}}^\pi)\right] \\
&= \sup_{\mu\in\Delta_0(\Theta)} \inf_{\substack{p_{\exp}\in\Delta(\Pi) \\ p_{\text{out}}\in\Delta(\Pi)}} \mathbb{E}_{\theta\sim\mu}\mathbb{E}_{\pi\sim p_{\text{out}}}\left[V_\theta(\pi_\theta) - V_\theta(\pi)\right] - \gamma\mathbb{E}_{\theta\sim\mu}\mathbb{E}_{\bar{\theta}\sim\overline{\mu}}\mathbb{E}_{\pi\sim p_{\exp}}\left[D_H^2(\mathbb{P}_\theta^\pi,\mathbb{P}_{\bar{\theta}}^\pi)\right].
\end{aligned}$$
$$(53)$$

Note that $|V_\theta(\pi) - V_{\bar{\theta}}(\pi)| \leqslant D_{\text{TV}}\left(\mathbb{P}_\theta^\pi,\mathbb{P}_{\bar{\theta}}^\pi\right) \leqslant D_H\left(\mathbb{P}_\theta^\pi,\mathbb{P}_{\bar{\theta}}^\pi\right)$. Therefore, we can take $p_{\text{out}} = p_\mu$, where $p_\mu$ is defined as $p_\mu(\pi) = \mu(\{\theta : \pi_\theta = \pi\})$. Then for a fixed $\alpha \in (0,1)$, we have

$$\begin{aligned}
&\mathbb{E}_{\theta\sim\mu}\mathbb{E}_{\pi\sim p_\mu}\left[V_\theta(\pi_\theta) - V_\theta(\pi)\right] \\
&\leqslant \mathbb{E}_{\theta\sim\mu}\mathbb{E}_{\bar{\theta}\sim\overline{\mu}}\mathbb{E}_{\pi\sim p_\mu}\left[D_H\left(\mathbb{P}_\theta^\pi,\mathbb{P}_{\bar{\theta}}^\pi\right)\right] + \mathbb{E}_{\theta\sim\mu}\mathbb{E}_{\bar{\theta}\sim\overline{\mu}}\mathbb{E}_{\pi\sim p_\mu}\left[V_\theta(\pi_\theta) - V_{\bar{\theta}}(\pi)\right] \\
&= \mathbb{E}_{\theta\sim\mu}\mathbb{E}_{\bar{\theta}\sim\overline{\mu}}\mathbb{E}_{\pi\sim p_\mu}\left[D_H\left(\mathbb{P}_\theta^\pi,\mathbb{P}_{\bar{\theta}}^\pi\right)\right] + \mathbb{E}_{\theta\sim\mu}\mathbb{E}_{\bar{\theta}\sim\overline{\mu}}\left[V_\theta(\pi_\theta) - V_{\bar{\theta}}(\pi_\theta)\right] \\
&\leqslant \frac{1}{4(1-\alpha)\gamma} + \gamma\mathbb{E}_{\theta\sim\mu}\mathbb{E}_{\bar{\theta}\sim\overline{\mu}}\mathbb{E}_{\pi\sim p_\mu}\left[D_H^2\left(\mathbb{P}_\theta^\pi,\mathbb{P}_{\bar{\theta}}^\pi\right)\right] + \mathbb{E}_{\theta\sim\mu}\mathbb{E}_{\bar{\theta}\sim\overline{\mu}}\left[V_\theta(\pi_\theta) - V_{\bar{\theta}}(\pi_\theta)\right],
\end{aligned}$$
$$(54)$$

where the equality is due to our choice of $p_\mu$:

$$\mathbb{E}_{\theta\sim\mu}\mathbb{E}_{\pi\sim p_\mu}\left[V_{\bar{\theta}}(\pi)\right] = \mathbb{E}_{\pi\sim p_\mu}\left[V_{\bar{\theta}}(\pi)\right] = \mathbb{E}_{\theta\sim\mu}\left[V_{\bar{\theta}}(\pi_\theta)\right],$$

and the last inequality is due to AM-GM inequality.

Therefore, we can take $p_{\exp} = \alpha p_\mu + (1-\alpha)p_e \in \Delta(\Pi)$, where $p_e$ is given by $p_e(\pi) = \mu(\{\theta : \varphi \diamond \pi_\theta = \pi\})$,[22] and using this choice of $p_{\exp}$ and $p_{\text{out}}$ in Eq. (53) and using Eq. (54), we get

$$\begin{aligned}
\text{edec}_\gamma(\Theta,\overline{\mu}) &\leqslant \sup_{\mu\in\Delta_0(\Theta)}\Big\{\mathbb{E}_{\bar{\theta}\sim\overline{\mu}}\big[\mathbb{E}_{\theta\sim\mu}\left[V_\theta(\pi_\theta) - V_{\bar{\theta}}(\pi_\theta)\right] \\
&\quad - \alpha\gamma\mathbb{E}_{\theta\sim\mu}\mathbb{E}_{\pi\sim\mu}\left[D_H^2(\mathbb{P}_\theta^{\varphi\diamond\pi},\mathbb{P}_{\bar{\theta}}^{\varphi\diamond\pi})\right]\big]\Big\} + \frac{1}{4(1-\alpha)\gamma} \\
&\leqslant \max_{\bar{\theta}\in\Theta}\text{psc}_{\alpha\gamma}^{\text{est}}(\Theta,\bar{\theta}) + \frac{1}{4(1-\alpha)\gamma}.
\end{aligned}$$
$$(55)$$

Recall that $\text{psc}_\gamma^{\text{est}}$ has been bounded in Theorem H.6. Taking $\alpha = 3/4$ yields $\text{edec}_\gamma(\Theta,\overline{\mu}) \leqslant (8\Lambda_{\text{B}}^2 dAU_A H^2 + 1)/\gamma$. This completes the proof of Theorem H.2. $\qquad\square$

## I.3 Proof of Theorem H.4

To prove Theorem H.4, due to Theorem H.3, we only need to bound the coefficients $\text{amdec}_\gamma(\Theta,\widehat{\mu})$ for all $\widehat{\mu}\in\Delta(\Theta)$. By strong duality (c.f. Theorem C.2), we have

$$\begin{aligned}
\text{amdec}_\gamma(\Theta,\widehat{\mu}) &= \inf_{p_{\exp}\in\Delta(\Pi),\mu_{\text{out}}\in\Delta(\Pi)} \sup_{\theta\in\Theta}\sup_{\bar{\pi}\in\Pi} \mathbb{E}_{\bar{\theta}\sim\mu_{\text{out}}}\big[D_{\text{TV}}\left(\mathbb{P}_\theta^{\bar{\pi}},\mathbb{P}_{\bar{\theta}}^{\bar{\pi}}\right)\big] - \gamma\mathbb{E}_{\pi\sim p_{\exp}}\mathbb{E}_{\widehat{\theta}\sim\widehat{\mu}}\Big[D_H^2\left(\mathbb{P}_\theta^\pi,\mathbb{P}_{\widehat{\theta}}^\pi\right)\Big] \\
&= \sup_{\nu\in\Delta_0(\Theta\times\Pi)} \inf_{p_{\exp}\in\Delta(\Pi),\mu_{\text{out}}\in\Delta(\Pi)} \mathbb{E}_{(\theta,\bar{\pi})\sim\nu}\mathbb{E}_{\bar{\theta}\sim\mu_{\text{out}}}\big[D_{\text{TV}}\left(\mathbb{P}_\theta^{\bar{\pi}},\mathbb{P}_{\bar{\theta}}^{\bar{\pi}}\right)\big] - \gamma\mathbb{E}_{\pi\sim p_{\exp}}\mathbb{E}_{\theta\sim\nu,\widehat{\theta}\sim\widehat{\mu}}\Big[D_H^2\left(\mathbb{P}_\theta^\pi,\mathbb{P}_{\widehat{\theta}}^\pi\right)\Big] \\
&\leqslant \sup_{\nu\in\Delta_0(\Theta\times\Pi)} \inf_{p_{\exp}\in\Delta(\Pi)} \mathbb{E}_{\bar{\theta}\sim\widehat{\mu}}\big[\mathbb{E}_{(\theta,\bar{\pi})\sim\nu}\big[D_{\text{TV}}\left(\mathbb{P}_\theta^{\bar{\pi}},\mathbb{P}_{\bar{\theta}}^{\bar{\pi}}\right)\big] - \gamma\mathbb{E}_{\pi\sim p_{\exp}}\mathbb{E}_{\theta\sim\nu}\big[D_H^2\left(\mathbb{P}_\theta^\pi,\mathbb{P}_{\bar{\theta}}^\pi\right)\big]\big] \\
&\leqslant \sup_{\nu\in\Delta_0(\Theta\times\Pi)} \mathbb{E}_{\bar{\theta}\sim\widehat{\mu}}\big[\mathbb{E}_{(\theta,\bar{\pi})\sim\nu}\big[D_{\text{TV}}\left(\mathbb{P}_\theta^{\bar{\pi}},\mathbb{P}_{\bar{\theta}}^{\bar{\pi}}\right)\big] - \gamma\mathbb{E}_{\pi\sim\nu}\mathbb{E}_{\theta\sim\nu}\big[D_H^2\left(\mathbb{P}_\theta^{\varphi\diamond\pi},\mathbb{P}_{\bar{\theta}}^{\varphi\diamond\pi}\right)\big]\big] \\
&\leqslant \sup_{\nu\in\Delta_0(\Theta\times\Pi)} \sup_{\bar{\theta}\in\Theta} \mathbb{E}_{(\theta,\bar{\pi})\sim\nu}\big[D_{\text{TV}}\left(\mathbb{P}_\theta^{\bar{\pi}},\mathbb{P}_{\bar{\theta}}^{\bar{\pi}}\right)\big] - \gamma\mathbb{E}_{\pi\sim\nu}\mathbb{E}_{\theta\sim\nu}\big[D_H^2\left(\mathbb{P}_\theta^{\varphi\diamond\pi},\mathbb{P}_{\bar{\theta}}^{\varphi\diamond\pi}\right)\big],
\end{aligned}$$

where the first inequality is because we take $\mu_{\text{out}} = \widehat{\mu}$ in $\inf_{\mu_{\text{out}}}$, and the second inequality is because we can take $p_{\exp}\in\Delta(\Pi)$ corresponds to $\varphi\diamond\pi$ with $\pi\sim\nu$. Applying Proposition I.2 gives

$$\text{amdec}_\gamma(\Theta,\widehat{\mu}) \leqslant \sup_{\bar{\theta}\in\Theta} \frac{6\Lambda_{\text{B}}^2 d_{\bar{\theta}} AU_A H^2}{\gamma} \leqslant \frac{6\Lambda_{\text{B}}^2 dAU_A H^2}{\gamma},$$

and thus the proof of Theorem H.4. $\qquad\square$

---

[22]Here, $p_e$ is technically a distribution over the set of mixed policies $\Delta(\Pi)$, and can be identified with a mixed policy in $\Delta(\Pi)$.

