# OpenReview forum: "Partially Observable RL with B-Stability: Unified Structural Condition and Sharp Sample-Efficient Algorithms"
_ICLR.cc/2023/Conference — ICLR 2023 notable top 25%_

### Official Review · Reviewer_jF7R · 2022-10-23

**Confidence:** 2
**Correctness:** 4
**Technical Novelty And Significance:** 3
**Empirical Novelty And Significance:** Not applicable
**Recommendation:** 6

**Clarity, Quality, Novelty And Reproducibility:**

The paper is clear and well organized. The contribution is incremental  --- which by no means a negative comment: building on the body of previous work is, in my opinion, an advantage of this research. However, since the paper is theoretical and most of the results can only be evaluated by careful verification of proofs in the appendices, 'reproducibility' is questionable (is reproducibility of a theoretical paper means being able to prove the results from the paper alone?)

**Strength And Weaknesses:**

Strengths: the paper is clearly written, gives a detailed account of related work and necessary background, and introduces a new property, B-stability, which is intuitive and possessed by several important classes of partially observable RL problems. This opens up directions for research related to details of proposed algorithms, learning of B-stable PSRs, and computational issues.

Weaknesses: the paper is of theoretical nature, and as such relies on definitions and proofs of theorems related to B-stability and algorithms. The proofs are not a part of the body of the paper but rather of the appendices, and the full length to be read to understand and check correctness is in excess of 40 pages. I tried to dive into the appendices but didn't have enough time and cannot confirm that the proofs are correct.  My humble opinion is that the paper, with all the contributions which it claims, is perhaps to long for a conference, either the reviewing process or presentation later on.

**Summary Of The Paper:**

This paper addresses theoretical aspects of sample based inference in partially observable RL problems. In particular, the paper adopts PSR formalism, and proposes B-stability, a structural condition that subsumes several common tractable practically observable RL problems. The paper shows  several algorithms for B-stable PSRs and argues that they have sharp sample complexities for this class of problems.

**Summary Of The Review:**

The paper is interesting, inspiring, and well written, however I am wondering whether it is the right paper for a conference. I cannot assess correctness of claims based on the paper only, and did not have enough time to go thoroughly through the proofs in the appendices. I would prefer a conference paper to be self-contained rather than an extended abstract of appendices significantly exceeding the paper in size.

---

> ### Author Response · Authors · 2022-11-15
> **Response to Reviewer jF7R**
>
> We thank the reviewer for the valuable feedback to our paper. We respond to the main concerns as follows.
>
> *--- “The proofs are not a part of the body of the paper but rather of the appendices, and the full length to be read to understand and check correctness is in excess of 40 pages”*
>
> We agree our results are theoretical in nature and require all the detailed proofs in the appendix. We have included a one-page “Overview of Techniques” in Appendix B, which outlines the main steps of the proofs (including their improvements over existing techniques) and gives pointers to the key lemmas. (We would have loved to include that in the main text if there was one more page for the space limit :) A shorter version of this overview is presented at the bottom of Page 7, after Theorem 9.
>
> We hope those could at least help the readers get a gist of the proof steps as well as their relationship with existing work, and further help save some time for reading the actual proofs.

---

### Official Review · Reviewer_PmjQ · 2022-10-24

**Confidence:** 4
**Correctness:** 3
**Technical Novelty And Significance:** 3
**Empirical Novelty And Significance:** Not applicable
**Recommendation:** 8

**Clarity, Quality, Novelty And Reproducibility:**

This paper is mainly theoretical. The main results seem technically sound. Detailed proofs are provided in Appendix. The idea of developing a unified theoretical framework for online learning in PSRs is certainly novel, and could have impact across disciplines in machine learning.

**Strength And Weaknesses:**

This paper provides a unified theoretical framework for online learning in PSRs. The proposed framework, called B-stability, subsumes several known tractable subclasses of models in the literature. I have not read through all details of the proof. However, the PAC results in Theorem 9 seem reasonable. Given the importance of latent representation learning and the relevance of partially observable RL, the unified theoretical framework in this paper could have an impact across different fields of machine learning.

As for the weakness, this paper is certainly notation heavy. While some of them might be inevitable given the technical nature of this paper, some of the notations are not well defined. For instance, it would be appreciated if the authors could further elaborate on Step 6 of OMLE. It is unclear how the exploratory policy is being set.

**Summary Of The Paper:**

This paper studies online reinforcement learning in the general setting of Predictive State Representations (PSRs) with a finite horizon. The authors propose a unified condition for PSRs called B-stability, which encompasses the vast majority of known tractable subclasses such as weakly revealing POMDPs, low-rank future-sufficient POMDPs, decodable POMDPs, and regular PSRs. The authors then provide novel online RL algorithms that could learn B-stable PSR with polynomial samples in relevant problem parameters and can outperform existing methods in the aforementioned subclasses.

**Summary Of The Review:**

This paper provides a unified theoretical framework for online learning in PSRs. The proposed framework, called B-stability, subsumes several known tractable subclasses of models in the literature. Given the importance of latent representation learning and the relevance of partially observable RL, the unified treatment presented in this paper could have an impact across different fields of machine learning.

---

> ### Author Response · Authors · 2022-11-15
> **Response to Reviewer PmjQ**
>
> We thank the reviewer for the valuable feedback to our paper. We respond to the main concerns as follows.
>
> *--- “For instance, it would be appreciated if the authors could further elaborate on Step 6 of OMLE. It is unclear how the exploratory policy is being set..”*
>
> The exploration policy $\pi_{h,{\rm exp}}^k$ appearing in Step 6 is defined via the $\circ_h$ notation as follows: Take $\pi^k$ for the first $h-1$ steps, take a uniform action ${\rm Unif}(\mathcal{A})$ at step $h$, take an action sequence sampled from ${\rm Unif}(\mathcal{U}_{A,h+1})$ at step $h+1$, and behave arbitrarily afterwards. This definition was presented in the text “2. (Data collection)” after Algorithm 1.
>
> To highlight this definition and avoid confusions, we have slightly changed the text to highlight where we define $\pi_{h,{\rm exp}}^k$ (together with the $\circ_h$ notation) in our revision.

---

### Official Review · Reviewer_PDoA · 2022-10-25

**Confidence:** 4
**Correctness:** 4
**Technical Novelty And Significance:** 3
**Empirical Novelty And Significance:** Not applicable
**Recommendation:** 8

**Clarity, Quality, Novelty And Reproducibility:**

The writing is in general very clear. But the paper could be hard for broader audience to read. In terms of originality, I feel that the paper could have emphasized more on technical innovations.

**Strength And Weaknesses:**


*Strength*

- The paper is very comprehensive -- the framework is quite general and can imply many previous results with improved upper bounds.

- It is interesting to see the connections with DEC.




*Weaknesses*

- In a current form, the paper sounds in nature too similar to (Zhan et al., 2022). It would have been better to position the paper by, for example, focusing more on the technical innovations to get better sample-complexity.

- The term "sharp" is used in several places to emphasize that the upper bound analysis is improved from previous work, but I think this term is somewhat conventionally used when the upper bound matches the lower bound which is not discussed in the paper.

- It is somewhat hard to appreciate the proposed B-stability condition (maybe it is too abstract? or only algebraically explained?). This is a personal opinion but impacts my overall impression of the paper. If there is a better and more intuitive explanation for this, it would be great.

**Summary Of The Paper:**

The submission considers a sample-efficient learning for PSRs, extending some recent works by (Liu et al., 2022) from POMDPs to PSRs. In some sense, this paper marginally extends a recent work (Zhan et al., 2022) which already claims almost the same contribution. The suggested B-stability is a slightly more general condition than the regularity condition suggested in (Zhan et al., 2022). Just like (Zhan et al., 2022), the algorithm of interest is O-MLE (Liu et al., 2022), and results obtained for PSR can imply several previous results. Proof ideas and techniques are also mostly adopted from (Liu et al., 2022), though there are some differences in technical details from (Zhan et al., 2022).

I think the real contribution of this paper could be an improved sample-complexity analysis of O-MLE, which improves some polynomial factors from previous bounds. The paper also makes connection to decision-estimation coefficient (DEC) first proposed by (Foster et al., 2021), and propose another posterior-sampling based algorithm in parallel. This is also interesting on its own, though slightly off from the main flow of the paper in my opinion.

**Summary Of The Review:**

Overall, I think the paper provides very comprehensive and advanced results which could be of significant interest to RL theory community.

---

> ### Author Response · Authors · 2022-11-15
> **Response to Reviewer PDoA**
>
> We thank the reviewer for the valuable feedback to our paper. We respond to the main concerns as follows.
>
> *--- “In a current form, the paper sounds in nature too similar to (Zhan et al., 2022). It would have been better to position the paper by, for example, focusing more on the technical innovations to get better sample-complexity.”*
>
> We agree that the technical innovations to obtain the improved sample complexity is an important part of our contribution. We have a brief overview of techniques at the bottom of Page 7 (after Theorem 9), with a more elaborated version in Appendix B, which presents the main steps (Performance decomposition, Bounding the squared-B errors, and generalized L2 Eluder argument) and discusses how they improve over existing work. In the rest of the main text, we chose to focus on other ingredients (the B-stability condition and algorithms) due to their necessity and the space limit.
>
> We also briefly remark that, apart from the sample complexity improvement, even just as a condition for learnability with polynomial samples, our B-stability condition is also more expressive than Zhan et al.'s “regular PSR” condition (see the paragraph on Page 6 after Example 8 for detailed discussions).
>
> *--- “The term "sharp" is used in several places to emphasize that the upper bound analysis is improved from previous work, but I think this term is somewhat conventionally used when the upper bound matches the lower bound which is not discussed in the paper.”*
>
> We indeed mainly used "sharp" to indicate our substantial sample complexity improvement over existing results. We agree that obtaining a lower bound for our problem (learning POMDPs / PSRs), as well as understanding its gap with the current upper bounds, is an important question. Currently, a (non-trivial) lower bound for learning tractable subclasses of POMDPs / PSRs (such as revealing / decodable POMDPs) is still largely open, which we believe is beyond the scope of the current paper but an important question for future work.
>
> *--- “ [...] somewhat hard to appreciate the proposed B-stability condition (maybe it is too abstract? or only algebraically explained?). If there is a better and more intuitive explanation for this, it would be great.”*
>
> Intuitively, the B-operator $\mathcal{B}_ {H:h}$ has the following probabilistic meaning: It maps any predictive state $\mathbf{q}=\mathbf{q}(\tau_{h-1})$
> to the vector
> $\mathcal{B}_ {H:h}\mathbf{q} = (\mathbb{P}(\tau_ {h:H}|\tau_ {h-1}))_ {\tau_ {h:H}}$
> , which governs the probability of transitioning to all possible futures (i.e. $\mathbb{P}(\tau_ {h:H}|\tau_ {h-1}) \times \pi(\tau_ {h:H}|\tau_{h-1}) = \mathbb{P}^\pi(\tau_ {h:H}|\tau_ {h-1})$ for any policy $\pi$).
>
> The B-stability condition in turn just requires the above B-operator to have a bounded operator norm, so that small errors in the input $\mathbf{q}_ 1-\mathbf{q}_ 2$ will result in the small error in the output $\mathcal{B}_ {H:h}(\mathbf{q}_1-\mathbf{q}_2)$, in suitable norms.
>
> To better illustrate this, we have added a short explanation of the above intuition for the B-operators at the beginning of Page 5 (Section 3.1) in our revision.

---

### Author Response · Authors · 2022-11-15
**Revision Uploaded**

We thank all reviewers again for their valuable feedback to our paper.

We have incorporated the reviewers’ suggestions and made several edits on the presentation in our revision. All changes (except for typo fixes) are marked in red, for clarity.

---

### Decision · Program_Chairs · 2023-01-20

**Decision:**

Accept: notable-top-25%

**Justification For Why Not Higher Score:**


The paper is a theoretical paper to provide a unified condition, under which the RL is sample efficient in the partial observable environment.

The paper is rigorous and comprehensive to cover several known sample-efficient subclasses of POMDP, and improves the known results with three different algorithms.

**Justification For Why Not Lower Score:**


The presentation of the paper can be further improved to emphasize the significance of the contribution, comparing to the existing literature (Zhan et al., 2022).

**Metareview: Summary, Strengths And Weaknesses:**


In this paper, the authors investigated the unified structural condition, i.e., B-stability, under which the PSR can be learned sample-efficiently. The authors also introduced the connection between the B-stable PSR and the learnable subclasses of POMDP and provided tighter bounds. Finally, the authors proved that these bounds can be achieved by three different algorithms.


The paper is well-organized and clear. As a theoretical paper, this paper is comprehensive in the sense that it covers a variety of models and plenty of algorithms. All of the reviewers agree this paper should be accepted.


Minor:

- The current version is relatively similar to (Zhan et al., 2022). It will be better if the paper can be re-organized to emphasize the novelty and significance.


**Note From Pc:**

if the above contains the word "oral" or "spotlight" please see: "oral" presentation means -> notable-top-5% and "spotlight" means -> notable-top-25%. As stated in our emails, we are disassociating presentation type from AC recommendations